# Dual Quaternion SE(3) Synchronization with Recovery Guarantees

Jianing Zhao [*1]  Linglingzhi Zhu [*2]  Anthony Man-Cho So [1]

## Abstract

Synchronization over the special Euclidean group SE(3) aims to recover absolute poses from noisy pairwise relative transformations and is a core primitive in robotics and 3D vision. Standard approaches often require multi-step heuristic procedures to recover valid poses, which are difficult to analyze and typically lack theoretical guarantees. This paper adopts a dual quaternion representation and formulates SE(3) synchronization directly over the unit dual quaternion. A two-stage algorithm is developed: A spectral initializer computed via the power method on a Hermitian dual quaternion measurement matrix, followed by a dual quaternion generalized power method (DQGPM) that enforces feasibility through per-iteration projection. The estimation error bounds are established for spectral estimators, and DQGPM is shown to admit a finite-iteration error bound and achieves linear error contraction up to an explicit noise-dependent threshold. Experiments on synthetic benchmarks and real-world multi-scan point-set registration demonstrate that the proposed pipeline improves both accuracy and efficiency over representative matrix-based methods.

## 1. Introduction

Global coordinate alignment, the task of unifying a network of local reference frames into a single and consistent global coordinate system, lies at the heart of 3D spatial perception. Whether reconstructing a trajectory in robotics or merging partial 3D scans in computer vision, the fundamental challenge is to recover the absolute poses (rotations

---
[*]Equal contribution [1]Department of Systems Engineering and Engineering Management, The Chinese University of Hong Kong, Shatin, NT, Hong Kong [2]H. Milton Stewart School of Industrial and Systems Engineering, Georgia Institute of Technology, Atlanta, GA, USA. Correspondence to: Linglingzhi Zhu <llzzhu@gatech.edu>.

*Proceedings of the 43rd International Conference on Machine Learning*, Seoul, South Korea. PMLR 306, 2026. Copyright 2026 by the author(s).

and translations) of a set of nodes given only noisy measurements of their relative transformations. Mathematically formalized as synchronization over the special Euclidean group SE(3), this problem serves as the foundation for diverse applications, including simultaneous localization and mapping (SLAM) (Liu et al., 2012; Rosen et al., 2019; Fan & Murphey, 2022), multiple point-set registration (Fusiello et al., 2002; Sharp et al., 2004; Arrigoni et al., 2016a), and structure from motion (SfM) (Govindu, 2004; Martinec & Pajdla, 2007). A prime example is multiple point-set registration, where the objective is to align a collection of 3D scans into a unified coordinate system. While early point-based approaches (Benjemaa & Schmitt, 1998; Zhou et al., 2016; Han et al., 2019) attempted to optimize point-to-point correspondences directly, they often suffer from prohibitive computational costs and susceptibility to local minima when initializations are poor. Consequently, modern pipelines favor frame-space methods (Chaudhury et al., 2015; Arrigoni et al., 2016a). These approaches decouple the problem into two steps: first, estimating pairwise relative transformations (e.g., via Iterative Closest Point (Besl & McKay, 1992)); second, globally synchronizing these noisy measurements to recover absolute poses. This abstraction effectively casts the registration task as an SE(3) synchronization problem (Jiang et al., 2013; Özyesil & Singer, 2015).

Consequently, the core task reduces to recovering the rigid motions (rotations and translations) that best satisfy a network of noisy relative constraints. Standard approaches typically represent SE(3) elements using matrix embeddings:

$$\begin{pmatrix} \boldsymbol{R} & \boldsymbol{t} \\ \boldsymbol{0}^\top & 1 \end{pmatrix} \in \mathbb{R}^{4 \times 4},$$

where $\boldsymbol{R} \in \mathrm{SO}(3)$ and $\boldsymbol{t} \in \mathbb{R}^3$. To address the nonconvexity of the rank and orthogonality constraints, spectral relaxation (Arie-Nachimson et al., 2012; Arrigoni et al., 2016b) and semidefinite relaxation (SDR) (Liu et al., 2012; Chaudhury et al., 2015; Rosen et al., 2019) are widely employed. Additionally, Lie-algebraic averaging techniques (Govindu, 2004) and Riemannian optimization methods (Tron & Daniilidis, 2014) have also been applied to recover rigid motions.

However, matrix-based representations suffer from a fundamental limitation on the representation gap. Ideally, a group representation should align the isometry group of the

*Table 1.* Rounding and Eigenspace–Synchronization Gap Across Groups

| Group | Rounding / Projection | Eigenspace–Sync Gap[*] |
|-------|----------------------|------------------------|
| $\mathrm{SO}(2)$ | $z_i \leftarrow z_i/|z_i|$ | $\mathrm{Iso}(E) \cong \mathrm{SO}(2) \Rightarrow$ **No gap** |
| $\mathrm{SO}(d)$ | $\boldsymbol{G}_i \leftarrow \boldsymbol{U}_i \operatorname{diag}(1, \ldots, 1, \det(\boldsymbol{U}_i \boldsymbol{V}_i^\top)) \, \boldsymbol{V}_i^\top$ | $\mathrm{Iso}(E) \cong \mathrm{O}(d) \Rightarrow$ **Moderate gap** |
| $\mathrm{SE}(3)$ | Multi-step heuristic | $\mathrm{Iso}(E)$ cannot encode translations $\Rightarrow$ **Substantial mismatch** |

[*] For the dominant eigenspace $E$, $\mathrm{Iso}(E)$ denotes the group of linear isometries.

nonzero eigenspace with the intrinsic global gauge symmetry of the synchronization problem. This alignment is essentially perfect for $\mathrm{SO}(2)$ under the unit complex representation: the eigenspace isometry matches $\mathrm{SO}(2)$, making rounding a trivial normalization. For $\mathrm{SO}(d)$, the eigenspace isometry expands to $\mathrm{O}(d)$, introducing only a mild gap (a reflection ambiguity) that is rigorously resolved via SVD projection. In contrast, as summarized in Table 1, the conventional $4 \times 4$ matrix representation of $\mathrm{SE}(3)$ breaks this favorable correspondence. While the relaxed eigenspace possesses a compact orthogonal geometry, $\mathrm{SE}(3) = \mathrm{SO}(3) \ltimes \mathbb{R}^3$ is inherently non-compact. Consequently, the relaxed solution often lies far from the manifold, creating a large eigenspace-sync gap. Rounding is no longer a benign projection that merely corrects noise; instead, it must artificially enforce a group structure absent from the eigenspace geometry. This necessitates complex, multi-step heuristic procedures that are often unstable (Arrigoni et al., 2016b; Rosen et al., 2019). Although alternative representations exist, such as axis-angle pairs (Thunberg et al., 2014; Jin et al., 2020), augmented unit quaternions (Grisetti et al., 2010; Chen et al., 2025), and dual quaternions (Cheng et al., 2016; Srivatsan et al., 2016; Hadi et al., 2024), they typically lack a unified algebraic framework that is amenable to rigorous optimization guarantees. Concurrent with our work, Zhao et al. (2026) further studied matrix-based spectral estimators, introducing an additional anchoring step to resolve the right-orthogonal ambiguity of the eigenspace.

In this paper, we adopt the dual quaternion representation of $\mathrm{SE}(3)$. Practically, this choice leads to a simpler and more stable rounding/projection step. Conceptually, it aligns the spectral ambiguity more closely with the intrinsic gauge symmetry of $\mathrm{SE}(3)$, thereby mitigating the eigenspace–synchronization mismatch induced by matrix embeddings. Specifically, we represent each pose by a unit dual quaternion and work over the constraint set $\mathrm{UDQ} := \{x \in \mathbb{DH} : |x| = 1\}$. Given noisy relative measurements, we estimate the unknown poses $\hat{\boldsymbol{x}} = (\hat{x}_1, \ldots, \hat{x}_n) \in \mathrm{UDQ}^n$ by solving the least-squares problem

$$\operatorname*{arg\,min}_{x \in \mathrm{UDQ}^n} \sum_{1 \leq i < j \leq n} \left| C_{ij} - x_i x_j^* \right|^2, \qquad (1)$$

where $\mathbb{DH}$ denotes the algebra of dual quaternions and $(\cdot)^*$

is the conjugate transpose, which coincides with the group inverse on UDQ. The measurement matrix $C \in \mathbb{DH}^{n \times n}$ is Hermitian with entries

$$C_{ij} = \hat{x}_i \hat{x}_j^* + \Delta_{ij}, \qquad 1 \leq i < j \leq n, \qquad (2)$$

and $\boldsymbol{\Delta} \in \mathbb{DH}^{n \times n}$ collects the Hermitian observation noise.

To efficiently tackle the nonconvex optimization problem (1), we build on the generalized power method (GPM) and develop a two-stage framework that combines a provably accurate spectral initializer with an iterative refinement scheme that preserves feasibility at every step. Variants of this spectral initializer with an iterative refinement strategy have proved effective for phase, orthogonal, and rotation group synchronization problems (Boumal, 2016; Ling, 2022; Liu et al., 2023; Zhu et al., 2023). However, establishing rigorous guarantees in the dual quaternion setting is substantially more challenging. First, dual quaternions form a ring with zero divisors rather than a field, so division is not generally well defined. Second, as the natural modulus on dual quaternions is merely a dual-number valued seminorm, a bounded modulus does not imply bounded components. This allows translations to grow arbitrarily large despite a finite modulus, thereby posing challenges for stability and convergence analysis. Third, identifying a dominant eigenpair typically invokes the lexicographic order on dual numbers, whereas estimation and convergence are naturally quantified in Euclidean metrics. By addressing these theoretical hurdles, we provide the first recovery guarantees for synchronization over the dual quaternion. Our main contributions are as follows:

- $\mathrm{SE}(3)$ synchronization is formulated as a least-squares problem over unit dual quaternions $\mathrm{UDQ}^n$ with a Hermitian dual quaternion measurement matrix. We formalize a normalization/projection map onto $\mathrm{UDQ}$, establish its stability (a Lipschitz-type bound), and provide a closed-form componentwise projection onto $\mathrm{UDQ}^n$. These results transform the rounding step, often a heuristic in matrix-based methods, into a well-posed and quantitatively controllable operation that underpins our recovery analysis.

- The spectral initialization scheme is developed by com-

puting the dominant eigenvector of a Hermitian dual quaternion measurement matrix via the power method, and then applying an elementwise projection to enforce feasibility in $\mathrm{UDQ}^n$. In contrast to heuristic initializations, we establish nonasymptotic estimation error guarantees for both the raw (generally infeasible) spectral estimator and its projected, feasible counterpart (Proposition 2.4 and Theorem 2.8). These bounds ensure that the initializer lies within a controlled neighborhood of the ground truth under operator norm noise, placing it inside the basin of attraction needed for subsequent GPM refinement and thereby reducing the risk of convergence to undesirable local minima.

- We propose a dual quaternion generalized power method (DQGPM) to refine the spectral initializer while strictly enforcing the unit dual quaternion constraints. Specifically, DQGPM applies an elementwise projection onto $\mathrm{UDQ}^n$ at every iteration, yielding a stop-anytime feasible iterate sequence. Under a mild noise regime and a sufficiently accurate initialization, we prove that the estimation error contracts at a linear rate up to an explicit error floor (Theorem 3.2). To the best of our knowledge, this provides the first finite-iteration recovery guarantee for $\mathrm{SE}(3)$ synchronization, especially for the dual quaternion formulation.

We evaluate our framework on both synthetic datasets and real-world multiple point-set registration tasks. Our results demonstrate that the proposed method significantly outperforms matrix-based approaches, including spectral relaxation and semidefinite relaxation, achieving higher recovery accuracy with greater computational efficiency.

### 1.1. Notation

Throughout the paper, the sets of dual numbers, quaternions, and dual quaternions are denoted by $\mathbb{D}$, $\mathbb{H}$, and $\mathbb{DH}$, respectively. For a dual number $a = a_{\mathrm{st}} + a_{\mathcal{I}}\epsilon$ with $a_{\mathrm{st}} \neq 0$, its inverse is $a^{-1} = a_{\mathrm{st}}^{-1}(1 - a_{\mathcal{I}}a_{\mathrm{st}}^{-1}\epsilon)$. The operator $(\cdot)^*$ denotes the (dual) quaternion conjugate when applied to a scalar element, and the conjugate transpose when applied to a vector or matrix. The scalar part of a quaternion $q$ is denoted by $\mathrm{sc}(q) = \frac{1}{2}(q + q^*)$. For two $n$-dimensional (dual) quaternion vectors $\boldsymbol{x} = (x_1, \ldots, x_n)$ and $\boldsymbol{y} = (y_1, \ldots, y_n)$, their inner product is defined as $\boldsymbol{x}^*\boldsymbol{y} = \sum_{i=1}^{n} x_i^* y_i$. The Frobenius norm of a (dual) quaternion matrix is denoted by $\|\cdot\|_F$, which reduces to the 2-norm $\|\cdot\|_2$ for a (dual) quaternion vector. The operator norm of a (dual) quaternion matrix is denoted by $\|\cdot\|_{\mathrm{op}}$. The set of unit dual quaternions is denoted by $\mathrm{UDQ} := \{q \in \mathbb{DH} : qq^* = 1\}$. We write $\mathrm{UDQ}^n := \{\boldsymbol{x} = (x_1, \ldots, x_n) : x_i \in \mathrm{UDQ}\}$, that is, the set of $n$-dimensional vectors whose entries are unit dual quaternions. For $\boldsymbol{x} \in \mathbb{DH}^n$, the element-wise projection onto $\mathrm{UDQ}^n$ is defined by $\Pi(\boldsymbol{x}) = \operatorname{argmin}_{\boldsymbol{u} \in \mathrm{UDQ}^n} \|\boldsymbol{u} - \boldsymbol{x}\|_2^2$.

For any $\boldsymbol{x}, \boldsymbol{y} \in \mathbb{DH}^n$, the distance between them, defined up to a global unit dual quaternion alignment, is given by $\mathrm{d}(\boldsymbol{x}, \boldsymbol{y}) = \min_{z \in \mathrm{UDQ}} \|\boldsymbol{x} - \boldsymbol{y}z\|_2$. We denote by $\mathrm{d}_{\mathrm{st}}(\boldsymbol{x}, \boldsymbol{y})$ and $\mathrm{d}_{\mathcal{I}}(\boldsymbol{x}, \boldsymbol{y})$ the standard part and the imaginary part of $\mathrm{d}(\boldsymbol{x}, \boldsymbol{y})$, respectively. For $z \in \mathbb{DH}$, we denote the normalization of $z$ onto UDQ as $\mathcal{N}(z)$. Given $\{a_k\} \subset \mathbb{D}$ and $\{c_k\} \subset \mathbb{R}$, we define $a_k = O_{\mathbb{D}}(c_k)$ iff $(a_k)_{\mathrm{st}} = O(c_k)$ and $(a_k)_{\mathcal{I}} = O(c_k)$.

## 2. Spectral Estimator

The least-squares Problem (1) can be reformulated as an equivalent quadratic program with quadratic constraints (QPQC) over unit dual quaternions:

$$\min_{\boldsymbol{x} \in \mathrm{UDQ}^n} \ \|\boldsymbol{C} - \boldsymbol{x}\boldsymbol{x}^*\|_F^2. \tag{3}$$

This reformulation naturally leads to a spectral related problem based on the dominant eigenvectors of the dual quaternion Hermitian matrix $\boldsymbol{C}$.

**Proposition 2.1** (Equivalent QPQC Formulation). *For a Hermitian dual quaternion matrix $\boldsymbol{C} \in \mathbb{DH}^{n \times n}$, Problem (3) is equivalent to*

$$\operatorname*{arg\,max}_{\boldsymbol{x} \in \mathrm{UDQ}^n} \ \boldsymbol{x}^*\boldsymbol{C}\boldsymbol{x}, \tag{P}$$

*and their optimal objective values differ only by a multiplicative factor of 2.*

The maximization problem (P) is nonconvex and can be challenging to solve directly. Nevertheless, by slightly relaxing the constraints, we obtain the following natural spectral relaxation, which reduces the computation to a leading-eigenvectors problem:

$$\operatorname*{arg\,max}_{\boldsymbol{x} \in \mathbb{DH}^n, \ \|\boldsymbol{x}\|_2^2 = n} \ \boldsymbol{x}^*\boldsymbol{C}\boldsymbol{x} \tag{4}$$

*Fact* 2.2 (Ling et al. (2022, Lemma 4.2)). Let $\boldsymbol{C} \in \mathbb{DH}^{n \times n}$ be a Hermitian matrix and $\lambda_1$ be the largest right eigenvalue of $\boldsymbol{C}$ with associated right eigenvector $\boldsymbol{u}_1 \in \mathbb{DH}^n$. Then

$$\lambda_1 = \max\left\{\|\boldsymbol{x}\|^{-2}(\boldsymbol{x}^*\boldsymbol{C}\boldsymbol{x}) \mid \boldsymbol{x} \in \mathbb{DH}^n \setminus \{\boldsymbol{0}\}\right\},$$

and the maximum is attained at $\boldsymbol{x} = \boldsymbol{u}_1$.

By Fact 2.2, the optimum of (4) is attained at the dominant right eigenvector $\boldsymbol{u}_1$ of $\boldsymbol{C}$, normalized so that $\|\boldsymbol{u}_1\|_2^2 = n$. Recently, Cui & Qi (2024) proposed a power iteration method for computing the dominant eigenvalue and a corresponding eigenvector of a dual quaternion Hermitian matrix, and established linear convergence of the resulting iterates to the dominant eigenpair. The procedure is summarized in Algorithm 1.

---

**Algorithm 1** Power Iteration for Eigenvector Estimator

---

1: **Input:** Hermitian matrix $\boldsymbol{C} \in \mathbb{DH}^{n \times n}$ with dominant right eigenvalue $\lambda_1$ satisfying $\lambda_{1,\text{st}} \neq 0$, random initial point $\boldsymbol{w}^0 \in \mathbb{DH}^n$ such that $(\boldsymbol{w}_{\text{st}}^0)^* \boldsymbol{u}_{1,\text{st}} \neq 0$.
2: **for** $k = 1, 2, \ldots$ **do**
3:     $\boldsymbol{y}^k = \boldsymbol{C} \boldsymbol{w}^{k-1}$.
4:     $\boldsymbol{w}^k = \boldsymbol{y}^k \cdot \left( \|\boldsymbol{y}^k\|_2 \right)^{-1}$.
5: **end for**
6: **Output:** $\sqrt{n} \boldsymbol{w}^{k-1}$.

---

*Remark* 2.3 (Well-definedness of the Inverse). The mild assumption $\lambda_{1,\text{st}} \neq 0$ and $(\boldsymbol{w}_{\text{st}}^0)^* \boldsymbol{u}_{1,\text{st}} \neq 0$ ensures that dominant eigenspace exists and the initialization has a nonzero standard part projection onto the dominant eigenspace. If $(\boldsymbol{w}_{\text{st}}^0)^* \boldsymbol{u}_{1,\text{st}} = 0$, then the power iteration cannot amplify (and hence cannot converge to) the dominant eigenspace. Under these assumptions, the standard part iterate remains nonzero, i.e., $\boldsymbol{y}_{\text{st}}^k \neq \boldsymbol{0}$, making the inverse well-defined.

We next characterize the estimation error of this spectral estimator to the ground truth $\hat{\boldsymbol{x}}$.

**Proposition 2.4** (Eigenvector Estimation Error). *Let* $\boldsymbol{x} \in \mathbb{DH}^n$ *satisfy* $\|\boldsymbol{x}\|_2^2 = n$ *and* $\boldsymbol{x}^* \boldsymbol{C} \boldsymbol{x} \geq \hat{\boldsymbol{x}}^* \boldsymbol{C} \hat{\boldsymbol{x}}$. *Then*

$$\mathrm{d}(\boldsymbol{x}, \hat{\boldsymbol{x}}) \leq \frac{4 \|\boldsymbol{\Delta}\|_{\text{op}}}{\sqrt{n}}.$$

Proposition 2.4 shows that the dominant eigenvector $\boldsymbol{u}_1$ of $\boldsymbol{C}$, normalized to satisfy $\|\boldsymbol{u}_1\|_2^2 = n$, is close to the ground truth $\hat{\boldsymbol{x}}$. However, $\boldsymbol{u}_1$ does not necessarily lie in $\text{UDQ}^n$. To obtain a feasible initializer, we therefore apply an element-wise projection of $\boldsymbol{u}_1$ onto $\text{UDQ}^n$.

We first define the normalization operator $\mathcal{N}(\cdot)$, which maps a dual quaternion $x \in \mathbb{DH}$ to a unit dual quaternion $\mathcal{N}(x) \in \text{UDQ}$. Let $u := \mathcal{N}(x)$. If $x_{\text{st}} \neq 0$, then

$$u_{\text{st}} = \frac{x_{\text{st}}}{|x_{\text{st}}|}, \ u_{\mathcal{I}} = \frac{x_{\mathcal{I}}}{|x_{\text{st}}|} - \frac{x_{\text{st}}}{|x_{\text{st}}|} \text{sc} \left( \frac{x_{\text{st}}^*}{|x_{\text{st}}|} \frac{x_{\mathcal{I}}}{|x_{\text{st}}|} \right).$$

If $x_{\text{st}} = 0$ and $x_{\mathcal{I}} \neq 0$, then

$$u_{\text{st}} = \frac{x_{\mathcal{I}}}{|x_{\mathcal{I}}|}, \ u_{\mathcal{I}} \text{ is any quaternion satisfying } \text{sc}(x_{\mathcal{I}}^* u_{\mathcal{I}}) = 0.$$

Next, we record a stability property of the normalization map $\mathcal{N}(\cdot)$. It shows that normalizing a dual quaternion cannot increase its distance to any unit dual quaternion by more than a constant factor. This bound will be used repeatedly to transfer error bounds from the (possibly infeasible) eigenvector $\boldsymbol{u}_1$ to its element-wise projection onto $\text{UDQ}^n$.

**Lemma 2.5** (Lipschitz Property of $\mathcal{N}$). *For any* $y \in \mathbb{DH}$ *and any* $z \in \text{UDQ}$*, we have*

$$|\mathcal{N}(y) - z| \leq 2|y - z|.$$

*Remark* 2.6. Related results were established in Liu et al. (2017) for the special case of $\text{SO}(2)$, where the proof exploits algebraic properties of complex numbers. A more general treatment for the $\text{O}(d)$ group was given in Liu et al. (2023), with arguments based on the definition of projection rather than an explicit normalization map. In contrast, our proof relies only on the algebraic properties of dual quaternions. Nevertheless, normalization and projection coincide in an important sense. In particular, Cui & Qi (2024, Theorem 3.1) shows that for any $x \in \mathbb{DH}$, the normalized dual quaternion $u = \mathcal{N}(x)$ is the metric projection of $x$ onto UDQ, namely

$$\mathcal{N}(x) \in \underset{u \in \text{UDQ}}{\arg \min} |u - x|^2.$$

Consequently, Lemma 2.5 can also be proved using standard properties of metric projections.

The following result shows that the projection of a dual quaternion vector $\boldsymbol{y}$ onto $\text{UDQ}^n$, denoted by $\Pi(\boldsymbol{y})$, admits a closed-form expression.

**Proposition 2.7** (Projection onto $\text{UDQ}^n$). *For any* $\boldsymbol{y} \in \mathbb{DH}^n$*, define*

$$\widetilde{y}_i = \begin{cases} \mathcal{N}(y_i) & \text{if } y_i \neq 0, \\ \mathcal{N}(\boldsymbol{e}^* \boldsymbol{y}) & \text{otherwise}, \end{cases}$$

*where* $\boldsymbol{e} \in \mathbb{DH}^n$ *is any vector satisfying* $\boldsymbol{e}^* \boldsymbol{y} \neq 0$*. If* $\boldsymbol{y} = \boldsymbol{0}$*, we set* $\widetilde{\boldsymbol{y}} = \boldsymbol{1}$*. Then* $\tilde{\boldsymbol{y}}$ *is the projection of* $\boldsymbol{y}$ *onto* $\text{UDQ}^n$*, i.e.,* $\widetilde{\boldsymbol{y}} = \Pi(\boldsymbol{y})$*.*

With the component-wise projection in Proposition 2.7, we can define an alternative feasible estimator

$$\widetilde{\boldsymbol{x}} := \Pi(\boldsymbol{u}_1), \tag{5}$$

which lies in $\text{UDQ}^n$ by construction. We next show that $\widetilde{\boldsymbol{x}}$ remains close to the ground truth signal $\hat{\boldsymbol{x}}$.

**Theorem 2.8** (Rounded Spectral Estimation Error). *For* $\widetilde{\boldsymbol{x}}$ *given by (5), we have*

$$\mathrm{d}(\widetilde{\boldsymbol{x}}, \hat{\boldsymbol{x}}) \leq \frac{8 \|\boldsymbol{\Delta}\|_{\text{op}}}{\sqrt{n}}.$$

Theorem 2.8 follows from the Lipschitz property of the normalization/projection map in Lemma 2.5. In particular, it implies that $\widetilde{\boldsymbol{x}}$ achieves the same order of accuracy as the (generally infeasible) spectral solution $\boldsymbol{u}_1$ to (4), while ensuring feasibility in $\text{UDQ}^n$.

## 3. Iterative Estimation Performance

Although Proposition 2.4 and Theorem 2.8 show that both the (infeasible) spectral solution of (4) and its rounded version yield accurate estimates close to the ground truth signal,

these estimators alone do not directly solve (3) beyond serving as a starting point. To obtain a fully iterative refinement procedure whose iterates remain feasible, we develop a *dual quaternion generalized power method* (DQGPM) as the second-stage algorithm.

The GPM was originally proposed by Journée et al. (2010) for maximizing a convex function over a compact set via a power-type iteration combined with a projection step. Specializing this idea to the synchronization problem (P), our method applies an element-wise projection onto $\mathrm{UDQ}^n$ at every iteration, thereby ensuring feasibility throughout the run. In contrast to eigenvector-based estimators (which only become feasible after a final rounding step), DQGPM can be stopped at any iterate, and the intermediate estimate is always feasible for (P).

---

**Algorithm 2** Dual Quaternion Generalized Power Method

---

1: **Input:** Hermitian matrix $\boldsymbol{C} \in \mathbb{DH}^{n \times n}$, initializer $\boldsymbol{x}^0 = \widetilde{\boldsymbol{x}} = \Pi(\boldsymbol{u}_1) \in \mathrm{UDQ}^n$.
2: **for** $k = 1, 2, \dots$ **do**
3: $\quad \boldsymbol{y}^k = \boldsymbol{C}\,\boldsymbol{x}^{k-1}$.
4: $\quad \boldsymbol{x}^k = \Pi(\boldsymbol{y}^k)$.
5: **end for**
6: **Output:** $\boldsymbol{x}^k$.

---

The following Theorem 3.2 shows that, under mild noise conditions and a sufficiently accurate initializer, the DQGPM iterates contract toward the ground truth at a linear rate up to an $O(\|\boldsymbol{\Delta}\hat{\boldsymbol{x}}\|_2/n)$ error floor.

*Remark* 3.1 (Algebraic Order vs. Estimation Metric). We need to distinguish the algebraic order used in Section 2 from the error metric used in our convergence analysis. While the spectral estimator relies on the lexicographical order of dual numbers to define the dominant eigenpair (strictly prioritizing the standard part), SE(3) synchronization requires bounded errors in both rotation and translation. Therefore, in this section, we analyze the estimation error by controlling the real-valued magnitudes of the standard component $\mathrm{d}_{\mathrm{st}}$ and the dual component $\mathrm{d}_{\mathcal{I}}$ individually. This component-wise analysis is consistent with the algebraic structure of dual numbers, where the convergence of the dual part ($\mathrm{d}_{\mathcal{I}}$) typically depends on the accuracy of the standard part ($\mathrm{d}_{\mathrm{st}}$).

**Theorem 3.2** (Estimation error of DQGPM). *Suppose that the measurement noise satisfies* $\|\boldsymbol{\Delta}\|_{\mathrm{op,st}} \leq n/350$, $\|\boldsymbol{\Delta}\|_{\mathrm{op},\mathcal{I}} \leq n/300$, *and the initialization is given by* $\boldsymbol{x}^0 = \widetilde{\boldsymbol{x}}$. *Then the sequence generated by Algorithm 2 satisfies*

$$\mathrm{d}_{\mathrm{st}}(\boldsymbol{x}^k, \hat{\boldsymbol{x}}) \leq \left(\frac{1}{10}\right)^k \frac{\sqrt{n}}{25} + \frac{700}{53n}\|(\boldsymbol{\Delta}\hat{\boldsymbol{x}})_{\mathrm{st}}\|_2. \quad (6)$$

*Moreover, assume that there exists $B \geq 0$ such that*

$$\max_{k \geq 0} \max_{1 \leq i \leq n} |(r_i^k)_{\mathcal{I}}| \leq B, \quad (7)$$

*where* $\boldsymbol{r}^k = \hat{\boldsymbol{x}} z_k$ *with* $z_k \in \mathrm{argmin}_{z \in \mathrm{UDQ}} \|\boldsymbol{x}^k - \hat{\boldsymbol{x}} z\|_2$, *the projection inputs are uniformly nondegenerate:*

$$\min_{k \geq 0} \min_{1 \leq i \leq n} \left|\left(\frac{1}{n}Cx^k\right)_{i,\mathrm{st}}\right| \geq \frac{1}{2}, \quad (8)$$

*and there exists a finite constant $\gamma \geq 0$ such that*

$$\delta_{\mathcal{I}}^{k+1} \leq \left\|(\boldsymbol{x}^{k+1} - \hat{\boldsymbol{x}}z_k)_{\mathcal{I}}\right\|_2 + \gamma \left\|(\boldsymbol{x}^{k+1} - \hat{\boldsymbol{x}}z_k)_{\mathrm{st}}\right\|_2, \quad (9)$$

*where* $\delta_{\mathcal{I}}^{k+1} := \|(\boldsymbol{x}^{k+1} - \hat{\boldsymbol{x}}z_{k+1})_{\mathcal{I}}\|_2$. *Then for all $k \geq 0$,*

$$\left|\mathrm{d}_{\mathcal{I}}(\boldsymbol{x}^k, \hat{\boldsymbol{x}})\right| \leq \delta_{\mathcal{I}}^k \leq \left(\frac{1}{10}\right)^k \delta_{\mathcal{I}}^0$$
$$+ \frac{186B + 48\gamma + 7}{1050}\left[\frac{\sqrt{n}}{25}k\left(\frac{1}{10}\right)^{k-1} + \frac{7000}{477n}\|(\boldsymbol{\Delta}\hat{\boldsymbol{x}})_{\mathrm{st}}\|_2\right]$$
$$+ \frac{10}{9}\left[\frac{7 + 6B}{1050}\sqrt{n} + \frac{6B + 2\gamma}{n}\|(\boldsymbol{\Delta}\hat{\boldsymbol{x}})_{\mathrm{st}}\|_2\right]. \quad (10)$$

*Remark* 3.3. It is worth noting the improvement in estimation accuracy provided by DQGPM over the rounded spectral estimator. Theorem 3.2 establishes that, asymptotically as $k \to \infty$ (assuming convergent iterates and a small noise constant regime where $\|\boldsymbol{\Delta}\|_{\mathrm{op,st}} \leq an$ for sufficiently small $a > 0$), the DQGPM estimation error satisfies:

$$\mathrm{d}_{\mathrm{st}}(\boldsymbol{x}^\infty, \hat{\boldsymbol{x}}) \lesssim \frac{12}{n}\|\boldsymbol{\Delta}\hat{\boldsymbol{x}}\|_{2,\mathrm{st}},$$
$$|\mathrm{d}_{\mathcal{I}}(\boldsymbol{x}^\infty, \hat{\boldsymbol{x}})| \leq \delta_{\mathcal{I}}^\infty \lesssim \frac{396B + 124\gamma}{45n}\|\boldsymbol{\Delta}\hat{\boldsymbol{x}}\|_{2,\mathrm{st}}.$$

In contrast, Theorem 2.8 guarantees an error bound for the rounded spectral estimator of order $8\|\boldsymbol{\Delta}\|_{\mathrm{op}}/\sqrt{n}$. Comparing these rates, DQGPM yields a tighter bound provided that $\|\boldsymbol{\Delta}\hat{\boldsymbol{x}}\|_{2,\mathrm{st}} \leq c\sqrt{n}\|\boldsymbol{\Delta}\|_{\mathrm{op,st}}$ for a constant $c < 2/3$. Note that the trivial bound $c = 1$ holds automatically by definition. In many stochastic noise models, however, $\hat{\boldsymbol{x}}$ is not aligned with the top singular directions of $\boldsymbol{\Delta}$, so $\|\boldsymbol{\Delta}\hat{\boldsymbol{x}}\|_2$ is typically much smaller than the worst-case bound (e.g., $c \approx 1/2$ for standard i.i.d. Gaussian noise), which highlights the advantage of the DQGPM refinement.

Finally, since the dominant eigenvector $\boldsymbol{u}_1$ cannot be computed exactly in practice, we quantify the following finite-iteration estimation error of the two-stage procedure power iteration for initialization followed by DQGPM refinement.

**Corollary 3.4** (Finite-Iteration Estimation Error). *Suppose the assumptions in Theorem 3.2 hold. Let* $\boldsymbol{C} \in \mathbb{DH}^{n \times n}$ *be Hermitian and admits a* unique *dominant eigenvalue with multiplicity one. Denote its right eigenvalues by* $\lambda_1 > \lambda_2 \geq$

*Table 2.* Rotation and translation errors in synthetic experiments of DQGPM, SPEC, and EIG across noise levels and observation rates.

| | NOISE LEVEL | | $(0.05, 5°)$ | $(0.10, 10°)$ | $(0.15, 15°)$ | $(0.20, 20°)$ |
|---|---|---|---|---|---|---|
| | | DQGPM | $\mathbf{0.132 \pm 0.042}$ | $\mathbf{0.230 \pm 0.048}$ | $\mathbf{0.326 \pm 0.051}$ | $\mathbf{0.424 \pm 0.060}$ |
| | ERROR_R | SPEC | $1.639 \pm 1.971$ | $1.889 \pm 2.022$ | $1.854 \pm 1.862$ | $2.035 \pm 1.823$ |
| $p = 0.05$ | | EIG | $0.174 \pm 0.156$ | $0.270 \pm 0.099$ | $0.406 \pm 0.190$ | $0.585 \pm 0.572$ |
| | | DQGPM | $\mathbf{0.102 \pm 0.032}$ | $\mathbf{0.190 \pm 0.046}$ | $\mathbf{0.279 \pm 0.061}$ | $\mathbf{0.369 \pm 0.078}$ |
| | ERROR_T | SPEC | $0.480 \pm 0.530$ | $0.532 \pm 0.504$ | $0.590 \pm 0.472$ | $0.660 \pm 0.455$ |
| | | EIG | $0.551 \pm 1.109$ | $1.609 \pm 6.676$ | $0.899 \pm 1.478$ | $1.043 \pm 1.359$ |
| | | DQGPM | $\mathbf{0.027 \pm 0.001}$ | $\mathbf{0.054 \pm 0.002}$ | $\mathbf{0.082 \pm 0.004}$ | $\mathbf{0.111 \pm 0.005}$ |
| | ERROR_R | SPEC | $0.032 \pm 0.013$ | $0.192 \pm 0.867$ | $0.094 \pm 0.037$ | $0.141 \pm 0.126$ |
| $p = 0.30$ | | EIG | $0.098 \pm 0.618$ | $0.134 \pm 0.614$ | $0.106 \pm 0.021$ | $0.219 \pm 0.613$ |
| | | DQGPM | $\mathbf{0.021 \pm 0.001}$ | $\mathbf{0.042 \pm 0.002}$ | $\mathbf{0.063 \pm 0.003}$ | $\mathbf{0.085 \pm 0.005}$ |
| | ERROR_T | SPEC | $0.023 \pm 0.001$ | $0.045 \pm 0.002$ | $0.067 \pm 0.004$ | $0.090 \pm 0.005$ |
| | | EIG | $0.137 \pm 0.441$ | $0.415 \pm 2.123$ | $0.249 \pm 0.677$ | $0.194 \pm 0.257$ |

$^*$ SDR excluded due to limited scalability under uniform edge dropout.

$\cdots \geq \lambda_n$, *where each* $\lambda_i = \lambda_{i,\mathrm{st}} + \lambda_{i,\mathcal{I}}\epsilon$ *is a dual number. If we run Algorithm 1 for* $K_{\mathrm{init}}$ *iterations satisfying*

$$K_{\mathrm{init}} \geq \log_r \left( \frac{70|\alpha_{2,\mathrm{st}}| + 69M_{\mathrm{st}}}{|\alpha_{1,\mathrm{st}}|} \right),$$

*then, initializing Algorithm 2 with* $\boldsymbol{x}^0 := \Pi(\sqrt{n}\boldsymbol{w}^{K_{\mathrm{init}}})$, *the resulting sequence* $\{\boldsymbol{x}^k\}_{k \geq 0}$ *satisfies* (6) *and* (10). *Here* $r := \left|\lambda_{1,\mathrm{st}}/\lambda_{2,\mathrm{st}}\right| > 1$, $M_{\mathrm{st}}$ *is a real constant depending on* $\{\alpha_j, \lambda_j\}_{j=1}^n$, *where* $\alpha_i$ *is the projection of initializer* $\boldsymbol{w}^0$ *onto the* $i$-*th orthonormal eigenvector.*

## 4. Numerical Experiments

In this section, we report the recovery performance and numerical efficiency of our proposed DQGPM for SE(3) synchronization tasks on both synthetic and real data, where the real data experiments are implemented on multiple point-set registration problem. We also compare our method with three existing matrix-based methods, which are the spectral decomposition (EIG) described in Arrigoni et al. (2016b), the semidefinite relaxation approach (SDR) introduced in Rosen et al. (2019) and the recent spectral relaxation method (SPEC) proposed in Doherty et al. (2022). All experiments were performed in Python 3.9 on a computer running Windows 11, equipped with an AMD Ryzen 7 6800HS Creator Edition (3.20 GHz) processor and 16 GB of RAM. Experimental results on synthetic data with additive noise are presented in Section 4.2, while Section 4.3 reports results obtained from real data in the multiple point-set registration experiment. The implementation code is available at https://github.com/jnzhao333/dq_sync.

### 4.1. Data Generation and Evaluation Metrics

**Data Generation**   We construct synthetic experiments by simulating a noisy measurement matrix $\boldsymbol{C}$ under the additive model

$$\boldsymbol{C} = \hat{\boldsymbol{x}}\hat{\boldsymbol{x}}^* + \boldsymbol{\Xi} - \mathbf{1}_{n \times n}.$$

Here, $\hat{\boldsymbol{x}} = (\hat{x}_1, \ldots, \hat{x}_n) \in \mathrm{UDQ}^n$ is the ground truth pose vector, and term $\boldsymbol{\Xi} = (\xi_{ij}) \in \mathrm{UDQ}^{n \times n}$ is a Hermitian dual quaternion noise matrix, all their entries are sampled i.i.d. as unit dual quaternions. The matrix $\mathbf{1}_{n \times n}$ denotes the all-one dual quaternion matrix. The term $\boldsymbol{\Xi} - \mathbf{1}_{n \times n}$ corresponds to the additive noise $\boldsymbol{\Delta}$ in the model (2).

To evaluate robustness under partial observation, we sparsify the measurements using an Erdős–Rényi (ER) graph $\boldsymbol{E} = (e_{ij})$, where $e_{ij} \sim \mathrm{Bernoulli}(p)$ and $p$ controls the observation rate. The resulting simulated measurement matrix $\boldsymbol{C}$ is defined entrywise as

$$C_{ij} = \begin{cases} e_{ij}\big(\hat{x}_i\hat{x}_j^* + \xi_{ij} - 1\big), & 1 \leq i < j \leq n, \\ 1, & i = j, \\ C_{ji}^*, & 1 \leq j < i \leq n. \end{cases}$$

We subtract $\mathbf{1}_{n \times n}$ for two reasons. First, since $\xi_{ij}$ is a unit dual quaternion, adding it directly to $\hat{x}_i\hat{x}_j^*$ can inflate the entry magnitude and distort the signal. Second, in dual quaternions the motion-free baseline is the identity element (an all-ones matrix), not zero. Therefore, we use the shifted model and subtract $\mathbf{1}_{n \times n}$.

To represent elements in SE(3) for both the ground truth poses and the noise terms, we sample $(\boldsymbol{u}, \theta, \boldsymbol{t})$ to generate a unit dual quaternion $x$. Each sampled dual quaternion $x$ is constructed from a random rotation represented by $q \in \mathbb{H}$ and translation $\boldsymbol{t} \in \mathbb{R}^3$: we first form the unit quaternion $q$ from a randomly sampled axis $\boldsymbol{u} \in \mathbb{R}^3$ and angle $\theta \in [0, 2\pi)$ via (11), and generate a randomly sampled $\boldsymbol{t}$ then convert $(q, \boldsymbol{t})$ to $x$ using (12).

**Evaluation Metrics**   We follow the evaluation and gauge-alignment procedure of Hadi et al. (2024). Define ground

*Table 3.* Cross-sectional views of the aligned point clouds on three real datasets for different methods.

truth poses as $\hat{\boldsymbol{x}} = \{(\hat{q}_i, \hat{t}_i)\}_{i=1}^n \in \mathrm{SE}(3)^n$ and estimates as $\boldsymbol{x} = \{(q_i, t_i)\}_{i=1}^n \in \mathrm{SE}(3)^n$, where $\hat{q}_i, q_i \in \mathbb{H}$ are unit quaternions and $\hat{t}_i, t_i \in \mathbb{R}^3$ are translations. We compute the estimation error as the distance between $\boldsymbol{x}$ and the right-aligned ground truth $\hat{\boldsymbol{x}}z$, where $z$ is the optimal aligner that minimizes the distance between $\boldsymbol{x}$ and $\hat{\boldsymbol{x}}z$. Specifically, we get the aligning SE(3) element $z = (q, t)$ by

$$q = \frac{s}{\|s\|}, \quad t = \frac{1}{n}\sum_{j=1}^n \psi_{\hat{q}_j^*}(t_j - \hat{t}_j),$$

where $s = \sum_{j=1}^n \hat{q}_j^* q_j$, and $\psi : \mathbb{R}^3 \to \mathbb{R}^3$ denotes the action of the quaternion-induced rotation on vectors (see Proposition B.1 in the Appendix).

After alignment, we report rotation and translation errors for each estimated pose, following Hadi et al. (2024):

$$d_R(q_1, q_2) = 2\arccos\bigl(2\langle q_1, q_2 \rangle^2 - 1\bigr),$$
$$d_T(t_1, t_2) = \|t_1 - t_2\|_2,$$

where $\langle \cdot, \cdot \rangle$ denotes the Euclidean inner product in $\mathbb{R}^4$. We report error_r and error_t as the mean of $d_R$ and $d_T$, respectively, over all $n$ targets.

### 4.2. Recovery Performance and Computational Time

For each ground truth pose $\hat{x}_i$, we sample the rotation axis $\boldsymbol{u}$ uniformly from the unit sphere in $\mathbb{R}^3$ and the rotation angle independently and uniformly from $[0, 2\pi)$. The translation $\boldsymbol{t} = (t_1, t_2, t_3) \in \mathbb{R}^3$ has i.i.d. entries following $\mathcal{N}(0, 1)$. To generate the noise matrix, we again sample rotation axes uniformly from the unit sphere, while rotation angles (in degrees) are drawn i.i.d. from $\mathcal{N}(0, \sigma_r^2)$ with $\sigma_r \in \{5°, 10°, 15°, 20°\}$. The corresponding translation noise has i.i.d. entries drawn from $\mathcal{N}(0, \sigma_t^2)$ with $\sigma_t \in \{0.05, 0.10, 0.15, 0.2\}$, respectively.

We run synthetic experiments with problem size $n = 100$ under three observation rates $p \in \{0.05, 0.3\}$, corresponding

to approximately 95% and 70% missing pairs, respectively. For each setting, we execute Algorithm 1 until the residual drops below $10^{-5}$, and then project the output elementwise onto $\mathrm{UDQ}^n$. All results are averaged over 100 independent trials.

Table 2 reports the rotation and translation errors (error_r, error_t) of DQGPM and the baselines SPEC and EIG across various noise levels and observation rates. The two metrics exhibit consistent trends, and DQGPM is uniformly more accurate and stable, with the largest gains in sparse graphs ($p = 0.05$). DQGPM maintains small errors across all noise levels, whereas EIG incurs larger errors with substantially higher variance. As the graph becomes denser ($p = 0.30$), both methods improve and the performance gap narrows; nonetheless, DQGPM continues to achieve lower errors and reduced variance, retaining a mild but consistent advantage in the dense regime. Besides, we record the total CPU time consumed by each method in Table 5. It can be observed that DQGPM is consistently faster than EIG across all settings, and its runtime is insensitive to the noise level.

Figure 1 plots estimation error (error_r and error_t) versus iteration for different problem sizes $n$, indicating a linear error contraction of estimation error for GPM refinement stage. Within a small number of iterations, the decrease gradually saturates and the errors flatten at a nonzero plateau, which corresponds to the asymptotic noise floor as analyzed in Remark 3.3.

### 4.3. Real Data: Stanford Bunny and 3DMatch

We evaluate our method on real data from the Stanford 3D Scanning Repository[1] and 3DMatch[2] dataset, and compare it against EIG, SDR and SPEC methods. To construct the

---

[1]http://graphics.stanford.edu/data/
3Dscanrep
[2]https://3dmatch.cs.princeton.edu/
#geometric-registration-benchmark

*Table 4.* Performance on four real datasets under $(\sigma_t, \sigma_r) = (0.01, 10°)$: missing rate, time, and errors.

| | | BUNNY | BUDDHA | | DRAGON | | ARMADILLO | |
|---|---|---|---|---|---|---|---|---|
| | | SPARSE | DENSE | SPARSE | DENSE | SPARSE | DENSE | SPARSE |
| MISSING | | 48.00% | 18.67% | 66.67% | 19.44% | 60.44% | 19.44% | 58.33% |
| TIME | DQGPM | 0.0010 | **0.0003** | 0.0021 | **0.0003** | 0.0014 | **0.0003** | 0.0009 |
| | SPEC | 0.0018 | 0.0043 | 0.0041 | 0.0043 | 0.0042 | 0.0028 | 0.0028 |
| | SDR | 0.0954 | 0.2166 | 0.1893 | 0.2306 | 0.1964 | 0.2410 | 0.1907 |
| | EIG | **0.0006** | 0.0008 | **0.0007** | 0.0009 | **0.0007** | 0.0007 | **0.0006** |
| ERROR_R | DQGPM | **0.1631** | **0.0898** | **0.2118** | **0.0935** | **0.1648** | **0.1095** | **0.1761** |
| | SPEC | 3.4489 | 0.1826 | 3.1018 | 0.2404 | 3.1359 | 0.5723 | 3.1249 |
| | SDR | 3.6349 | 0.1937 | 3.1181 | 0.2843 | 3.1320 | 0.5757 | 3.1236 |
| | EIG | 3.4460 | 0.1736 | 3.0993 | 0.2623 | 3.1349 | 0.5785 | 3.1260 |
| ERROR_T | DQGPM | **0.0107** | 0.0044 | 0.0117 | 0.0047 | 0.0085 | 0.0056 | 0.0088 |
| | SPEC | 0.0656 | **0.0038** | **0.0021** | **0.0038** | **0.0024** | **0.0040** | **0.0026** |
| | SDR | 0.0675 | 0.0042 | 0.0021 | 0.0042 | 0.0024 | 0.0043 | 0.0027 |
| | EIG | 0.0509 | 0.0039 | 0.0021 | 0.0039 | 0.0024 | 0.0042 | 0.0026 |

*Table 5.* CPU time of DQGPM, SPEC and EIG across noise levels and observation rates.

| | | $(0.05, 5°)$ | $(0.10, 10°)$ | $(0.15, 15°)$ | $(0.20, 20°)$ |
|---|---|---|---|---|---|
| | DQGPM | **0.005** | **0.005** | **0.005** | **0.004** |
| $p = 0.05$ | SPEC | 0.084 | 0.089 | 0.069 | 0.069 |
| | EIG | 0.006 | 0.005 | 0.005 | 0.004 |
| | DQGPM | **0.002** | **0.001** | **0.001** | **0.001** |
| $p = 0.30$ | SPEC | 0.143 | 0.140 | 0.140 | 0.142 |
| | EIG | 0.003 | 0.004 | 0.004 | 0.004 |

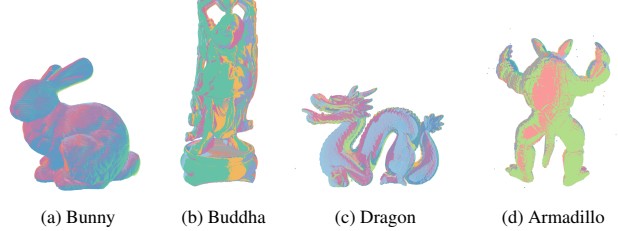

(a) Bunny    (b) Buddha    (c) Dragon    (d) Armadillo

*Figure 2.* 3D reconstructions obtained by DQGPM on four real datasets. Colors indicate individual point clouds.

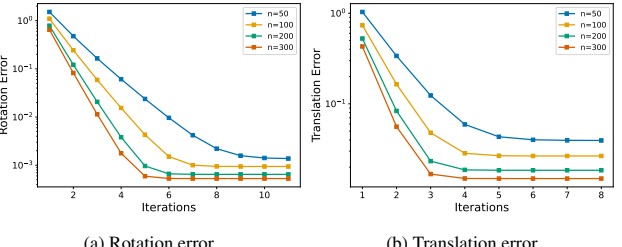

(a) Rotation error    (b) Translation error

*Figure 1.* Error decay over iterations: (a) rotation error, (b) translation error, with $(\sigma_t, \sigma_r) = (0.1, 10°)$ and $p = 0.3$.

the shapes after applying DQGPM to align and merge the corresponding point clouds. To evaluate robustness under sparse observations, we construct two types of ICP-based measurement matrices for the last three datasets. The sparse version is obtained by running ICP without any initialization, whereas the dense version uses an initialization in which the relative orientation is inferred from the point-cloud file names. The bunny dataset only admits the sparse construction, since its file names do not encode relative orientations and therefore do not provide a meaningful ICP initializer.

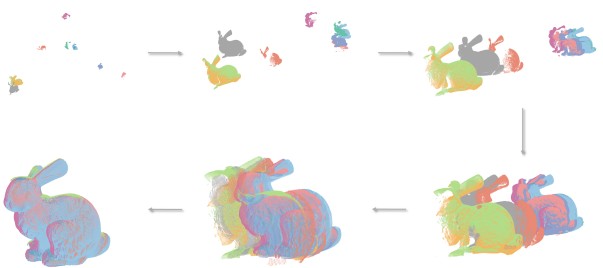

*Figure 3.* Point-cloud alignment over DQGPM iterations.

measurement matrix, we first use Iterative Closest Point (ICP) algorithm (Besl & McKay, 1992) to estimate relative rigid motions, on which we apply additional rigid-motion perturbation. Specifically, the rotation angle (in degrees) is sampled from $\mathcal{N}(0, 10^2)$, the rotation axis is drawn uniformly from the unit sphere, and the translation vector (in meters) has i.i.d. entries following $\mathcal{N}(0, 0.01^2)$, consistent with the scale of translations in the datasets. All results are averaged over 100 independent trials.

Four datasets are considered in our real-data experiments: bunny, buddha, dragon, and armadillo. Figure 2 visualizes

*Table 6.* Performance on the 3DMatch dataset.

|  |  | OFFICE1 | OFFICE2 | LIVINGROOM1 | LIVINGROOM2 | REDKITCHEN | HOTEL3 |
|---|---|---|---|---|---|---|---|
| NO. NODES |  | 53 | 50 | 57 | 47 | 60 | 37 |
| MISSING |  | 81.67% | 83.36% | 78.12% | 79.86% | 70.22% | 89.41% |
| ERROR_R | DQGPM | **0.1209** | **0.0066** | **7.78E-7** | **9.00E-6** | **2.89E-5** | **3.96E-7** |
|  | SPEC | 1.5195 | 1.7705 | 1.3342 | 2.0904 | 1.0184 | 1.2469 |
|  | SDR | 1.5038 | 1.7616 | 1.3195 | 2.0946 | 1.0082 | 1.2460 |
|  | EIG | 1.5093 | 1.7661 | 1.3238 | 2.0951 | 1.0128 | 1.2469 |
| ERROR_T | DQGPM | **0.2454** | **0.0155** | **3.12E-6** | **1.06E-5** | **4.71E-5** | **3.87E-7** |
|  | SPEC | 1.8254 | 1.5662 | 1.7862 | 1.6364 | 0.8466 | 0.3994 |
|  | SDR | 2.4180 | 2.0358 | 2.2108 | 1.7699 | 1.1597 | 0.4336 |
|  | EIG | 1.8860 | 1.6429 | 1.7988 | 1.8343 | 0.8658 | 0.4029 |

Table 3 visualizes cross-sectional slices of the registered point clouds produced by different methods. Better alignment manifests as sharper, cleaner contours with fewer ghosting artifacts and outliers. Figure 3 shows the evolution of the alignment over DQGPM iterations, where the mis-registration decreases monotonically and the point clouds gradually collapse to a consistent configuration. The results for estimation error and CPU time are summarized in Table 4. Overall, DQGPM attains comparable accuracy with substantially lower runtime than SDR, while maintaining strong performance even when a large fraction of measurements is missing.

In addition to the Stanford object datasets, we evaluate our method on real indoor scenes from the 3DMatch dataset. Different from the ICP-based construction above, here we directly use the sparse pairwise rigid motions provided in `gt.log` to build the measurement matrix $C$, and no additional perturbation is added. For each scene, let $G = (V, E)$ denote the graph of point-cloud fragments, where each observed edge $(i, j) \in E$ is associated with a reference relative transformation $\widehat{X}_{ij} \in \mathrm{SE}(3)$. Given the estimated absolute poses $\{X_i\}_{i=1}^n$, we reconstruct the relative motion on each observed edge as $X_{ij} = X_i X_j^{-1}$, or equivalently $x_{ij} = x_i x_j^*$ in the dual quaternion representation. Since ground truth is available only on the observed edges, we evaluate directly in the edge space and report

$$\mathrm{error}_r = \frac{1}{|E|} \sum_{(i,j) \in E} \mathrm{d}_R(q_{ij}, \widehat{q}_{ij}),$$

$$\mathrm{error}_t = \frac{1}{|E|} \sum_{(i,j) \in E} \mathrm{d}_T(t_{ij}, \widehat{t}_{ij}),$$

where $(q_{ij}, t_{ij})$ and $(\widehat{q}_{ij}, \widehat{t}_{ij})$ are extracted from $X_{ij}$ and $\widehat{X}_{ij}$, respectively, and $\mathrm{d}_R, \mathrm{d}_T$ are defined as in Section 4.1. The results are summarized in Table 6. The 3D reconstructions in Figure 4 and 5 show that DQGPM produces more accurate and visually consistent alignments than SPEC, SDR, and EIG under highly sparse observations.

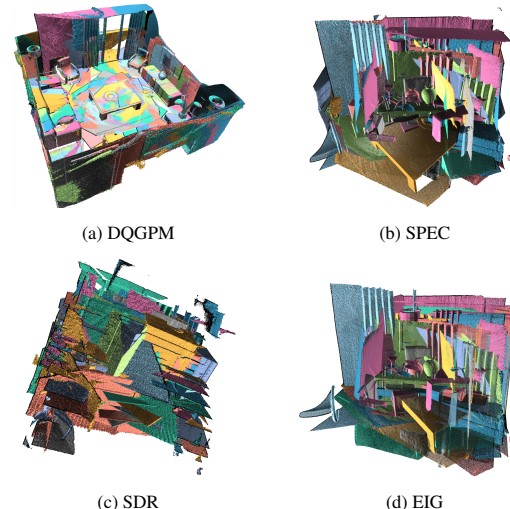

(a) DQGPM      (b) SPEC

(c) SDR      (d) EIG

*Figure 4.* 3D reconstructions of 3DMatch Livingroom2 dataset ($n = 47$ and observation missing rate 79.86%).

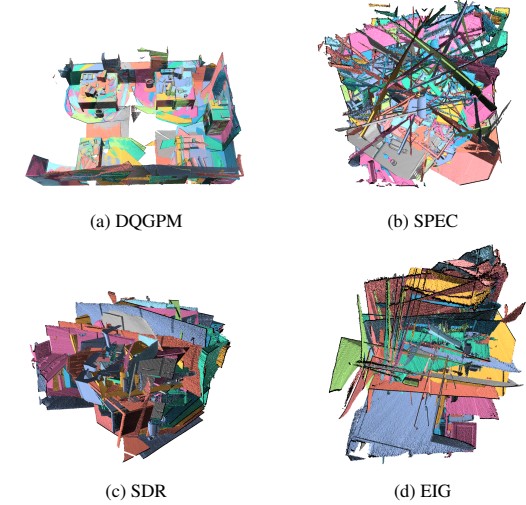

(a) DQGPM      (b) SPEC

(c) SDR      (d) EIG

*Figure 5.* 3D reconstructions of 3DMatch Office2 dataset ($n = 50$ and observation missing rate 83.36%).

## Acknowledgements

The authors thank Professor Chunfeng Cui and Mr. Hao Yang for their careful reading of an earlier version of the manuscript and for pointing out missing details in the proof of Theorem 3.2. This research is supported in part by the Hong Kong Research Grants Council (RGC) General Research Fund (GRF) project CUHK 14206525.

## Impact Statement

This paper presents work whose goal is to advance the field of Machine Learning. There are many potential societal consequences of our work, none which we feel must be specifically highlighted here.

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

# A. Algebras

## A.1. Dual Numbers

We denote the set of dual numbers as $\mathbb{D}$. A dual number $d$ has the form of $d = d_{\text{st}} + d_{\mathcal{I}}\epsilon$ with $d_{\text{st}}$ and $d_{\mathcal{I}}$ in real number field. We call $d_{\text{st}}$ the real part or standard part, and $d_{\mathcal{I}}$ is regarded as the dual part or infinitesimal part. $\epsilon$ is a dual unit satisfying $\epsilon^2 = 0$, which is commutative in multiplication with real numbers, complex numbers, and quaternion numbers. A dual number is called infinitesimal if it only has dual part, otherwise, we call it appreciable.

Addition and multiplication on dual number can be defined as follows: for $d_i \in \mathbb{D}$, where $d_i = d_{i,\text{st}} + d_{i,\mathcal{I}}\epsilon$, $i = 1, 2$, we have

$$d_1 + d_2 = (d_{1,\text{st}} + d_{2,\text{st}}) + (d_{1,\mathcal{I}} + d_{2,\mathcal{I}})\epsilon$$
$$d_1 d_2 = d_{1,\text{st}}d_{2,\text{st}} + (d_{1,\text{st}}d_{2,\mathcal{I}} + d_{1,\mathcal{I}}d_{2,\text{st}})\epsilon$$

Although the above properties show the similarity between dual numbers and complex number field, we should notice that the algebra of dual numbers form a ring but not a field as infinitesimal dual number does not have an inverse element. For the dual number ring, $d = d_{\text{st}} + d_{\mathcal{I}}\epsilon \in \mathbb{D}$ with $d_{\text{st}} \neq 0$ has an inverse of

$$d^{-1} = d_{\text{st}}^{-1}(1 - d_{\mathcal{I}}d_{\text{st}}^{-1}\epsilon),$$

The zero element of $\mathbb{D}$ is $0_{\mathbb{D}} := 0 + 0\epsilon$ and the identity element is $1_{\mathbb{D}} := 1 + 0\epsilon$. Moreover, Qi et al. (2022) defined the square root and absolute value of $d$ as

$$\sqrt{d} = \sqrt{d_{\text{st}}} + \frac{d_{\mathcal{I}}}{2\sqrt{d_{\text{st}}}}\epsilon$$

and

$$|d| = \begin{cases} |d_{\text{st}}| + \text{sgn}(d_{\text{st}})d_{\mathcal{I}}\epsilon, & \text{if } d_{\text{st}} \neq 0, \\ |d_{\mathcal{I}}|\epsilon, & \text{otherwise.} \end{cases}$$

where $\text{sgn}(u) = \frac{u}{|u|}$ for nonzero real number $u$ and $\text{sgn}(u) = 0$ otherwise. For dual numbers $x = a + b\epsilon$ and $y = c + d\epsilon$, a total order $x > y$ is given in Qi et al. (2022) by $a > c$ or $a = c$ and $b > d$, and $x = y$ if and only if $a = c$ and $b = d$.

## A.2. Quaternions

We denote the set of quaternions by $\mathbb{H}$. Let $\boldsymbol{i}, \boldsymbol{j}, \boldsymbol{k}$ be an orthonormal basis of $\mathbb{R}^3$. A quaternion $q \in \mathbb{H}$ is represented as the sum of a scalar part $q_0$ and a vector part $\boldsymbol{q} = (q_1, q_2, q_3)$, namely,

$$q = q_0 + \boldsymbol{q} = q_0 + q_1\boldsymbol{i} + q_2\boldsymbol{j} + q_3\boldsymbol{k}.$$

Equivalently, $q$ can be written as the four-dimensional vector $(q_0, q_1, q_2, q_3)$. The real part of $q$ is $\text{Re}(q) = q_0$, and the imaginary part is $\text{Im}(q) = q_1\boldsymbol{i} + q_2\boldsymbol{j} + q_3\boldsymbol{k}$.

Motivated by the dot product and cross product in $\mathbb{R}^3$, the product of two quaternions $p = p_0 + \boldsymbol{p}$ and $q = q_0 + \boldsymbol{q}$ is given by

$$pq = p_0q_0 - \boldsymbol{p} \cdot \boldsymbol{q} + p_0\boldsymbol{q} + q_0\boldsymbol{p} + \boldsymbol{p} \times \boldsymbol{q},$$

where $\boldsymbol{pq}$ denotes the dot product, i.e., the inner product of $\boldsymbol{p}$ and $\boldsymbol{q}$. Quaternion multiplication satisfies the distributive law but is generally noncommutative. Consequently, the quaternion algebra forms a noncommutative division ring.

The conjugate of a quaternion $q = q_0 + q_1\boldsymbol{i} + q_2\boldsymbol{j} + q_3\boldsymbol{k}$ is defined as $q^* := q_0 - q_1\boldsymbol{i} - q_2\boldsymbol{j} - q_3\boldsymbol{k}$. Its magnitude is $|q| = \sqrt{q_0^2 + q_1^2 + q_2^2 + q_3^2}$. Consequently, the inverse of any nonzero quaternion $q$ is given by $q^{-1} = q^*/|q|^2$. Moreover, for any two quaternions $p$ and $q$, the conjugation reverses the order of multiplication, i.e., $(pq)^* = q^*p^*$. A quaternion $q$ with $|q| = 1$ is called a unit quaternion (or a rotation quaternion).

We denote by $\mathbb{H}^n$ the set of $n$-dimensional quaternion vectors. For $\boldsymbol{x} = (x_1, x_2, \ldots, x_n)^\top$ and $\boldsymbol{y} = (y_1, y_2, \ldots, y_n)^\top \in \mathbb{H}^n$, we define

$$\boldsymbol{x}^*\boldsymbol{y} = \sum_{i=1}^{n} x_i^* y_i,$$

where $\boldsymbol{x}^* = (x_1^*, x_2^*, \ldots, x_n^*)$ denotes the conjugate transpose of $\boldsymbol{x}$.

## A.3. Dual Quaternion

We denote the set of dual quaternions by $\mathbb{DH}$. Any dual quaternion $q \in \mathbb{DH}$ can be written as

$$q = q_{\mathrm{st}} + q_{\mathcal{I}}\epsilon,$$

where $q_{\mathrm{st}}, q_{\mathcal{I}} \in \mathbb{H}$ are referred to as the standard part and the dual part of $q$, respectively. When $q_{\mathrm{st}} \neq 0$, the dual quaternion $q$ is called appreciable. Moreover, if both $q_{\mathrm{st}}$ and $q_{\mathcal{I}}$ are imaginary quaternions, then $q$ is called an imaginary dual quaternion.

The conjugate of $q$ is defined by

$$q^* = q_{\mathrm{st}}^* + q_{\mathcal{I}}^*\epsilon.$$

If $q = q^*$, then $q$ is a dual number. If $q$ is imaginary, then $q^* = -q$.

The magnitude of $q$ is defined as

$$|q| := \begin{cases} |q_{\mathrm{st}}| + \dfrac{(q_{\mathrm{st}}q_{\mathcal{I}}^* + q_{\mathcal{I}}q_{\mathrm{st}}^*)}{2|q_{\mathrm{st}}|}\epsilon, & \text{if } q_{\mathrm{st}} \neq 0, \\ |q_{\mathcal{I}}|\epsilon, & \text{otherwise}, \end{cases}$$

which is a dual number.

A dual quaternion $q$ is said to be invertible if there exists a dual quaternion $p$ such that $pq = qp = 1$. A dual quaternion $q$ is invertible if and only if it is appreciable. In this case, its inverse is

$$q^{-1} = q_{\mathrm{st}}^{-1} - q_{\mathrm{st}}^{-1}q_{\mathcal{I}}q_{\mathrm{st}}^{-1}\epsilon.$$

We denote the set of unit dual quaternions by UDQ. A dual quaternion $q$ is called a unit dual quaternion if $|q| = 1$. Any unit dual quaternion is invertible, and its inverse coincides with its conjugate, i.e., $q^{-1} = q^*$. Moreover, $q$ is a unit dual quaternion if and only if $q_{\mathrm{st}}$ is a unit quaternion and

$$q_{\mathrm{st}}q_{\mathcal{I}}^* + q_{\mathcal{I}}q_{\mathrm{st}}^* = q_{\mathrm{st}}^*q_{\mathcal{I}} + q_{\mathcal{I}}^*q_{\mathrm{st}} = 0.$$

Consider two dual quaternions $p = p_{\mathrm{st}} + p_{\mathcal{I}}\epsilon$ and $q = q_{\mathrm{st}} + q_{\mathcal{I}}\epsilon$. Their sum is given by

$$p + q = (p_{\mathrm{st}} + q_{\mathrm{st}}) + (p_{\mathcal{I}} + q_{\mathcal{I}})\epsilon,$$

and their product is

$$pq = p_{\mathrm{st}}q_{\mathrm{st}} + (p_{\mathrm{st}}q_{\mathcal{I}} + p_{\mathcal{I}}q_{\mathrm{st}})\epsilon.$$

Denote the set of $n$-dimensional dual quaternion vectors by $\mathbb{DH}^n$. For $\boldsymbol{x} = (x_1, x_2, \ldots, x_n)$, the 2-norm is defined as

$$\|\boldsymbol{x}\|_2 := \begin{cases} \sqrt{\sum_{i=1}^n |x_i|^2}, & \boldsymbol{x}_{\mathrm{st}} \neq \boldsymbol{0}_n, \\ \sqrt{\sum_{i=1}^n |(x_i)_{\mathcal{I}}|^2}\,\epsilon, & \text{otherwise}. \end{cases}$$

# B. Representation of SO(3) and SE(3)

Let $\boldsymbol{R} \in \mathrm{SO}(3)$ be a rotation element and a point $\boldsymbol{v}$ under rotation $\boldsymbol{R}$ has the form of $\boldsymbol{R}\boldsymbol{v}$, Proposition B.1 implies that $qvq^*$ is a pure quaternion with the form of $0 + \boldsymbol{R}\boldsymbol{v}$. It gives a result for representing SO(3) element with unit quaternion (Garling, 2011, Sec. 4.9), which shows that quaternions can be used to obtain double cover of SO(3) group. More specifically, $\psi : \mathbb{UQ} \to \mathrm{SO}(3)$ is a 2-to-1 surjective group homomorphism of $\mathbb{UQ}$ onto SO(3) with kernel $\{1, -1\}$, such that $\psi^{-1}(\boldsymbol{R}) = \{q, -q\}$ for every $\boldsymbol{R} \in \mathrm{SO}(3)$.

**Proposition B.1** (Horn (1987, Sec. 3)). *For any unit quaternion with a representation of*

$$q = q_0 + \boldsymbol{q_R} = \cos\frac{\theta}{2} + \hat{\boldsymbol{u}}\sin\frac{\theta}{2}, \tag{11}$$

*a rotation of random vector $\boldsymbol{v_R} \in \mathbb{R}^3$ about an axis (denoted by unit vector $\hat{\boldsymbol{u}} \in \mathbb{R}^3$) by an angle $\theta$ can be represented by the operator*

$$\psi_q(\boldsymbol{v_R}) = qvq^*,$$

*where $v$ is treated as a quaternion with scalar part $0$ and vector part $\boldsymbol{v_R}$.*

Let $(\boldsymbol{R}, \boldsymbol{t}) \in \mathrm{SE}(3)$ represents an element of rigid motion with rotation $\boldsymbol{R} \in \mathrm{SO}(3)$ and translation $\boldsymbol{t} \in \mathbb{R}^3$. Consider a rigid line $\ell$ in the Plücker coordinates $(\hat{\boldsymbol{l}}, \boldsymbol{m})$. Similar to the quaternion and $\mathrm{SO}(3)$ case in the above subsection, a correspondence between $\mathrm{SE}(3)$ and unit dual quaternion can be constructed. To be operated on by dual quaternion, the line $\ell$ needs to be converted into a dual quaternion form $\ell = \hat{\boldsymbol{l}} + \epsilon \boldsymbol{m}$. The transformation operator is given by a unit dual quaternion which contains information of rotation and translation at the same time.

**Proposition B.2** (Daniilidis (1999, Sec. 3)). *Let $\ell_i = \hat{\boldsymbol{l}}_i + \epsilon \boldsymbol{m}_i, i = 1, 2$ be the dual quaternion describing of line $\ell_i$, line $\ell_1$ is transformed into line $\ell_2$ with a rotation $R \in SO(3)$ followed by a translation $\boldsymbol{t} \in \mathbb{R}^3$. Then*

$$\ell_2 = \phi_x(\ell_1) = x \ell_1 x^*,$$

*where $x$ is a unit dual quaternion in the form of*

$$x = \left(1 + \frac{\epsilon}{2}\boldsymbol{t}\right)q, \tag{12}$$

*with $q$ being the unit quaternion representation of $R$ following the rule in Proposition B.1 and $\boldsymbol{t} \in \mathbb{R}^3$ being the translation vector.*

Proposition B.2 offers an approach to represent elements in $\mathrm{SE}(3)$ in nearly the same way as we represent element in $\mathrm{SO}(3)$ by quaternions, where $(\boldsymbol{R}, \boldsymbol{t}) \in \mathrm{SE}(3)$ is described by $x = (1 + \frac{\epsilon}{2}\boldsymbol{t})q \in \mathrm{UDQ}$. $\phi$ can also be described as a 2-to-1 surjective group homomorphism from $\mathrm{UDQ}$ to $\mathrm{SE}(3)$, such that $\phi^{-1}(\boldsymbol{R}, \boldsymbol{t}) = \{-x, x\}$ for all $(\boldsymbol{R}, \boldsymbol{t}) \in \mathrm{SE}(3)$. Particularly, when $\boldsymbol{t} = 0$, $x$ becomes a pure quaternion $q$ and the $\mathrm{SE}(3)$ element degenerate into a $\mathrm{SO}(3)$ element accordingly. This shows a special case for Proposition B.2, which is consistent with Proposition B.1. We will focus on studying the synchronization problem on $\mathrm{SE}(3)$ case in the whole work, and this can also cover results in $\mathrm{SO}(3)$ by a degeneration of $\mathrm{SE}(3)$ elements.

## C. Proofs in Section 2

### C.1. Proof of Proposition 2.1

For any $\boldsymbol{x} \in \mathrm{UDQ}^n$, let $\boldsymbol{X} := \boldsymbol{x}\boldsymbol{x}^*$. Using $\|\boldsymbol{A}\|_F^2 = \mathrm{tr}(\boldsymbol{A}\boldsymbol{A}^*)$ and $\boldsymbol{C}^* = \boldsymbol{C}$ (Hermitian),

$$\|\boldsymbol{C} - \boldsymbol{X}\|_F^2 = \mathrm{tr}\big((\boldsymbol{C} - \boldsymbol{X})(\boldsymbol{C} - \boldsymbol{X})^*\big) = \mathrm{tr}(\boldsymbol{C}^2) - \mathrm{tr}(\boldsymbol{C}\boldsymbol{X} + \boldsymbol{X}\boldsymbol{C}) + \mathrm{tr}(\boldsymbol{X}^2).$$

Moreover, $\boldsymbol{X}^2 = \boldsymbol{x}\boldsymbol{x}^*\boldsymbol{x}\boldsymbol{x}^* = \boldsymbol{x}(\boldsymbol{x}^*\boldsymbol{x})\boldsymbol{x}^* = n\boldsymbol{X}$ and $\mathrm{tr}(\boldsymbol{X}) = \sum_{i=1}^n |\boldsymbol{x}_i|^2 = \boldsymbol{x}^*\boldsymbol{x} = n$, hence $\mathrm{tr}(\boldsymbol{X}^2) = n^2$. Therefore, minimizing $\|\boldsymbol{C} - \boldsymbol{x}\boldsymbol{x}^*\|_F^2$ is equivalent to maximizing $\mathrm{tr}(\boldsymbol{C}\boldsymbol{X} + \boldsymbol{X}\boldsymbol{C})$.

By the unitary decomposition of Hermitian dual quaternion matrices, $\boldsymbol{X} = \boldsymbol{U}\boldsymbol{\Sigma}\boldsymbol{U}^*$ with $\boldsymbol{\Sigma} = \mathrm{diag}(n, 0, \ldots, 0)$ and the first column of $\boldsymbol{U}$ equal to $\boldsymbol{u}_1 = \boldsymbol{x}/\sqrt{n}$ (cf. (Qi et al., 2022, Theorem 4.1)). Let $\boldsymbol{Y} := \boldsymbol{U}^*\boldsymbol{C}\boldsymbol{U} = (y_{ij})$. Using invariance of the (scalar/dual-number) trace under unitary similarity,

$$\mathrm{tr}(\boldsymbol{C}\boldsymbol{X} + \boldsymbol{X}\boldsymbol{C}) = \mathrm{tr}\big(\boldsymbol{U}(\boldsymbol{\Sigma}\boldsymbol{Y} + \boldsymbol{Y}\boldsymbol{\Sigma})\boldsymbol{U}^*\big) = \mathrm{tr}(\boldsymbol{\Sigma}\boldsymbol{Y} + \boldsymbol{Y}\boldsymbol{\Sigma}) = 2n \cdot y_{11}.$$

Since $y_{11} = \boldsymbol{u}_1^*\boldsymbol{C}\boldsymbol{u}_1 = (\boldsymbol{x}^*\boldsymbol{C}\boldsymbol{x})/n$, we obtain $\mathrm{tr}(\boldsymbol{C}\boldsymbol{X} + \boldsymbol{X}\boldsymbol{C}) = 2\boldsymbol{x}^*\boldsymbol{C}\boldsymbol{x}$. Hence, the argmin of (3) equals the argmax of (P), and the optimal values differ only by a factor of 2.

### C.2. Proof of Proposition 2.4

Let

$$\hat{z} := \arg\min_{z \in \mathrm{UDQ}} \|\boldsymbol{x} - \hat{\boldsymbol{x}}z\|_2.$$

Expanding the square and using $\|\boldsymbol{x}\|_2^2 = \|\hat{\boldsymbol{x}}\|_2^2 = n$ and $\|\hat{\boldsymbol{x}}z\|_2 = \|\hat{\boldsymbol{x}}\|_2$ for all $z \in \mathrm{UDQ}$,

$$\|\boldsymbol{x} - \hat{\boldsymbol{x}}z\|_2^2 = \|\boldsymbol{x}\|_2^2 + \|\hat{\boldsymbol{x}}z\|_2^2 - \big(\boldsymbol{x}^*\hat{\boldsymbol{x}}z + (\boldsymbol{x}^*\hat{\boldsymbol{x}}z)^*\big) = 2n - \big(z^*\hat{\boldsymbol{x}}^*\boldsymbol{x} + (z^*\hat{\boldsymbol{x}}^*\boldsymbol{x})^*\big).$$

Hence,

$$\hat{z} = \arg\min_{z \in \mathrm{UDQ}} \|\boldsymbol{x} - \hat{\boldsymbol{x}}z\|_2 = \arg\max_{z \in \mathrm{UDQ}} \big(z^*\hat{\boldsymbol{x}}^*\boldsymbol{x} + (z^*\hat{\boldsymbol{x}}^*\boldsymbol{x})^*\big). \tag{13}$$

Write $\hat{z}^* = c + d\epsilon$ and $\hat{\boldsymbol{x}}^*\boldsymbol{x} = a + b\epsilon$ with $c, d, a, b \in \mathbb{H}$. Since $\hat{z} \in \mathrm{UDQ}$, we have

$$c^*c = 1,$$

$$dc^* + cd^* = 0.$$

Moreover,

$$z^*\hat{\boldsymbol{x}}^*\boldsymbol{x} + (z^*\hat{\boldsymbol{x}}^*\boldsymbol{x})^* = ca + a^*c^* + (cb + b^*c^* + da + a^*d^*)\epsilon,$$

and thus (13) is equivalent to

$$\max_{c,d\in\mathbb{H}} \quad ca + a^*c^* + (cb + b^*c^* + da + a^*d^*)\epsilon \tag{15}$$
$$\text{s.t.} \quad dc^* + cd^* = 0, \qquad c^*c = 1.$$

Since the objective is a dual number and comparisons are made lexicographically (first by the primal part, then by the dual part), we consider first the primal (rotation) part:

$$\max_{c\in\mathbb{H}} \quad ca + a^*c^* \quad \text{s.t.} \quad c^*c = 1.$$

If $a \neq 0$, the maximizer is

$$c = \frac{a^*}{|a|},$$

and the dual part $d$ for maximizer satisfies $dc^* + cd^* = 0$, which yields a fixed dual part objective for (15).

If $a = 0$, the lexicographic maximization reduces to maximizing the dual component in (15). If $b \neq 0$, the dual component is maximized by choosing

$$c = \frac{b^*}{|b|}.$$

For this choice of $c$, any $d$ satisfying the unit-dual-quaternion constraint together with

$$db + b^*d^* = 0$$

attains the same maximum. If $b = 0$, then the dual component is also identically zero, and therefore any unit quaternion $c$ gives an optimal choice of $\hat{z}$. Thus, in all cases, the maximum is attained. Let

$$p := \max_{z\in\mathrm{UDQ}} z^*\hat{\boldsymbol{x}}^*\boldsymbol{x},$$

we have

$$\max_{z\in\mathrm{UDQ}} \left(z^*\hat{\boldsymbol{x}}^*\boldsymbol{x} + (z^*\hat{\boldsymbol{x}}^*\boldsymbol{x})^*\right) = p + p^* = 2|\hat{\boldsymbol{x}}^*\boldsymbol{x}|.$$

Plugging this into (13) yields

$$\mathrm{d}^2(\hat{\boldsymbol{x}}, \boldsymbol{x}) = \min_{z\in\mathrm{UDQ}} \|\boldsymbol{x} - \hat{\boldsymbol{x}}z\|_2^2 = 2n - 2|\hat{\boldsymbol{x}}^*\boldsymbol{x}|. \tag{16}$$

In addition,

$$\boldsymbol{x}^*\hat{\boldsymbol{x}}\hat{\boldsymbol{x}}^*\boldsymbol{x} \leq \left|\boldsymbol{x}^*\hat{\boldsymbol{x}}\hat{\boldsymbol{x}}^*\boldsymbol{x}\right| = \left|\langle\boldsymbol{x}, \hat{\boldsymbol{x}}\hat{\boldsymbol{x}}^*\boldsymbol{x}\rangle\right| \leq \|\boldsymbol{x}\|_2 \|\hat{\boldsymbol{x}}\hat{\boldsymbol{x}}^*\boldsymbol{x}\|_2 \tag{17}$$
$$= \sqrt{n}\,\|\hat{\boldsymbol{x}}\|_2\,|\hat{\boldsymbol{x}}^*\boldsymbol{x}| = n\,|\hat{\boldsymbol{x}}^*\boldsymbol{x}|.$$

Here, $\boldsymbol{x}^*\hat{\boldsymbol{x}}\hat{\boldsymbol{x}}^*\boldsymbol{x} = |\hat{\boldsymbol{x}}^*\boldsymbol{x}|^2$ is a dual number and hence admits a total order. The first inequality follows from Qi et al. (2022, Theorem 2), and the second from the Cauchy–Schwarz inequality (Ling et al., 2022, Proposition 4.7). Therefore,

$$|\hat{\boldsymbol{x}}^*\boldsymbol{x}| \geq \frac{1}{n}\boldsymbol{x}^*\hat{\boldsymbol{x}}\hat{\boldsymbol{x}}^*\boldsymbol{x} \implies \mathrm{d}^2(\hat{\boldsymbol{x}}, \boldsymbol{x}) = 2n - 2|\hat{\boldsymbol{x}}^*\boldsymbol{x}| \leq \frac{1}{n}\left(2n^2 - 2\boldsymbol{x}^*\hat{\boldsymbol{x}}\hat{\boldsymbol{x}}^*\boldsymbol{x}\right).$$

Recall that $\boldsymbol{C} = \hat{\boldsymbol{x}}\hat{\boldsymbol{x}}^* + \boldsymbol{\Delta}$. Then

$$2n^2 - 2\boldsymbol{x}^*\hat{\boldsymbol{x}}\hat{\boldsymbol{x}}^*\boldsymbol{x} = 2\hat{\boldsymbol{x}}^*\hat{\boldsymbol{x}}\hat{\boldsymbol{x}}^*\hat{\boldsymbol{x}} - 2\boldsymbol{x}^*\hat{\boldsymbol{x}}\hat{\boldsymbol{x}}^*\boldsymbol{x}$$
$$= 2\hat{\boldsymbol{x}}^*(\boldsymbol{C} - \boldsymbol{\Delta})\hat{\boldsymbol{x}} - 2\boldsymbol{x}^*(\boldsymbol{C} - \boldsymbol{\Delta})\boldsymbol{x}$$
$$\leq 2\boldsymbol{x}^*\boldsymbol{\Delta}\boldsymbol{x} - 2\hat{\boldsymbol{x}}^*\boldsymbol{\Delta}\hat{\boldsymbol{x}}$$
$$= (\boldsymbol{x} + \hat{\boldsymbol{x}})^*\boldsymbol{\Delta}(\boldsymbol{x} - \hat{\boldsymbol{x}}) + (\boldsymbol{x} - \hat{\boldsymbol{x}})^*\boldsymbol{\Delta}(\boldsymbol{x} + \hat{\boldsymbol{x}}),$$

where the inequality uses the assumption $x^* C x \geq \hat{x}^* C \hat{x}$. Combining the above display with (16)–(17) gives

$$d^2(\hat{x}, x) \leq \frac{1}{n}\left[(x + \hat{x})^* \Delta (x - \hat{x}) + (x - \hat{x})^* \Delta (x + \hat{x})\right] \leq \frac{2}{n}\left|(x + \hat{x})^* \Delta (x - \hat{x})\right|.$$

Using the Cauchy–Schwarz inequality and the definition of the induced operator norm, for any compatible $u, v$ we have $|u^* \Delta v| \leq \|u\|_2 \|\Delta\|_{\mathrm{op}} \|v\|_2$. Therefore,

$$d^2(\hat{x}, x) \leq \frac{2}{n} \|x + \hat{x}\|_2 \|\Delta\|_{\mathrm{op}} \|x - \hat{x}\|_2.$$

Finally, by right-multiplying $x$ with the optimal aligner $\hat{z}$ (which does not change the hypothesis $x^* C x \geq \hat{x}^* C \hat{x}$ nor the value of $d(\hat{x}, x)$), we may assume $d(\hat{x}, x) = \|x - \hat{x}\|_2$. Then $\|x + \hat{x}\|_2 \leq \|x\|_2 + \|\hat{x}\|_2 = 2\sqrt{n}$ and

$$d^2(\hat{x}, x) \leq \frac{2}{n} \cdot 2\sqrt{n} \|\Delta\|_{\mathrm{op}} \, d(\hat{x}, x) = \frac{4}{\sqrt{n}} \|\Delta\|_{\mathrm{op}} \, d(\hat{x}, x). \tag{18}$$

Taking standard parts of (18) gives

$$d_{\mathrm{st}}(x, \hat{x})^2 \leq \frac{4}{\sqrt{n}} \|\Delta\|_{\mathrm{op,st}} \, d_{\mathrm{st}}(x, \hat{x}).$$

Suppose $\|\Delta\|_{\mathrm{op,st}} \neq 0$. If $d_{\mathrm{st}}(x, \hat{x}) = 0$ and $\|\Delta\|_{\mathrm{op,st}} > 0$, the desired bound is immediate. Otherwise, $d_{\mathrm{st}}(x, \hat{x}) > 0$, and multiplying both sides of (18) by $d(x, \hat{x})^{-1}$ yields

$$d(x, \hat{x}) \leq \frac{4}{\sqrt{n}} \|\Delta\|_{\mathrm{op}},$$

which completes the proof.

### C.3. Proof of Lemma 2.5

Let $u := \mathcal{N}(y) \in \mathrm{UDQ}$, and write

$$y = y_{\mathrm{st}} + y_{\mathcal{I}}\epsilon, \qquad z = z_{\mathrm{st}} + z_{\mathcal{I}}\epsilon, \qquad u = u_{\mathrm{st}} + u_{\mathcal{I}}\epsilon,$$

with $y_{\mathrm{st}}, y_{\mathcal{I}}, z_{\mathrm{st}}, z_{\mathcal{I}}, u_{\mathrm{st}}, u_{\mathcal{I}} \in \mathbb{H}$. Recall that for any dual quaternion $q = q_{\mathrm{st}} + q_{\mathcal{I}}\epsilon$,

$$|q|^2 = qq^* = |q_{\mathrm{st}}|^2 + 2\epsilon \operatorname{sc}(q_{\mathrm{st}} q_{\mathcal{I}}^*),$$

which is a nonnegative dual number; hence it admits a total order, and it suffices to prove $|u - z|^2 \leq 4|y - z|^2$.

**Step 1: Expand $|u - z|^2$.** Since $u, z \in \mathrm{UDQ}$, we have $|u_{\mathrm{st}}| = |z_{\mathrm{st}}| = 1$ and $\operatorname{sc}(u_{\mathrm{st}} u_{\mathcal{I}}^*) = \operatorname{sc}(z_{\mathrm{st}} z_{\mathcal{I}}^*) = 0$. Thus,

$$\begin{aligned}
|u - z|^2 &= \left|(u_{\mathrm{st}} - z_{\mathrm{st}}) + (u_{\mathcal{I}} - z_{\mathcal{I}})\epsilon\right|^2 \\
&= |u_{\mathrm{st}} - z_{\mathrm{st}}|^2 + 2\epsilon \operatorname{sc}\left((u_{\mathrm{st}} - z_{\mathrm{st}})(u_{\mathcal{I}} - z_{\mathcal{I}})^*\right) \\
&= \left(2 - 2\operatorname{sc}(u_{\mathrm{st}} z_{\mathrm{st}}^*)\right) - 2\epsilon\left(\operatorname{sc}(u_{\mathrm{st}} z_{\mathcal{I}}^*) + \operatorname{sc}(z_{\mathrm{st}} u_{\mathcal{I}}^*)\right).
\end{aligned}$$

**Step 2: Expand $4|y - z|^2$.** Similarly,

$$\begin{aligned}
4|y - z|^2 &= 4\left|(y_{\mathrm{st}} - z_{\mathrm{st}}) + (y_{\mathcal{I}} - z_{\mathcal{I}})\epsilon\right|^2 \\
&= 4|y_{\mathrm{st}} - z_{\mathrm{st}}|^2 + 8\epsilon \operatorname{sc}\left((y_{\mathrm{st}} - z_{\mathrm{st}})(y_{\mathcal{I}} - z_{\mathcal{I}})^*\right) \\
&= 4\left(|y_{\mathrm{st}}|^2 + 1 - 2\operatorname{sc}(y_{\mathrm{st}} z_{\mathrm{st}}^*)\right) + 8\epsilon\left(\operatorname{sc}(y_{\mathrm{st}} y_{\mathcal{I}}^*) - \operatorname{sc}(y_{\mathrm{st}} z_{\mathcal{I}}^*) - \operatorname{sc}(z_{\mathrm{st}} y_{\mathcal{I}}^*)\right),
\end{aligned}$$

where we used $\operatorname{sc}(z_{\mathrm{st}} z_{\mathcal{I}}^*) = 0$ for $z \in \mathrm{UDQ}$.

**Step 3: Compare the standard parts.**

*Case 1:* $y_{\mathrm{st}} \neq 0$. By the definition of $\mathcal{N}$, we have $u_{\mathrm{st}} = y_{\mathrm{st}}/|y_{\mathrm{st}}|$. Let $r := |y_{\mathrm{st}}| > 0$ and $s := \mathrm{sc}(y_{\mathrm{st}} z_{\mathrm{st}}^*)$. Note that $|z_{\mathrm{st}}| = 1$ and $|\mathrm{sc}(y_{\mathrm{st}} z_{\mathrm{st}}^*)| \leq |y_{\mathrm{st}} z_{\mathrm{st}}^*| = |y_{\mathrm{st}}| = r$, hence $s \in [-r, r]$. Moreover,

$$\mathrm{sc}(u_{\mathrm{st}} z_{\mathrm{st}}^*) = \mathrm{sc}\left(\frac{y_{\mathrm{st}}}{r} z_{\mathrm{st}}^*\right) = \frac{s}{r}.$$

Therefore, the standard parts satisfy

$$\left(|u - z|^2\right)_{\mathrm{st}} = 2 - 2\frac{s}{r},$$
$$\left(4|y - z|^2\right)_{\mathrm{st}} = 4(r^2 + 1 - 2s) = 4r^2 + 4 - 8s.$$

We claim that for all $r > 0$ and $s \in [-r, r]$,

$$2 - 2\frac{s}{r} \leq 4r^2 + 4 - 8s. \tag{19}$$

Indeed, set $t := s/r \in [-1, 1]$. Then (19) is equivalent to

$$0 \leq 4r^2 + 2 + t(2 - 8r).$$

If $r \leq \frac{1}{4}$, then $2 - 8r \geq 0$ and the right-hand side is minimized at $t = -1$, giving $4r^2 + 2 - (2 - 8r) = 4r^2 + 8r \geq 0$. If $r \geq \frac{1}{4}$, then $2 - 8r \leq 0$ and the right-hand side is minimized at $t = 1$, giving $4r^2 + 2 + (2 - 8r) = 4(r - 1)^2 \geq 0$. This proves (19). Moreover, equality can only occur when $r = 1$ and $t = 1$, i.e., $|y_{\mathrm{st}}| = 1$ and $\mathrm{sc}(y_{\mathrm{st}} z_{\mathrm{st}}^*) = 1$, which implies $y_{\mathrm{st}} z_{\mathrm{st}}^* = 1$ and hence $u_{\mathrm{st}} = y_{\mathrm{st}} = z_{\mathrm{st}}$.

*Case 2:* $y_{\mathrm{st}} = 0$. In this case $\left(4|y - z|^2\right)_{\mathrm{st}} = 4$. By the definition of $\Pi$, we know $u_{\mathrm{st}} = y_{\mathcal{I}}/|y_{\mathcal{I}}|$ and $|u_{\mathrm{st}}| = 1$. Since $z_{\mathrm{st}}$ is also unit, we have $\mathrm{sc}(u_{\mathrm{st}} z_{\mathrm{st}}^*) \geq -1$, and thus

$$\left(|u - z|^2\right)_{\mathrm{st}} = 2 - 2\,\mathrm{sc}(u_{\mathrm{st}} z_{\mathrm{st}}^*) \leq 4 = \left(4|y - z|^2\right)_{\mathrm{st}}.$$

**Step 4: Conclude the dual-number inequality.** Since $|u - z|^2$ and $4|y - z|^2$ are dual numbers and the order is lexicographic (standard part first), the standard-part comparison implies $|u - z|^2 \leq 4|y - z|^2$ whenever the standard inequality is strict. In the only equality case from Case 1 (namely $u_{\mathrm{st}} = z_{\mathrm{st}}$), we have $u - z = (u_{\mathcal{I}} - z_{\mathcal{I}})\epsilon$, hence $|u - z|^2 = 0$, and also $y_{\mathrm{st}} = z_{\mathrm{st}}$ implies $|y - z|^2 = 0$. Thus the inequality still holds.

Finally, since both sides are nonnegative dual numbers, taking square roots preserves the order and yields $|u - z| \leq 2|y - z|$.

**C.4. Proof of Proposition 2.7**

Consider any $\boldsymbol{u} \in \mathrm{UDQ}^n$. If $\boldsymbol{y} \neq \boldsymbol{0}$, let $\mathcal{I} := \{i : y_i \neq 0\}$. For $i \in \mathcal{I}$, Remark 2.6 has shown that $\mathcal{N}(y_i) = \Pi(y_i)$, then

$$|\widetilde{y}_i - y_i| = |\Pi(y_i) - y_i| \leq |u_i - y_i|.$$

For $i \notin \mathcal{I}$,

$$|\widetilde{y}_i - y_i| = |\mathcal{N}(\boldsymbol{e}^* \boldsymbol{y}) - 0| = 1 = |u_i - y_i|.$$

If $\boldsymbol{y} = \boldsymbol{0}$,

$$\|\widetilde{\boldsymbol{y}} - \boldsymbol{y}\|_2^2 = \|\boldsymbol{1} - \boldsymbol{0}\|_2^2 = n = \|\boldsymbol{u}\|_2^2 = \|\boldsymbol{u} - \boldsymbol{y}\|_2^2.$$

Hence,

$$\|\widetilde{\boldsymbol{y}} - \boldsymbol{y}\|_2^2 = \sum_{i \notin \mathcal{I}} |\widetilde{y}_i - y_i|^2 + \sum_{i \in \mathcal{I}} |\widetilde{y}_i - y_i|^2 \leq \|\boldsymbol{u} - \boldsymbol{y}\|_2^2$$

for any $\boldsymbol{u} \in \mathrm{UDQ}^n$, which shows $\widetilde{\boldsymbol{y}} = \Pi(\boldsymbol{y})$.

### C.5. Proof of Theorem 2.8

Let $\boldsymbol{x}$ be a dominant right eigenvector of $\boldsymbol{C}$ scaled such that $\|\boldsymbol{x}\|_2^2 = n$. Since $\boldsymbol{x}$ is defined up to a global right multiplication by any $z \in \mathrm{UDQ}$, we may choose $z_\star \in \mathrm{UDQ}$ attaining the minimum in the definition of $\mathrm{d}(\boldsymbol{x}, \hat{\boldsymbol{x}})$ and replace $\boldsymbol{x} \leftarrow \boldsymbol{x} z_\star$, so that

$$\|\boldsymbol{x} - \hat{\boldsymbol{x}}\|_2 = \mathrm{d}(\boldsymbol{x}, \hat{\boldsymbol{x}}).$$

Define $\widetilde{\boldsymbol{x}} = \Pi(\boldsymbol{x})$ as in (5). Then

$$\mathrm{d}(\widetilde{\boldsymbol{x}}, \hat{\boldsymbol{x}}) = \min_{z \in \mathrm{UDQ}} \|\widetilde{\boldsymbol{x}} - \hat{\boldsymbol{x}} z\|_2 \leq \|\widetilde{\boldsymbol{x}} - \hat{\boldsymbol{x}}\|_2.$$

Let $\mathcal{I} := \{i : x_i \neq 0\}$,

$$
\begin{aligned}
\|\Pi(\boldsymbol{x}) - \hat{\boldsymbol{x}}\|_2^2 &= \sum_{i \in \mathcal{I}} |\mathcal{N}(\boldsymbol{x}_i) - \hat{\boldsymbol{x}}_i|^2 + \sum_{i \notin \mathcal{I}} |\mathcal{N}(\boldsymbol{e}^* \boldsymbol{x}) - \hat{\boldsymbol{x}}_i|^2 \\
&\leq \sum_{i \in \mathcal{I}} 4|\boldsymbol{x}_i - \hat{\boldsymbol{x}}_i|^2 + 4\big(n - |\mathcal{I}|\big) \\
&= \sum_{i \in \mathcal{I}} 4|\boldsymbol{x}_i - \hat{\boldsymbol{x}}_i|^2 + \sum_{i \notin \mathcal{I}} 4|\boldsymbol{x}_i - \hat{\boldsymbol{x}}_i|^2 \\
&= 4\|\boldsymbol{x} - \hat{\boldsymbol{x}}\|_2^2,
\end{aligned}
$$

where the second last equality holds from

$$n - |\mathcal{I}| = \sum_{i \notin \mathcal{I}} |\hat{\boldsymbol{x}}_i|^2 = \sum_{i \notin \mathcal{I}} |\boldsymbol{x}_i - \hat{\boldsymbol{x}}_i|^2.$$

Next, using Lemma 2.5 componentwise and summing over $i = 1, \ldots, n$, we obtain

$$\|\Pi(\boldsymbol{x}) - \hat{\boldsymbol{x}}\|_2^2 = \sum_{i=1}^n |\Pi(\boldsymbol{x}_i) - \hat{\boldsymbol{x}}_i|^2 \leq \sum_{i=1}^n 4|\boldsymbol{x}_i - \hat{\boldsymbol{x}}_i|^2 = 4\|\boldsymbol{x} - \hat{\boldsymbol{x}}\|_2^2,$$

hence $\|\widetilde{\boldsymbol{x}} - \hat{\boldsymbol{x}}\|_2 \leq 2\|\boldsymbol{x} - \hat{\boldsymbol{x}}\|_2$. Therefore,

$$\mathrm{d}(\widetilde{\boldsymbol{x}}, \hat{\boldsymbol{x}}) \leq \|\widetilde{\boldsymbol{x}} - \hat{\boldsymbol{x}}\|_2 \leq 2\|\boldsymbol{x} - \hat{\boldsymbol{x}}\|_2 = 2\,\mathrm{d}(\boldsymbol{x}, \hat{\boldsymbol{x}}) \leq 2 \cdot \frac{4\|\boldsymbol{\Delta}\|_{\mathrm{op}}}{\sqrt{n}} = \frac{8\|\boldsymbol{\Delta}\|_{\mathrm{op}}}{\sqrt{n}},$$

where the last inequality follows from Proposition 2.4.

## D. Proof of Theorem 3.2

The proof for Theorem 3.2 is closely based on the following results.

**Lemma D.1.** *For any $\boldsymbol{y} \in \mathbb{DH}^n$ and any real scalar $a > 0$, the projection $\Pi(\cdot)$ onto $\mathrm{UDQ}^n$ satisfies*

$$\Pi(a\boldsymbol{y}) = \Pi(\boldsymbol{y}).$$

This result implies that $\Pi(\cdot)$ depends only on the direction of its argument, not its magnitude. Thus, scaling the input by a positive real scalar leaves the projection unchanged.

**Lemma D.2.** *For any $x, y \in \mathbb{DH}^n$, we have*

$$\|\Pi(\boldsymbol{x} + \Pi(\boldsymbol{x}) + \boldsymbol{y}) - \Pi(\boldsymbol{x})\|_2 \leq 2\|\boldsymbol{y}\|_2.$$

This lemma can be regarded as a analogical version of Lemma 2 given in Liu et al. (2023). Although some slight differences were demonstrated by the unit dual quaternion set, we can still use a semblable way to give the proof here. As an extension of Lemma D.2, we have

**Lemma D.3.** *For any $p, q \in \mathbb{DH}^n$ and $r \in \mathrm{UDQ}^n$, we have*

$$\left\|\Pi(p+q) - r\right\|_2 \leq 2\|q - r\|_2 + 3\left\|\Pi(p) - r\right\|_2. \tag{20}$$

By letting $p = cr$ for $c \in \mathbb{R}_{++}$ in (20), and forcing $c$ to zero, we can find that Lemma D.3 becomes

$$\|\Pi(q) - r\|_2 \leq 2\|q - r\|_2, \tag{21}$$

which coincides with the result in Lemma 2.5.

**Lemma D.4.** *Let $r \in \mathrm{UDQ}^n$ and $h \in \mathbb{DH}^n$. Assume that*

$$\min_{1 \leq i \leq n} |(r_i + h_i)_{\mathrm{st}}| \geq \frac{1}{2}, \tag{22}$$

*then*

$$\left\|\left(\Pi(r+h) - r\right)_{\mathcal{I}}\right\|_2 \leq 2\|h_{\mathcal{I}}\|_2 + 6 \max_{1 \leq i \leq n} |(r_i)_{\mathcal{I}}| \, \|h_{\mathrm{st}}\|_2. \tag{23}$$

To give a full picture of the proof for Theorem 3.2, we introduce the following lemma that demonstrates geometric properties of $\mathrm{UDQ}^n$.

**Lemma D.5.** *For any $x^k \in \mathrm{UDQ}^n$, there exists an optimizer*

$$z_k \in \mathrm{argmin}_{z \in \mathrm{UDQ}} \|x^k - \hat{x}z\|_2, \tag{24}$$

*we have*

$$\left|\hat{x}^* x^k - nz_k\right| = \frac{1}{2} \, \mathrm{d}(x^k, \hat{x})^2.$$

The right hand side of (24) is the squared distance between $x^k$ and $\hat{x}$ after alignment, thus Lemma D.5 can be interpreted as an error bound condition. This lemma implies that sequence $\{x^{(k)}\}_{k \geq 0}$ generated by DQGPM also satisfies an error bound condition, which plays an important role in proving Theorem 3.2 by bounding estimation error with distance terms. Additionally, as a degenerated case for SE(3), elements in SO(3) represented by unit dual quaternions with zero infinitesimal part should also satisfy (24). This extension on SO(3) meets the error bound attained for SO($d$) in Liu et al. (2023), but (24) is tighter for SO(3) case since the equality always holds.

Lastly, the below theorem gives a recursive relation for the dual part of the sequence $\{x^{(k)}\}_{k \geq 0}$ generated by DQGPM.

**Lemma D.6.** *Under Assumption of Theorem 3.2, the aligned dual residual*

$$\delta_{\mathcal{I}}^k = \left\|\left(x^k - r^k\right)_{\mathcal{I}}\right\|_2, \quad \text{where } r^k = \hat{x}z_k,$$

*satisfies*

$$\begin{aligned}
\delta_{\mathcal{I}}^{k+1} \leq & \left(\frac{2}{\sqrt{n}} \, \mathrm{d}_{\mathrm{st}}(x^k, \hat{x}) + \frac{2}{n}\|\Delta\|_{\mathrm{op,st}}\right) \delta_{\mathcal{I}}^k \\
& + \left(\frac{2}{n}\|\Delta\|_{\mathrm{op},\mathcal{I}} + \frac{6B + 2\gamma}{n}\|\Delta\|_{\mathrm{op,st}} + \frac{4B + \gamma}{\sqrt{n}} \, \mathrm{d}_{\mathrm{st}}(x^k, \hat{x})\right) \mathrm{d}_{\mathrm{st}}(x^k, \hat{x}) \\
& + \frac{2}{\sqrt{n}}\left(\|\Delta\|_{\mathrm{op},\mathcal{I}} + B\|\Delta\|_{\mathrm{op,st}}\right) + \frac{6B + 2\gamma}{n}\|(\Delta\hat{x})_{\mathrm{st}}\|_2.
\end{aligned} \tag{25}$$

*Proof of Theorem 3.2.* For each $k$, select $z_k \in \arg\min_{z \in \mathrm{UDQ}} \|x^k - \hat{x}z\|_2$ such that Lemma D.5 holds (such a minimizer exists by Lemma D.5). Then

$$\mathrm{d}(x^{k+1}, \hat{x}) = \|x^{k+1} - \hat{x}z_{k+1}\|_2 \leq \|x^{k+1} - \hat{x}z_k\|_2 = \left\|\Pi\left(\frac{2}{n}Cx^k\right) - \hat{x}z_k\right\|_2,$$

where the last equality holds from Lemma D.1. Using Lemma D.3 with

$$q = \hat{x}z_k + 2\left[\frac{1}{n}(C - \Delta)x^k - \hat{x}z_k\right] + \frac{2}{n}\Delta(x^k - \hat{x}z_k), \quad p = \hat{x}z_k + \frac{2}{n}\Delta\hat{x}z_k, \quad r = \hat{x}z_k,$$

(note that $\boldsymbol{p} + \boldsymbol{q} = \frac{2}{n}\boldsymbol{C}\boldsymbol{x}^k$), we obtain

$$\left\|\Pi\left(\frac{2}{n}\boldsymbol{C}\boldsymbol{x}^k\right) - \hat{\boldsymbol{x}}z_k\right\|_2 \leq 4\left\|\frac{1}{n}(\boldsymbol{C}-\boldsymbol{\Delta})\boldsymbol{x}^k - \hat{\boldsymbol{x}}z_k + \frac{1}{n}\boldsymbol{\Delta}(\boldsymbol{x}^k - \hat{\boldsymbol{x}}z_k)\right\|_2 + 3\|\Pi(\boldsymbol{p}) - \boldsymbol{r}\|_2$$

$$\leq \frac{4}{n}\|(\boldsymbol{C}-\boldsymbol{\Delta})\boldsymbol{x}^k - n\hat{\boldsymbol{x}}z_k\|_2 + \frac{4}{n}\|\boldsymbol{\Delta}(\boldsymbol{x}^k - \hat{\boldsymbol{x}}z_k)\|_2 + 3\|\Pi(\boldsymbol{p}) - \boldsymbol{r}\|_2.$$

For the first term, since $\boldsymbol{C} - \boldsymbol{\Delta} = \hat{\boldsymbol{x}}\hat{\boldsymbol{x}}^*$ and $\|\hat{\boldsymbol{x}}\|_2 = \sqrt{n}$, Lemma D.5 yields

$$\|(\boldsymbol{C}-\boldsymbol{\Delta})\boldsymbol{x}^k - n\hat{\boldsymbol{x}}z_k\|_2 = \|\hat{\boldsymbol{x}}\|_2\,|\hat{\boldsymbol{x}}^*\boldsymbol{x}^k - nz_k| = \frac{\sqrt{n}}{2}\|\boldsymbol{x}^k - \hat{\boldsymbol{x}}z_k\|_2^2.$$

For the second term, by the definition of operator norm,

$$\|\boldsymbol{\Delta}(\boldsymbol{x}^k - \hat{\boldsymbol{x}}z_k)\|_2 \leq \|\boldsymbol{\Delta}\|_{\mathrm{op}}\|\boldsymbol{x}^k - \hat{\boldsymbol{x}}z_k\|_2.$$

For the third term, we use the right-equivariance of the projection: $\Pi(\boldsymbol{y}z) = \Pi(\boldsymbol{y})z$ for $\boldsymbol{y} \in \mathbb{DH}^n$ and $z \in \mathrm{UDQ}$. Thus,

$$\|\Pi(\boldsymbol{p}) - \boldsymbol{r}\|_2 = \left\|\Pi\left(\hat{\boldsymbol{x}} + \frac{2}{n}\boldsymbol{\Delta}\hat{\boldsymbol{x}}\right)z_k - \hat{\boldsymbol{x}}z_k\right\|_2 = \left\|\Pi\left(\hat{\boldsymbol{x}} + \frac{2}{n}\boldsymbol{\Delta}\hat{\boldsymbol{x}}\right) - \hat{\boldsymbol{x}}\right\|_2 \leq 2\left\|\frac{2}{n}\boldsymbol{\Delta}\hat{\boldsymbol{x}}\right\|_2,$$

where the inequality follows from Lemma D.3. Combining the bounds gives

$$\mathrm{d}(\boldsymbol{x}^{k+1}, \hat{\boldsymbol{x}}) \leq \left(\frac{2}{\sqrt{n}}\mathrm{d}(\boldsymbol{x}^k, \hat{\boldsymbol{x}}) + \frac{4}{n}\|\boldsymbol{\Delta}\|_{\mathrm{op}}\right)\mathrm{d}(\boldsymbol{x}^k, \hat{\boldsymbol{x}}) + \frac{12}{n}\|\boldsymbol{\Delta}\hat{\boldsymbol{x}}\|_2. \tag{26}$$

First, we analyse the standard part. Recall that inequality (26) is in the lexicographic order of dual numbers. Hence it directly implies the following recursive relation for the standard component:

$$\mathrm{d}_{\mathrm{st}}(\boldsymbol{x}^{k+1}, \hat{\boldsymbol{x}}) \leq \left(\frac{2}{\sqrt{n}}\mathrm{d}_{\mathrm{st}}(\boldsymbol{x}^k, \hat{\boldsymbol{x}}) + \frac{4}{n}\|\boldsymbol{\Delta}\|_{\mathrm{op,st}}\right)\mathrm{d}_{\mathrm{st}}(\boldsymbol{x}^k, \hat{\boldsymbol{x}}) + \frac{12}{n}\|(\boldsymbol{\Delta}\hat{\boldsymbol{x}})_{\mathrm{st}}\|_2.$$

At this stage, we only use the standard component of inequality (26); the infinitesimal component will be controlled separately. We prove by induction that, for all $k = 0, 1, 2, \ldots,$

$$\mathrm{d}_{\mathrm{st}}(\boldsymbol{x}^k, \hat{\boldsymbol{x}}) \leq \frac{\sqrt{n}}{25}. \tag{27}$$

For the base case $k = 0$, the claim follows immediately from Theorem 2.8 with the assumption $\|\boldsymbol{\Delta}\|_{\mathrm{op,st}} \leq \frac{n}{350}$. Assume next that (27) holds for some $k \geq 0$. It remains to show that $\mathrm{d}_{\mathrm{st}}(\boldsymbol{x}^{k+1}, \hat{\boldsymbol{x}}) \leq \frac{\sqrt{n}}{25}$. Combining the inductive hypothesis with again the noise assumptions, the standard part for right-hand side of inequality (26) and $\|\boldsymbol{\Delta}\hat{\boldsymbol{x}}\|_2 \leq \sqrt{n}\|\boldsymbol{\Delta}\|_{\mathrm{op}}$ yield

$$\left(\frac{2}{\sqrt{n}}\mathrm{d}_{\mathrm{st}}(\boldsymbol{x}^k, \hat{\boldsymbol{x}}) + \frac{4}{n}\|\boldsymbol{\Delta}\|_{\mathrm{op,st}}\right)\mathrm{d}_{\mathrm{st}}(\boldsymbol{x}^k, \hat{\boldsymbol{x}}) + \frac{12}{\sqrt{n}}\|\boldsymbol{\Delta}\|_{\mathrm{op,st}} \leq \frac{16}{175}\mathrm{d}_{\mathrm{st}}(\boldsymbol{x}^k, \hat{\boldsymbol{x}}) + \frac{6}{175}\sqrt{n} < \frac{\sqrt{n}}{25},$$

which closes the induction and proves (27) for all $k \geq 0$.

Using (26) again together with the fact that $\mathrm{d}_{\mathrm{st}}(\boldsymbol{x}^k, \hat{\boldsymbol{x}}) \leq \frac{\sqrt{n}}{25}$, we obtain the linear recursion for standard part:

$$\mathrm{d}_{\mathrm{st}}(\boldsymbol{x}^k, \hat{\boldsymbol{x}}) \leq \frac{16}{175}\mathrm{d}_{\mathrm{st}}(\boldsymbol{x}^{k-1}, \hat{\boldsymbol{x}}) + \frac{12}{n}\|\boldsymbol{\Delta}\hat{\boldsymbol{x}}\|_{2,st}.$$

Unrolling it gives

$$\mathrm{d}_{\mathrm{st}}(\boldsymbol{x}^k, \hat{\boldsymbol{x}}) \leq \left(\frac{16}{175}\right)^k \mathrm{d}_{\mathrm{st}}(\boldsymbol{x}^0, \hat{\boldsymbol{x}}) + \left(1 + \frac{16}{175} + \left(\frac{16}{175}\right)^2 + \cdots\right)\frac{12}{n}\|\boldsymbol{\Delta}\hat{\boldsymbol{x}}\|_{2,st}$$

$$\leq \left(\frac{16}{175}\right)^k \mathrm{d}_{\mathrm{st}}(\boldsymbol{x}^0, \hat{\boldsymbol{x}}) + \frac{700}{53n}\|\boldsymbol{\Delta}\hat{\boldsymbol{x}}\|_{2,st}$$

$$< \left(\frac{1}{10}\right)^k \mathrm{d}_{\mathrm{st}}(\boldsymbol{x}^0, \hat{\boldsymbol{x}}) + \frac{700}{53n}\|\boldsymbol{\Delta}\hat{\boldsymbol{x}}\|_{2,st}.$$

Second, we control the dual coefficient. Unlike the standard component, this part is not obtained by taking the infinitesimal component of inequality (26). Instead, we use the aligned dual residual. Let

$$\boldsymbol{u} = \boldsymbol{x}^k - \hat{\boldsymbol{x}} z_k = \boldsymbol{u}_{\mathrm{st}} + \epsilon \boldsymbol{u}_{\mathcal{I}}.$$

If $\boldsymbol{u}_{\mathrm{st}} \neq 0$, then by the definition of the dual-valued norm, we know that

$$(\|\boldsymbol{u}\|_2)_{\mathcal{I}} = \frac{\sum_i \mathrm{sc}\big((\boldsymbol{u}_i)_{\mathrm{st}}^*(\boldsymbol{u}_i)_{\mathcal{I}}\big)}{\|\boldsymbol{u}_{\mathrm{st}}\|_2}.$$

By the ordinary Cauchy–Schwarz inequality on $\mathbb{H}^n \simeq \mathbb{R}^{4n}$, we have

$$|(\|\boldsymbol{u}\|_2)_{\mathcal{I}}| \leq \|\boldsymbol{u}_{\mathcal{I}}\|_2.$$

If $\boldsymbol{u}_{\mathrm{st}} = 0$, then

$$\|\boldsymbol{u}\|_2 = \epsilon \|\boldsymbol{u}_{\mathcal{I}}\|_2,$$

and the same inequality is immediate. Since $z_k$ is chosen as the alignment element in the definition of $\mathrm{d}(\boldsymbol{x}^k, \hat{\boldsymbol{x}})$, we have $\mathrm{d}(\boldsymbol{x}^k, \hat{\boldsymbol{x}}) = \|\boldsymbol{x}^k - \hat{\boldsymbol{x}} z_k\|_2$. Therefore,

$$\left| \mathrm{d}_{\mathcal{I}}(\boldsymbol{x}^k, \hat{\boldsymbol{x}}) \right| \leq \left\| \big(\boldsymbol{x}^k - \hat{\boldsymbol{x}} z_k\big)_{\mathcal{I}} \right\|_2 = \delta_{\mathcal{I}}^k.$$

It remains to bound $\delta_{\mathcal{I}}^k$. By Lemma D.6, we have

$$\delta_{\mathcal{I}}^{k+1} \leq \left( \frac{2}{\sqrt{n}} \mathrm{d}_{\mathrm{st}}(\boldsymbol{x}^k, \hat{\boldsymbol{x}}) + \frac{2}{n} \|\boldsymbol{\Delta}\|_{\mathrm{op},\mathrm{st}} \right) \delta_{\mathcal{I}}^k + \left( \frac{2}{n} \|\boldsymbol{\Delta}\|_{\mathrm{op},\mathcal{I}} + \frac{6B + 2\gamma}{n} \|\boldsymbol{\Delta}\|_{\mathrm{op},\mathrm{st}} + \frac{4B + \gamma}{\sqrt{n}} \mathrm{d}_{\mathrm{st}}(\boldsymbol{x}^k, \hat{\boldsymbol{x}}) \right) \mathrm{d}_{\mathrm{st}}(\boldsymbol{x}^k, \hat{\boldsymbol{x}})$$

$$+ \frac{2}{\sqrt{n}} \left( \|\boldsymbol{\Delta}\|_{\mathrm{op},\mathcal{I}} + B \|\boldsymbol{\Delta}\|_{\mathrm{op},\mathrm{st}} \right) + \frac{6B + 2\gamma}{n} \|(\boldsymbol{\Delta}\hat{\boldsymbol{x}})_{\mathrm{st}}\|_2.$$

The standard-part basin bound and the noise assumption imply that the coefficient of $\delta_{\mathcal{I}}^k$ is strictly contractive:

$$\frac{2}{\sqrt{n}} \mathrm{d}_{\mathrm{st}}(\boldsymbol{x}^k, \hat{\boldsymbol{x}}) + \frac{2}{n} \|\boldsymbol{\Delta}\|_{\mathrm{op},\mathrm{st}} \leq \frac{2}{25} + \frac{2}{350} = \frac{3}{35} < \frac{1}{10}.$$

Moreover, by (27),

$$\frac{4B + \gamma}{\sqrt{n}} \mathrm{d}_{\mathrm{st}}(\boldsymbol{x}^k, \hat{\boldsymbol{x}})^2 \leq \frac{4B + \gamma}{25} \mathrm{d}_{\mathrm{st}}(\boldsymbol{x}^k, \hat{\boldsymbol{x}}).$$

Thus the one-step bound simplifies to

$$\delta_{\mathcal{I}}^{k+1} \leq \frac{1}{10} \delta_{\mathcal{I}}^k + \left( \frac{2}{n} \|\boldsymbol{\Delta}\|_{\mathrm{op},\mathcal{I}} + \frac{6B + 2\gamma}{n} \|\boldsymbol{\Delta}\|_{\mathrm{op},\mathrm{st}} + \frac{4B + \gamma}{25} \right) \mathrm{d}_{\mathrm{st}}(\boldsymbol{x}^k, \hat{\boldsymbol{x}})$$

$$+ \frac{2}{\sqrt{n}} \left( \|\boldsymbol{\Delta}\|_{\mathrm{op},\mathcal{I}} + B \|\boldsymbol{\Delta}\|_{\mathrm{op},\mathrm{st}} \right) + \frac{6B + 2\gamma}{n} \|(\boldsymbol{\Delta}\hat{\boldsymbol{x}})_{\mathrm{st}}\|_2.$$

Unrolling this recursion gives, for every $k \geq 1$,

$$\delta_{\mathcal{I}}^k \leq \left( \frac{1}{10} \right)^k \delta_{\mathcal{I}}^0 + \left( \frac{2}{n} \|\boldsymbol{\Delta}\|_{\mathrm{op},\mathcal{I}} + \frac{6B + 2\gamma}{n} \|\boldsymbol{\Delta}\|_{\mathrm{op},\mathrm{st}} + \frac{4B + \gamma}{25} \right) \sum_{\ell=0}^{k-1} \left( \frac{1}{10} \right)^{k-1-\ell} \mathrm{d}_{\mathrm{st}}(\boldsymbol{x}^\ell, \hat{\boldsymbol{x}})$$

$$+ \left[ \frac{2}{\sqrt{n}} \left( \|\boldsymbol{\Delta}\|_{\mathrm{op},\mathcal{I}} + B \|\boldsymbol{\Delta}\|_{\mathrm{op},\mathrm{st}} \right) + \frac{6B + 2\gamma}{n} \|(\boldsymbol{\Delta}\hat{\boldsymbol{x}})_{\mathrm{st}}\|_2 \right] \sum_{\ell=0}^{k-1} \left( \frac{1}{10} \right)^{k-1-\ell}.$$

We now substitute the standard-part estimate. From (6), we obtain

$$\delta_{\mathcal{I}}^k \leq \left( \frac{1}{10} \right)^k \delta_{\mathcal{I}}^0 + \left( \frac{2}{n} \|\boldsymbol{\Delta}\|_{\mathrm{op},\mathcal{I}} + \frac{6B + 2\gamma}{n} \|\boldsymbol{\Delta}\|_{\mathrm{op},\mathrm{st}} + \frac{4B + \gamma}{25} \right) \left[ \frac{\sqrt{n}}{25} k \left( \frac{1}{10} \right)^{k-1} + \frac{7000}{477n} \|(\boldsymbol{\Delta}\hat{\boldsymbol{x}})_{\mathrm{st}}\|_2 \right]$$

$$+ \frac{10}{9} \left[ \frac{2}{\sqrt{n}} \left( \|\boldsymbol{\Delta}\|_{\mathrm{op},\mathcal{I}} + B \|\boldsymbol{\Delta}\|_{\mathrm{op},\mathrm{st}} \right) + \frac{6B + 2\gamma}{n} \|(\boldsymbol{\Delta}\hat{\boldsymbol{x}})_{\mathrm{st}}\|_2 \right].$$

This proves (10) combining the assumptions on noise $\|\boldsymbol{\Delta}\|_{\mathrm{op},\mathrm{st}}$ and $\|\boldsymbol{\Delta}\|_{\mathrm{op},\mathcal{I}}$. Together with $|\mathrm{d}_{\mathcal{I}}(\boldsymbol{x}^k, \hat{\boldsymbol{x}})| \leq \delta_{\mathcal{I}}^k$, it also gives the desired bound on the infinitesimal coefficient of the dual-valued distance. $\square$

## D.1. Proof of Lemma D.1

We first show that the normalization operator $\mathcal{N}(\cdot)$ is scale-invariant for any nonzero dual quaternion under positive real scaling, i.e.,

$$\mathcal{N}(ax) = \mathcal{N}(x), \quad \forall\, x \in \mathbb{DH} \setminus \{0\},\ a \in \mathbb{R}_{++}. \tag{28}$$

If $x_{\mathrm{st}} \neq 0$, then by the definition of $\mathcal{N}(\cdot)$ and $|ax_{\mathrm{st}}| = a|x_{\mathrm{st}}|$, the standard part satisfies

$$\frac{(ax)_{\mathrm{st}}}{|(ax)_{\mathrm{st}}|} = \frac{ax_{\mathrm{st}}}{a|x_{\mathrm{st}}|} = \frac{x_{\mathrm{st}}}{|x_{\mathrm{st}}|},$$

and the image part satisfies

$$\frac{ax_{\mathcal{I}}}{|ax_{\mathrm{st}}|} - \frac{ax_{\mathrm{st}}}{|ax_{\mathrm{st}}|}\mathrm{sc}\!\left(\frac{(ax_{\mathrm{st}})^*(ax_{\mathcal{I}})}{|ax_{\mathrm{st}}|\,|ax_{\mathrm{st}}|}\right) = \frac{x_{\mathcal{I}}}{|x_{\mathrm{st}}|} - \frac{x_{\mathrm{st}}}{|x_{\mathrm{st}}|}\mathrm{sc}\!\left(\frac{x_{\mathrm{st}}^* x_{\mathcal{I}}}{|x_{\mathrm{st}}|\,|x_{\mathrm{st}}|}\right),$$

which establishes (28) in this case.

If $x_{\mathrm{st}} = 0$ and $x_{\mathcal{I}} \neq 0$, similarly,

$$\frac{(ax)_{\mathcal{I}}}{|(ax)_{\mathcal{I}}|} = \frac{ax_{\mathcal{I}}}{a|x_{\mathcal{I}}|} = \frac{x_{\mathcal{I}}}{|x_{\mathcal{I}}|},$$

and the feasibility condition for $u_{\mathcal{I}}$ is invariant under positive scaling. Hence, (28) holds in this case as well.

Next, we prove that the projection $\Pi(\cdot)$ is scale-invariant under positive real scaling factor for any $\boldsymbol{y} \in \mathbb{DH}^n$. If $\boldsymbol{y} = \boldsymbol{0}$, then $a\boldsymbol{y} = \boldsymbol{0}$, and by definition, $\Pi(a\boldsymbol{y}) = \boldsymbol{1} = \Pi(\boldsymbol{y})$. Otherwise, let $\tilde{\boldsymbol{y}} := \Pi(\boldsymbol{y})$ and $\widetilde{\boldsymbol{y}}_a := \Pi(a\boldsymbol{y})$. For each index $i$, if $y_i \neq 0$, then $(a\boldsymbol{y})_i \neq 0$, and by (28),

$$\left[\widetilde{\boldsymbol{y}}_a\right]_i = \mathcal{N}\big((a\boldsymbol{y})_i\big) = \mathcal{N}(ay_i) = \mathcal{N}(y_i) = \tilde{y}_i.$$

If $y_i = 0$, the definition uses $\mathcal{N}(\boldsymbol{e}^*\boldsymbol{y})$ for any $\boldsymbol{e}$ with $\boldsymbol{e}^*\boldsymbol{y} \neq 0$. Since $a > 0$, the same $\boldsymbol{e}$ satisfies $\boldsymbol{e}^*(a\boldsymbol{y}) = a(\boldsymbol{e}^*\boldsymbol{y}) \neq 0$, and again by (28),

$$\left[\widetilde{\boldsymbol{y}}_a\right]_i = \mathcal{N}\big(\boldsymbol{e}^*(a\boldsymbol{y})\big) = \mathcal{N}\big(a(\boldsymbol{e}^*\boldsymbol{y})\big) = \mathcal{N}(\boldsymbol{e}^*\boldsymbol{y}) = \tilde{y}_i.$$

Thus, $\Pi(a\boldsymbol{y}) = \Pi(\boldsymbol{y})$ for all $a \in \mathbb{R}_{++}$ and $\boldsymbol{y} \in \mathbb{DH}^n$.

## D.2. Proof of Lemma D.2

*Proof.* **Notation used in this proof:** For given $\boldsymbol{u} = (u_1, u_2, \ldots, u_n), \boldsymbol{v} = (v_1, v_2, \ldots, v_n) \in \mathbb{DH}^n$, with $u_i = u_{i,\mathrm{st}} + u_{i,\mathcal{I}}\epsilon$, $v_i = v_{i,\mathrm{st}} + v_{i,\mathcal{I}}\epsilon$. Denote $\langle \boldsymbol{u}, \boldsymbol{v} \rangle$ as $(\boldsymbol{u}^*\boldsymbol{v} + \boldsymbol{v}^*\boldsymbol{u})/2$. Thus we have

$$\langle \boldsymbol{u}, \boldsymbol{v} \rangle = \sum_{i=1}^{n} \mathrm{sc}(u_{i,\mathrm{st}} v_{i,\mathrm{st}}^*) + \sum_{i=1}^{n} \big(\mathrm{sc}(u_{i,\mathrm{st}} v_{i,\mathcal{I}}^*) + \mathrm{sc}(v_{i,\mathrm{st}} u_{i,\mathcal{I}}^*)\big)\epsilon.$$

Let $\boldsymbol{z} := \Pi(\boldsymbol{x} + \Pi(\boldsymbol{x}) + \boldsymbol{y})$. From the definition of $\Pi(\cdot)$, taking $\Pi(\boldsymbol{x})$ as a feasible point, we have

$$\|\boldsymbol{x} + \Pi(\boldsymbol{x}) + \boldsymbol{y} - \boldsymbol{z}\|_2^2 \leq \|\boldsymbol{x} + \Pi(\boldsymbol{x}) + \boldsymbol{y} - \Pi(\boldsymbol{x})\|_2^2. \tag{29}$$

Using $\|\boldsymbol{u} - \boldsymbol{v}\|_2^2 = \|\boldsymbol{u}\|_2^2 + \|\boldsymbol{v}\|_2^2 - 2\langle \boldsymbol{u}, \boldsymbol{v} \rangle$ for any $\boldsymbol{u}, \boldsymbol{v}$ and $\|\boldsymbol{z}\|_2^2 = \|\Pi(\boldsymbol{x})\|_2^2 = n$ (since $\boldsymbol{z}, \Pi(\boldsymbol{x}) \in \mathrm{UDQ}^n$), inequality (29) simplifies to

$$\langle \boldsymbol{y}, \boldsymbol{z} - \Pi(\boldsymbol{x}) \rangle \geq \langle \boldsymbol{x} + \Pi(\boldsymbol{x}), \Pi(\boldsymbol{x}) - \boldsymbol{z} \rangle. \tag{30}$$

On the other hand, since $\Pi(\boldsymbol{x})$ is the projection of $\boldsymbol{x}$ onto $\mathrm{UDQ}^n$ and $\boldsymbol{z} \in \mathrm{UDQ}^n$,

$$\|\boldsymbol{x} - \Pi(\boldsymbol{x})\|_2^2 \leq \|\boldsymbol{x} - \boldsymbol{z}\|_2^2,$$

which implies

$$\langle \boldsymbol{x}, \Pi(\boldsymbol{x}) - \boldsymbol{z} \rangle \geq 0. \tag{31}$$

Combining (30) and (31), we obtain

$$\langle \boldsymbol{y}, \boldsymbol{z} - \Pi(\boldsymbol{x}) \rangle \geq \langle \boldsymbol{x} + \Pi(\boldsymbol{x}), \Pi(\boldsymbol{x}) - \boldsymbol{z} \rangle = \langle \boldsymbol{x}, \Pi(\boldsymbol{x}) - \boldsymbol{z} \rangle + \langle \Pi(\boldsymbol{x}), \Pi(\boldsymbol{x}) - \boldsymbol{z} \rangle \geq \langle \Pi(\boldsymbol{x}), \Pi(\boldsymbol{x}) - \boldsymbol{z} \rangle.$$

Moreover,

$$\langle \Pi(\boldsymbol{x}), \Pi(\boldsymbol{x}) - \boldsymbol{z} \rangle = \langle \Pi(\boldsymbol{x}), \Pi(\boldsymbol{x}) \rangle - \langle \Pi(\boldsymbol{x}), \boldsymbol{z} \rangle = n - \langle \Pi(\boldsymbol{x}), \boldsymbol{z} \rangle = \frac{1}{2}\|\Pi(\boldsymbol{x}) - \boldsymbol{z}\|_2^2.$$

Hence

$$\langle \boldsymbol{y}, \boldsymbol{z} - \Pi(\boldsymbol{x}) \rangle \geq \frac{1}{2}\|\Pi(\boldsymbol{x}) - \boldsymbol{z}\|_2^2 \geq 0. \tag{32}$$

By the Cauchy–Schwarz inequality (e.g., Proposition 4.7 in Ling et al. (2022)),

$$\langle \boldsymbol{y}, \boldsymbol{z} - \Pi(\boldsymbol{x}) \rangle \leq \|\boldsymbol{y}\|_2 \|\boldsymbol{z} - \Pi(\boldsymbol{x})\|_2.$$

Together with (32), this yields

$$\frac{1}{2}\|\boldsymbol{z} - \Pi(\boldsymbol{x})\|_2^2 \leq \|\boldsymbol{y}\|_2 \|\boldsymbol{z} - \Pi(\boldsymbol{x})\|_2.$$

If $\|\boldsymbol{z} - \Pi(\boldsymbol{x})\|_2 = 0$, the claim is trivial. Otherwise, factoring out the term $\|\boldsymbol{z} - \Pi(\boldsymbol{x})\|_2$, we obtain

$$\|\boldsymbol{z} - \Pi(\boldsymbol{x})\|_2 \cdot \left( \frac{1}{2}\|\boldsymbol{z} - \Pi(\boldsymbol{x})\|_2 - \|\boldsymbol{y}\|_2 \right) \leq 0.$$

Since norms are non-negative, the second factor must be non-positive, i.e.,

$$\|\boldsymbol{z} - \Pi(\boldsymbol{x})\|_2 \leq 2\|\boldsymbol{y}\|_2,$$

which proves the lemma. $\square$

### D.3. Proof of Lemma D.3

By the triangle inequality,

$$\|\Pi(\boldsymbol{p} + \boldsymbol{q}) - \boldsymbol{r}\|_2 \leq \|\Pi(\boldsymbol{p} + \boldsymbol{q}) - \Pi(\boldsymbol{p})\|_2 + \|\Pi(\boldsymbol{p}) - \boldsymbol{r}\|_2.$$

Applying Lemma D.2 with $\boldsymbol{x} = \boldsymbol{p}$ and $\boldsymbol{y} = \boldsymbol{q} - \Pi(\boldsymbol{p})$ yields

$$\|\Pi(\boldsymbol{p} + \boldsymbol{q}) - \Pi(\boldsymbol{p})\|_2 = \|\Pi(\boldsymbol{p} + \Pi(\boldsymbol{p}) + (\boldsymbol{q} - \Pi(\boldsymbol{p}))) - \Pi(\boldsymbol{p})\|_2 \leq 2\|\boldsymbol{q} - \Pi(\boldsymbol{p})\|_2.$$

Finally, using the triangle inequality again,

$$\|\boldsymbol{q} - \Pi(\boldsymbol{p})\|_2 \leq \|\boldsymbol{q} - \boldsymbol{r}\|_2 + \|\Pi(\boldsymbol{p}) - \boldsymbol{r}\|_2.$$

Combining the above three displays gives (20).

### D.4. Proof of Lemma D.4

We first prove the estimate componentwise. Fix $i \in \{1, \ldots, n\}$, and write $r_i = q_i + \epsilon p_i$, $h_i = e_i + \epsilon f_i$, where $q_i, p_i, e_i, f_i \in \mathbb{H}$. Since $r_i \in \text{UDQ}$, we have

$$|q_i| = 1, \qquad \text{sc}(q_i^* p_i) = 0.$$

Let $\Pi(r_i + h_i) = \tilde{q}_i + \epsilon \tilde{p}_i$. The local-chart condition (22) gives

$$|q_i + e_i| \geq \frac{1}{2},$$

in particular $q_i + e_i \neq 0$. By the definition in Proposition 2.7, we therefore have

$$\tilde{q}_i = \frac{q_i + e_i}{|q_i + e_i|}.$$

For any unit quaternion $q$, define the tangent projection

$$P_q(\xi) := \xi - q\,\text{sc}(q^*\xi), \qquad \xi \in \mathbb{H}.$$

Then $P_q$ is the Euclidean orthogonal projection onto

$$T_q \mathbb{S}^3 = \{\xi \in \mathbb{H} : \operatorname{sc}(q^*\xi) = 0\}.$$

In particular, $\|P_q\| \leq 1$ and $P_{q_i}(p_i) = p_i$. Using the explicit form of the dual part of the componentwise normalization, we have

$$\tilde{p}_i = \frac{1}{|q_i + e_i|} P_{\tilde{q}_i}(p_i + f_i).$$

Subtracting $p_i = P_{q_i}(p_i)$, we decompose the difference as

$$\tilde{p}_i - p_i = \frac{1}{|q_i + e_i|} P_{\tilde{q}_i}(f_i) + \left(\frac{1}{|q_i + e_i|} - 1\right) P_{\tilde{q}_i}(p_i) + \left(P_{\tilde{q}_i} - P_{q_i}\right)p_i.$$

We bound these three terms separately.

First, since $|q_i + e_i| \geq 1/2$ and $\|P_{\tilde{q}_i}\| \leq 1$,

$$\left|\frac{1}{|q_i + e_i|} P_{\tilde{q}_i}(f_i)\right| \leq 2|f_i|.$$

Second, again using $|q_i + e_i| \geq 1/2$, together with the reverse triangle inequality, we obtain

$$\left|\frac{1}{|q_i + e_i|} - 1\right| = \frac{\big||q_i + e_i| - 1\big|}{|q_i + e_i|} \leq 2|e_i|,$$

and therefore

$$\left|\left(\frac{1}{|q_i + e_i|} - 1\right) P_{\tilde{q}_i}(p_i)\right| \leq 2|p_i|\,|e_i|.$$

It remains to control the change of tangent projection from $P_{q_i}$ to $P_{\tilde{q}_i}$. For any $\xi \in \mathbb{H}$, we write

$$\left(P_{\tilde{q}_i} - P_{q_i}\right)\xi = q_i \operatorname{sc}(q_i^*\xi) - \tilde{q}_i \operatorname{sc}(\tilde{q}_i^*\xi) = (q_i - \tilde{q}_i)\operatorname{sc}(q_i^*\xi) + \tilde{q}_i \operatorname{sc}\big((q_i - \tilde{q}_i)^*\xi\big).$$

Using

$$|\operatorname{sc}(q_i^*\xi)| \leq |\xi|, \qquad |\operatorname{sc}((q_i - \tilde{q}_i)^*\xi)| \leq |q_i - \tilde{q}_i|\,|\xi|, \qquad |\tilde{q}_i| = 1,$$

we obtain $|(P_{\tilde{q}_i} - P_{q_i})\xi| \leq 2|\tilde{q}_i - q_i|\,|\xi|$, and then

$$\|P_{\tilde{q}_i} - P_{q_i}\| \leq 2|\tilde{q}_i - q_i| \leq 4|e_i|.$$

Consequently,

$$\left|\left(P_{\tilde{q}_i} - P_{q_i}\right)p_i\right| \leq 4|p_i|\,|e_i|.$$

Combining the three estimates above gives the following elementwise bound for the dual part:

$$\left|\big(\Pi(r_i + h_i) - r_i\big)_{\mathcal{I}}\right| = |\tilde{p}_i - p_i| \leq 2|f_i| + 6|p_i|\,|e_i|.$$

We now pass from the elementwise estimates to the vectorized estimates. Since $\Pi$ acts componentwise on $\boldsymbol{x} \in \mathbb{DH}^n$ when $x_{i,\mathrm{st}} \neq 0$, we have

$$\Pi(\boldsymbol{r} + \boldsymbol{h})_i = \Pi(r_i + h_i).$$

For the dual part, we use the elementwise dual estimate and the triangle inequality in $\ell_2$. This yields

$$\left\|\big(\Pi(\boldsymbol{r} + \boldsymbol{h}) - \boldsymbol{r}\big)_{\mathcal{I}}\right\|_2 \leq 2\left(\sum_{i=1}^n |(h_i)_{\mathcal{I}}|^2\right)^{1/2} + 6\left(\sum_{i=1}^n |(r_i)_{\mathcal{I}}|^2 |(h_i)_{\mathrm{st}}|^2\right)^{1/2}$$

$$\leq 2\|\boldsymbol{h}_{\mathcal{I}}\|_2 + 6 \max_{1 \leq i \leq n} |(r_i)_{\mathcal{I}}|\,\|\boldsymbol{h}_{\mathrm{st}}\|_2.$$

This proves (23).

## D.5. Proof of Lemma D.5

Let $s_k := \hat{\boldsymbol{x}}^* \boldsymbol{x}^k \in \mathbb{DH}$ and define $p_k := z_k^* s_k$. Since $|z_k| = 1$, we have

$$\left|\hat{\boldsymbol{x}}^* \boldsymbol{x}^k - n z_k\right| = \left|z_k^* (\hat{\boldsymbol{x}}^* \boldsymbol{x}^k - n z_k)\right| = |p_k - n|.$$

On the other hand,

$$\mathrm{d}(\boldsymbol{x}^k, \hat{\boldsymbol{x}})^2 = \|\boldsymbol{x}^k - \hat{\boldsymbol{x}} z_k\|_2^2 = 2n - (p_k + p_k^*),$$

where we used $\|\boldsymbol{x}^k\|_2^2 = \|\hat{\boldsymbol{x}} z_k\|_2^2 = n$ and the definition of the inner product.

Therefore it suffices to show that $p_k$ is Hermitian and satisfies $0 \preceq p_k \preceq n + 0\epsilon$ (in the total order on dual numbers). In that case,

$$\frac{1}{2}\,\mathrm{d}(\boldsymbol{x}^k, \hat{\boldsymbol{x}})^2 = n - \frac{p_k + p_k^*}{2} = n - p_k = |n - p_k| = |p_k - n|,$$

which yields the desired identity.

It remains to justify that $p_k$ is a dual number (i.e., $p_k = p_k^*$) and $0 \preceq p_k \preceq n + 0\epsilon$. Write $s_k = a_k + b_k \epsilon$ and $z_k^* = c_k + d_k \epsilon$ with $a_k, b_k, c_k, d_k \in \mathbb{H}$. The constraint $z_k \in \mathrm{UDQ}$ implies $c_k^* c_k = 1$ and $d_k c_k^* + c_k d_k^* = 0$. Moreover, $z_k$ is chosen to minimize $\|\boldsymbol{x}^k - \hat{\boldsymbol{x}} z\|_2^2$, which is equivalent to maximizing the dual number $p + p^*$ over $z \in \mathrm{UDQ}$.

*Case 1: $a_k \neq 0$.* Maximizing the standard (rotation) part yields

$$c_k = \frac{a_k^*}{|a_k|}, \qquad \text{so that} \qquad \mathrm{st}(c_k a_k) = |a_k| \in \mathbb{R}_{\geq 0}.$$

With this choice, the UDQ constraint $d_k c_k^* + c_k d_k^* = 0$ is equivalent to

$$d_k a_k + a_k^* d_k^* = 0,$$

i.e., $d_k a_k$ can be any pure quaternion. Moreover,

$$p_k = (c_k + d_k \epsilon)(a_k + b_k \epsilon) = |a_k| + \left(\frac{a_k^* b_k}{|a_k|} + d_k a_k\right)\epsilon.$$

We choose $d_k$ so that the $\epsilon$-coefficient is real, namely,

$$d_k a_k = -\mathrm{im}\left(\frac{a_k^* b_k}{|a_k|}\right),$$

which is feasible since the right-hand side is a pure quaternion (with $\mathrm{im}(\cdot)$ denoting the imaginary vector part). With this choice, the $\epsilon$-coefficient reduces to $\mathrm{sc}\left(\frac{a_k^* b_k}{|a_k|}\right) \in \mathbb{R}$, hence $p_k \in \mathbb{D} = \mathbb{R} + \epsilon\mathbb{R}$ and in particular $p_k = p_k^*$. Furthermore, since $|z_k| = 1$ and by Cauchy–Schwarz,

$$|p_k| = |z_k^* s_k| \leq |s_k| = |\hat{\boldsymbol{x}}^* \boldsymbol{x}^k| \leq \|\hat{\boldsymbol{x}}\|_2 \|\boldsymbol{x}^k\|_2 = n,$$

and $\mathrm{st}(p_k) \geq 0$ by construction. Therefore $0 \preceq p_k \preceq n + 0\epsilon$ under the standard order on dual numbers.

*Case 2: $a_k = 0$.* Then $s_k = b_k \epsilon$. If $b_k = 0$, then $p_k = 0$ for every $z_k \in \mathrm{UDQ}$, and

$$\frac{1}{2}\,\mathrm{d}(\boldsymbol{x}^k, \hat{\boldsymbol{x}})^2 = n = |p_k - n|.$$

Assume now that $b_k \neq 0$. Write $z_k^* = c_k + d_k \epsilon$. Then

$$p_k = z_k^* s_k = c_k b_k \epsilon, \qquad p_k + p_k^* = 2\,\mathrm{sc}(c_k b_k)\epsilon.$$

Since the standard part of the objective is zero for every unit $c_k$, we break ties lexicographically by maximizing $\mathrm{sc}(c_k b_k)$. This is achieved by choosing $c_k = \frac{b_k^*}{|b_k|}$ and $d_k = 0$. Hence $p_k = |b_k|\epsilon \in \mathbb{D}$, so $p_k = p_k^*$ and $0 \preceq p_k \preceq n + 0\epsilon$. Moreover,

$$\frac{1}{2}\,\mathrm{d}(\boldsymbol{x}^k, \hat{\boldsymbol{x}})^2 = n - \frac{p_k + p_k^*}{2} = n - |b_k|\epsilon = |p_k - n|.$$

The proof is complete.

### D.6. Proof of Lemma D.6

By the scale invariance of the componentwise normalization map $\Pi$,

$$\boldsymbol{x}^{k+1} = \Pi(\boldsymbol{C}\boldsymbol{x}^k) = \Pi\left(\frac{1}{n}\boldsymbol{C}\boldsymbol{x}^k\right).$$

Since $\boldsymbol{C} - \boldsymbol{\Delta} = \hat{\boldsymbol{x}}\hat{\boldsymbol{x}}^*$, we write

$$\frac{1}{n}\boldsymbol{C}\boldsymbol{x}^k = \boldsymbol{r}^k + \boldsymbol{h}^k,$$

where

$$\boldsymbol{h}^k := \frac{1}{n}\big((\boldsymbol{C} - \boldsymbol{\Delta})\boldsymbol{x}^k - n\boldsymbol{r}^k\big) + \frac{1}{n}\boldsymbol{\Delta}(\boldsymbol{x}^k - \boldsymbol{r}^k) + \frac{1}{n}\boldsymbol{\Delta}\boldsymbol{r}^k.$$

By Assumption (8), the normalization input satisfies

$$\min_i \big|(\boldsymbol{r}_i^k + \boldsymbol{h}_i^k)_{\mathrm{st}}\big| = \min_i \left|\left(\frac{1}{n}\boldsymbol{C}\boldsymbol{x}^k\right)_{i,\mathrm{st}}\right| \geq \frac{1}{2}.$$

Therefore, applying Lemma D.4 with $\boldsymbol{r} = \boldsymbol{r}^k$ and $\boldsymbol{h} = \boldsymbol{h}^k$, and using Assumption (7), we get

$$\big\|(\boldsymbol{x}^{k+1} - \boldsymbol{r}^k)_{\mathcal{I}}\big\|_2 \leq 2\|(\boldsymbol{h}^k)_{\mathcal{I}}\|_2 + 6B\|(\boldsymbol{h}^k)_{\mathrm{st}}\|_2,$$

and

$$\big\|(\boldsymbol{x}^{k+1} - \boldsymbol{r}^k)_{\mathrm{st}}\big\|_2 \leq 2\|(\boldsymbol{h}^k)_{\mathrm{st}}\|_2.$$

Then the stable gauge-switching condition (9) gives

$$\delta_{\mathcal{I}}^{k+1} = \big\|(\boldsymbol{x}^{k+1} - \hat{\boldsymbol{x}}z_{k+1})_{\mathcal{I}}\big\|_2 \leq \big\|(\boldsymbol{x}^{k+1} - \hat{\boldsymbol{x}}z_k)_{\mathcal{I}}\big\|_2 + \gamma\big\|(\boldsymbol{x}^{k+1} - \hat{\boldsymbol{x}}z_k)_{\mathrm{st}}\big\|_2$$
$$\leq 2\|(\boldsymbol{h}^k)_{\mathcal{I}}\|_2 + (6B + 2\gamma)\|(\boldsymbol{h}^k)_{\mathrm{st}}\|_2.$$

It remains to bound the standard and dual components of $\boldsymbol{h}^k$. Define the scalar dual number $\eta^k := z_k^*\hat{\boldsymbol{x}}^*\boldsymbol{x}^k - n$. Then

$$(\boldsymbol{C} - \boldsymbol{\Delta})\boldsymbol{x}^k - n\boldsymbol{r}^k = \hat{\boldsymbol{x}}\hat{\boldsymbol{x}}^*\boldsymbol{x}^k - n\hat{\boldsymbol{x}}z_k = \boldsymbol{r}^k\eta^k.$$

By Lemma D.5, we have

$$|\eta^k| = \frac{1}{2}\,\mathrm{d}(\boldsymbol{x}^k, \hat{\boldsymbol{x}})^2.$$

Since this is an equality of dual numbers, its standard part gives

$$|(\eta^k)_{\mathrm{st}}| = \frac{1}{2}\,\mathrm{d}_{\mathrm{st}}(\boldsymbol{x}^k, \hat{\boldsymbol{x}})^2.$$

Its infinitesimal component satisfies

$$|(\eta^k)_{\mathcal{I}}| \leq \mathrm{d}_{\mathrm{st}}(\boldsymbol{x}^k, \hat{\boldsymbol{x}})\,|\,\mathrm{d}_{\mathcal{I}}(\boldsymbol{x}^k, \hat{\boldsymbol{x}})|.$$

Moreover, for any $\boldsymbol{u} = \boldsymbol{u}_{\mathrm{st}} + \epsilon\boldsymbol{u}_{\mathcal{I}}$, the infinitesimal coefficient of the dual-valued norm satisfies $|(\|\boldsymbol{u}\|_2)_{\mathcal{I}}| \leq \|\boldsymbol{u}_{\mathcal{I}}\|_2$. Applying this to $\boldsymbol{u} = \boldsymbol{x}^k - \boldsymbol{r}^k$ gives $|\,\mathrm{d}_{\mathcal{I}}(\boldsymbol{x}^k, \hat{\boldsymbol{x}})| \leq \delta_{\mathcal{I}}^k$. Hence

$$|(\eta^k)_{\mathcal{I}}| \leq \mathrm{d}_{\mathrm{st}}(\boldsymbol{x}^k, \hat{\boldsymbol{x}})\delta_{\mathcal{I}}^k. \tag{33}$$

Now, we first bound $\|(\boldsymbol{h}^k)_{\mathrm{st}}\|_2$. From the definition of $\boldsymbol{h}^k$,

$$\|(\boldsymbol{h}^k)_{\mathrm{st}}\|_2 \leq \frac{1}{n}\big\|\big((\boldsymbol{C} - \boldsymbol{\Delta})\boldsymbol{x}^k - n\boldsymbol{r}^k\big)_{\mathrm{st}}\big\|_2 + \frac{1}{n}\big\|\big(\boldsymbol{\Delta}(\boldsymbol{x}^k - \boldsymbol{r}^k)\big)_{\mathrm{st}}\big\|_2 + \frac{1}{n}\big\|(\boldsymbol{\Delta}\boldsymbol{r}^k)_{\mathrm{st}}\big\|_2.$$

For the first term, since $(\boldsymbol{C} - \boldsymbol{\Delta})\boldsymbol{x}^k - n\boldsymbol{r}^k = \boldsymbol{r}^k\eta^k$, we have

$$\big\|\big((\boldsymbol{C} - \boldsymbol{\Delta})\boldsymbol{x}^k - n\boldsymbol{r}^k\big)_{\mathrm{st}}\big\|_2 = \|(\boldsymbol{r}^k)_{\mathrm{st}}\|_2|(\eta^k)_{\mathrm{st}}| = \frac{\sqrt{n}}{2}\,\mathrm{d}_{\mathrm{st}}(\boldsymbol{x}^k, \hat{\boldsymbol{x}})^2.$$

For the second term,

$$\left\|\left(\boldsymbol{\Delta}(\boldsymbol{x}^k - \boldsymbol{r}^k)\right)_{\mathrm{st}}\right\|_2 \le \|\boldsymbol{\Delta}\|_{\mathrm{op,st}} \, \mathrm{d}_{\mathrm{st}}(\boldsymbol{x}^k, \hat{\boldsymbol{x}}).$$

For the third term, using $\boldsymbol{r}^k = \hat{\boldsymbol{x}} z_k$ and the fact that right multiplication by the unit quaternion $(z_k)_{\mathrm{st}}$ preserves the Euclidean norm,

$$\left\|(\boldsymbol{\Delta}\boldsymbol{r}^k)_{\mathrm{st}}\right\|_2 = \left\|(\boldsymbol{\Delta}\hat{\boldsymbol{x}})_{\mathrm{st}}\right\|_2.$$

Therefore,

$$\|(\boldsymbol{h}^k)_{\mathrm{st}}\|_2 \le \frac{1}{2\sqrt{n}} \, \mathrm{d}_{\mathrm{st}}(\boldsymbol{x}^k, \hat{\boldsymbol{x}})^2 + \frac{1}{n}\|\boldsymbol{\Delta}\|_{\mathrm{op,st}} \, \mathrm{d}_{\mathrm{st}}(\boldsymbol{x}^k, \hat{\boldsymbol{x}}) + \frac{1}{n}\left\|(\boldsymbol{\Delta}\hat{\boldsymbol{x}})_{\mathrm{st}}\right\|_2.$$

Next we bound $\|(\boldsymbol{h}^k)_{\mathcal{I}}\|_2$. Again using $(\boldsymbol{C} - \boldsymbol{\Delta})\boldsymbol{x}^k - n\boldsymbol{r}^k = \boldsymbol{r}^k \eta^k$, we obtain

$$\left\|\left((\boldsymbol{C} - \boldsymbol{\Delta})\boldsymbol{x}^k - n\boldsymbol{r}^k\right)_{\mathcal{I}}\right\|_2 = \|(\boldsymbol{r}^k \eta^k)_{\mathcal{I}}\|_2 \le \|(\boldsymbol{r}^k)_{\mathrm{st}}\|_2 |(\eta^k)_{\mathcal{I}}| + \|(\boldsymbol{r}^k)_{\mathcal{I}}\|_2 |(\eta^k)_{\mathrm{st}}|.$$

Since

$$\|(\boldsymbol{r}^k)_{\mathrm{st}}\|_2 = \sqrt{n}, \qquad \|(\boldsymbol{r}^k)_{\mathcal{I}}\|_2 \le B\sqrt{n}, \tag{34}$$

this together with (33) gives

$$\left\|\left((\boldsymbol{C} - \boldsymbol{\Delta})\boldsymbol{x}^k - n\boldsymbol{r}^k\right)_{\mathcal{I}}\right\|_2 \le \sqrt{n} \, \mathrm{d}_{\mathrm{st}}(\boldsymbol{x}^k, \hat{\boldsymbol{x}})\delta_{\mathcal{I}}^k + \frac{B\sqrt{n}}{2} \, \mathrm{d}_{\mathrm{st}}(\boldsymbol{x}^k, \hat{\boldsymbol{x}})^2.$$

For the perturbation term,

$$\begin{aligned}
\left\|\left(\boldsymbol{\Delta}(\boldsymbol{x}^k - \boldsymbol{r}^k)\right)_{\mathcal{I}}\right\|_2 &\le \|\boldsymbol{\Delta}\|_{\mathrm{op,st}} \left\|(\boldsymbol{x}^k - \boldsymbol{r}^k)_{\mathcal{I}}\right\|_2 + \|\boldsymbol{\Delta}\|_{\mathrm{op,\mathcal{I}}} \left\|(\boldsymbol{x}^k - \boldsymbol{r}^k)_{\mathrm{st}}\right\|_2 \\
&= \|\boldsymbol{\Delta}\|_{\mathrm{op,st}}\delta_{\mathcal{I}}^k + \|\boldsymbol{\Delta}\|_{\mathrm{op,\mathcal{I}}} \, \mathrm{d}_{\mathrm{st}}(\boldsymbol{x}^k, \hat{\boldsymbol{x}}).
\end{aligned}$$

Finally, from (34) we get

$$\left\|(\boldsymbol{\Delta}\boldsymbol{r}^k)_{\mathcal{I}}\right\|_2 \le \sqrt{n}\left(\|\boldsymbol{\Delta}\|_{\mathrm{op,\mathcal{I}}} + B\|\boldsymbol{\Delta}\|_{\mathrm{op,st}}\right).$$

Combining the three estimates yields

$$\begin{aligned}
\|(\boldsymbol{h}^k)_{\mathcal{I}}\|_2 \le{}& \frac{1}{\sqrt{n}} \, \mathrm{d}_{\mathrm{st}}(\boldsymbol{x}^k, \hat{\boldsymbol{x}})\delta_{\mathcal{I}}^k + \frac{B}{2\sqrt{n}} \, \mathrm{d}_{\mathrm{st}}(\boldsymbol{x}^k, \hat{\boldsymbol{x}})^2 + \frac{1}{n}\|\boldsymbol{\Delta}\|_{\mathrm{op,st}}\delta_{\mathcal{I}}^k \\
&+ \frac{1}{n}\|\boldsymbol{\Delta}\|_{\mathrm{op,\mathcal{I}}} \, \mathrm{d}_{\mathrm{st}}(\boldsymbol{x}^k, \hat{\boldsymbol{x}}) + \frac{1}{\sqrt{n}}\left(\|\boldsymbol{\Delta}\|_{\mathrm{op,\mathcal{I}}} + B\|\boldsymbol{\Delta}\|_{\mathrm{op,st}}\right).
\end{aligned}$$

Substituting the bounds for $\|(\boldsymbol{h}^k)_{\mathrm{st}}\|_2$ and $\|(\boldsymbol{h}^k)_{\mathcal{I}}\|_2$ into $\delta_{\mathcal{I}}^{k+1} \le 2\|(\boldsymbol{h}^k)_{\mathcal{I}}\|_2 + (6B + 2\gamma)\|(\boldsymbol{h}^k)_{\mathrm{st}}\|_2$ gives

$$\begin{aligned}
\delta_{\mathcal{I}}^{k+1} \le{}& \left(\frac{2}{\sqrt{n}} \, \mathrm{d}_{\mathrm{st}}(\boldsymbol{x}^k, \hat{\boldsymbol{x}}) + \frac{2}{n}\|\boldsymbol{\Delta}\|_{\mathrm{op,st}}\right)\delta_{\mathcal{I}}^k \\
&+ \left(\frac{2}{n}\|\boldsymbol{\Delta}\|_{\mathrm{op,\mathcal{I}}} + \frac{6B + 2\gamma}{n}\|\boldsymbol{\Delta}\|_{\mathrm{op,st}} + \frac{4B + \gamma}{\sqrt{n}} \, \mathrm{d}_{\mathrm{st}}(\boldsymbol{x}^k, \hat{\boldsymbol{x}})\right)\mathrm{d}_{\mathrm{st}}(\boldsymbol{x}^k, \hat{\boldsymbol{x}}) \\
&+ \frac{2}{\sqrt{n}}\left(\|\boldsymbol{\Delta}\|_{\mathrm{op,\mathcal{I}}} + B\|\boldsymbol{\Delta}\|_{\mathrm{op,st}}\right) + \frac{6B + 2\gamma}{n}\left\|(\boldsymbol{\Delta}\hat{\boldsymbol{x}})_{\mathrm{st}}\right\|_2.
\end{aligned}$$

This is exactly (25).

### D.7. Proof of Corollary 3.4

Denote the sequence generated by Algorithm 1 as $\{\boldsymbol{w}^k\}_{k \ge 0}$. Write $\boldsymbol{w}^0$ as $\boldsymbol{w}^0 = \sum_{i=1}^n \boldsymbol{v}_j \alpha_j$, where $\boldsymbol{v}_1, \ldots, \boldsymbol{v}_n$ are orthonormal eigenvectors of $C$ corresponding to $\lambda_1, \ldots, \lambda_n$ respectively. Suppose $\lambda_{1,\mathrm{st}} > 0$ and $|\alpha_{1,\mathrm{st}}| \ne 0$. Since Algorithm 1 is a (normalized) power iteration, we can write

$$\boldsymbol{w}^k = \sum_{j=1}^n \boldsymbol{v}_j \, t_j^k, \qquad t_j^k := \lambda_j^k \alpha_j \left(\sqrt{\sum_{\ell=1}^n |\lambda_\ell^k \alpha_\ell|^2}\right)^{-1}.$$

As $k \to \infty$, using $|\lambda_{1,\mathrm{st}}| > |\lambda_{2,\mathrm{st}}|$ and standard dual-number expansion,

$$\sqrt{\sum_{j=1}^{n} |\lambda_j^k \alpha_j|^2} - |\lambda_1|^k |\alpha_1|$$

$$= \left( \sqrt{\sum_{j=1}^{n} |\lambda_j^k \alpha_j|^2} - |\lambda_1|^k |\alpha_1| \right) \left( \sqrt{\sum_{j=1}^{n} |\lambda_j^k \alpha_j|^2} + |\lambda_1|^k |\alpha_1| \right) \left( \sqrt{\sum_{j=1}^{n} |\lambda_j^k \alpha_j|^2} + |\lambda_1|^k |\alpha_1| \right)^{-1}$$

$$= \sum_{j=2}^{n} |\lambda_j^k \alpha_j|^2 \left( \sqrt{\sum_{j=1}^{n} |\lambda_j^k \alpha_j|^2} + |\lambda_1|^k |\alpha_1| \right)^{-1}$$

$$= \sum_{j=2}^{n} |\lambda_{j,\mathrm{st}}|^{2k} (1 + 2k \lambda_{j,\mathrm{st}}^{-1} \lambda_{j,\mathcal{I}} \epsilon) |\alpha_j|^2 A^{-1}$$

$$= O_D \left( \left( \frac{\lambda_{2,\mathrm{st}}^2}{\lambda_{1,\mathrm{st}}} \right)^k \right),$$

where

$$A := \left( \sum_{j=1}^{n} |\lambda_{j,\mathrm{st}}|^{2k} (1 + 2k \lambda_{j,\mathrm{st}}^{-1} \lambda_{j,\mathcal{I}} \epsilon) |\alpha_j|^2 \right)^{1/2} + |\lambda_{1,\mathrm{st}}|^k (1 + k \lambda_{1,\mathrm{st}}^{-1} \lambda_{1,\mathcal{I}} \epsilon) |\alpha_1|.$$

This implies

$$t_j^k = \lambda_j^k \alpha_j \left[ |\lambda_1|^k |\alpha_1| + O_D \big( (|\lambda_{2,\mathrm{st}}|^2 / |\lambda_{1,\mathrm{st}}|)^k \big) \right]^{-1}.$$

For $j = 1$, since $\lambda_1 > 0$ we have $\lambda_1^k / |\lambda_1|^k = 1$, then

$$t_1^k - \lambda_1^k \alpha_1 \left( |\lambda_1|^k |\alpha_1| \right)^{-1} = \lambda_1^k \alpha_1 \left[ |\lambda_1|^k |\alpha_1| + O_D \big( (|\lambda_{2,\mathrm{st}}|^2 / |\lambda_{1,\mathrm{st}}|)^k \big) \right]^{-1} - \lambda_1^k \alpha_1 \left( |\lambda_1|^k |\alpha_1| \right)^{-1}$$

$$= -\lambda_1^k \alpha_1 \left( |\lambda_1|^k |\alpha_1| \right)^{-1} \cdot O_D \big( (|\lambda_{2,\mathrm{st}}|^2 / |\lambda_{1,\mathrm{st}}|)^k \big) \left[ |\lambda_1|^k |\alpha_1| + O_D \big( (|\lambda_{2,\mathrm{st}}|^2 / |\lambda_{1,\mathrm{st}}|)^k \big) \right]^{-1}$$

$$= -\alpha_1 |\alpha_1|^{-1} \cdot O_D \big( (|\lambda_{2,\mathrm{st}}|^2 / |\lambda_{1,\mathrm{st}}|)^k \big) \left[ |\lambda_1|^k |\alpha_1| + O_D \big( (|\lambda_{2,\mathrm{st}}|^2 / |\lambda_{1,\mathrm{st}}|)^k \big) \right]^{-1}.$$

Thus, we have

$$\boldsymbol{w}^k = \boldsymbol{v}_1 \alpha_1 |\alpha_1|^{-1} \left\{ 1 - O_D \big( (|\lambda_{2,\mathrm{st}}|^2 / |\lambda_{1,\mathrm{st}}|)^k \big) \left[ |\lambda_1|^k |\alpha_1| + O_D \big( (|\lambda_{2,\mathrm{st}}|^2 / |\lambda_{1,\mathrm{st}}|)^k \big) \right]^{-1} \right\}$$

$$+ \sum_{j=2}^{n} \boldsymbol{v}_j \lambda_j^k \alpha_j \left( |\lambda_1|^k |\alpha_1| + O_D \big( (|\lambda_{2,\mathrm{st}}|^2 / |\lambda_{1,\mathrm{st}}|)^k \big) \right)^{-1}.$$

Using the distance definition and the feasible choice $z = \alpha_1 |\alpha_1|^{-1} \in \mathrm{UDQ}$,

$$\mathrm{d}(\boldsymbol{w}^k, \boldsymbol{v}_1) = \min_{z \in \mathrm{UDQ}} \|\boldsymbol{w}^k - \boldsymbol{v}_1 z\|_2 \leq \left\| \boldsymbol{w}^k - \boldsymbol{v}_1 \alpha_1 |\alpha_1|^{-1} \right\|_2$$

$$\leq \|\boldsymbol{v}_1\|_2 O_D \big( (|\lambda_{2,\mathrm{st}}|^2 / |\lambda_{1,\mathrm{st}}|)^k \big) \left[ |\lambda_1|^k |\alpha_1| + O_D \big( (|\lambda_{2,\mathrm{st}}|^2 / |\lambda_{1,\mathrm{st}}|)^k \big) \right]^{-1}$$

$$+ \sum_{j=2}^{n} \|\boldsymbol{v}_j\|_2 |\lambda_j|^k |\alpha_j| \left[ |\lambda_1|^k |\alpha_1| + O_D \big( (|\lambda_{2,\mathrm{st}}|^2 / |\lambda_{1,\mathrm{st}}|)^k \big) \right]^{-1} \tag{35}$$

$$= \left[ \sum_{j=2}^{n} |\lambda_j|^k |\alpha_j| + O_D \big( (|\lambda_{2,\mathrm{st}}|^2 / |\lambda_{1,\mathrm{st}}|)^k \big) \right] \left[ |\lambda_1|^k |\alpha_1| + O_D \big( (|\lambda_{2,\mathrm{st}}|^2 / |\lambda_{1,\mathrm{st}}|)^k \big) \right]^{-1}.$$

We use $\Pi(\sqrt{n}\boldsymbol{w}^k)$ as an initializer for Algorithm 2. As in the proof of Theorem 3.2, it suffices to enforce

$$\mathrm{d}_{\mathrm{st}}(\boldsymbol{w}^k, \boldsymbol{v}_1) \leq \frac{1}{70} \tag{36}$$

to guarantee that $\Pi(\sqrt{n}\boldsymbol{w}^k)$ enters the basin of attraction with

$$\mathrm{d}_{\mathrm{st}}(\Pi(\sqrt{n}\boldsymbol{w}^k), \hat{\boldsymbol{x}}) \leq 2\,\mathrm{d}_{\mathrm{st}}(\sqrt{n}\boldsymbol{w}^k, \sqrt{n}\boldsymbol{v}_1) + \mathrm{d}_{\mathrm{st}}(\sqrt{n}\boldsymbol{v}_1, \hat{\boldsymbol{x}}) \leq \frac{\sqrt{n}}{25}.$$

By the dual quaternion calculation rules, for all sufficiently large $k$, there exists $M_{\mathrm{st}} > 0$ depending on $\{\alpha_j, \lambda_j\}_{j=1}^n$, such that for all sufficiently large $k$

$$\mathrm{d}_{\mathrm{st}}(\boldsymbol{w}^k, \boldsymbol{v}_1) \leq \frac{|\lambda_{2,\mathrm{st}}|^k|\alpha_{2,\mathrm{st}}| + M_{\mathrm{st}}|\lambda_{2,\mathrm{st}}|^k}{|\lambda_{1,\mathrm{st}}|^k|\alpha_{1,\mathrm{st}}| + M_{\mathrm{st}}|\lambda_{2,\mathrm{st}}|^k}. \tag{37}$$

Let $r := |\lambda_{1,\mathrm{st}}/\lambda_{2,\mathrm{st}}| > 1$. Solving (37) under (36) yields

$$K_{\mathrm{init}} \geq \log_r\left(\frac{70|\alpha_{2,\mathrm{st}}| + 69M_{\mathrm{st}}}{|\alpha_{1,\mathrm{st}}|}\right).$$

With $K_{\mathrm{init}}$ iterations, the initial point $\boldsymbol{x}^0$ satisfies the basin of attraction conditions:

$$\mathrm{d}_{\mathrm{st}}(\boldsymbol{x}^0, \hat{\boldsymbol{x}}) \leq \frac{\sqrt{n}}{25}.$$

Substituting these bounds into the estimation error recursive formulas derived in Theorem 3.2 (specifically Eq. (6) directly yields the claimed error bounds for $\boldsymbol{x}^k$. This completes the proof.

## E. Remarks on Assumptions in Theorem 3.2

Condition (7) is a bounded aligned translation assumption. Indeed, under the unit-dual-quaternion representation

$$r_i^k = \left(1 + \frac{\epsilon}{2}\hat{t}_i^k\right)\hat{q}_i^k,$$

we have

$$|(r_i^k)_{\mathcal{I}}| = \frac{1}{2}\|\hat{t}_i^k\|_2.$$

Then (7) is equivalent to

$$\max_{k,i}\|\hat{t}_i^k\|_2 \leq 2B.$$

Thus, $B$ controls the radius of the ground truth translations after the global alignment $z_k$.

Condition (8) is a local nondegeneracy condition for the normalization map. The dual part of the normalization $\mathcal{N}(\boldsymbol{y})$ contains a factor $|\boldsymbol{y}_{\mathrm{st}}|^{-1}$. Hence the normalization is uniformly Lipschitz in its dual component only when $|\boldsymbol{y}_{\mathrm{st}}|$ is bounded away from zero. In the DQGPM step, the normalization input is $\frac{1}{n}\boldsymbol{C}\boldsymbol{x}^k$. In the noiseless case and at the aligned ground truth, namely when $\boldsymbol{C} = \hat{\boldsymbol{x}}\hat{\boldsymbol{x}}^*$ and $\boldsymbol{x}^k = \hat{\boldsymbol{x}}z_k$, we have

$$\frac{1}{n}\boldsymbol{C}\boldsymbol{x}^k = \frac{1}{n}\hat{\boldsymbol{x}}\hat{\boldsymbol{x}}^*\hat{\boldsymbol{x}}z_k = \hat{\boldsymbol{x}}z_k = \boldsymbol{x}^k,$$

whose standard component has unit norm entrywise. Therefore the threshold $\frac{1}{2}$ simply requires the iterates to remain in a neighborhood where the normalization map is nonsingular.

Condition (9) controls the effect of changing the lexicographic aligner from $z_k$ to $z_{k+1}$. Since the aligner is chosen according to the lexicographic dual-number distance, a change of alignment may transfer standard mismatch into the dual component. The constant $\gamma$ quantifies this leakage. It does not need to be small for the contraction argument; it only appears in the additive forcing terms of the dual-component recursion.

## F. Additional Experiments on Heavy-tailed Perturbations and Scalability

We further test robustness under heavy-tailed perturbations by replacing the Gaussian model above with a Gaussian-mixture noise model on the observed edges. Specifically, with probability $1 - \eta$, an observed relative motion is corrupted by the base perturbation level $(\sigma_t, \sigma_r) = (0.1, 10°)$, while with probability $\eta$, it is corrupted by a larger perturbation $(\sigma_t, \sigma_r) = (0.4, 40°)$. $\text{error}_r$ and $\text{error}_t$ are reported in Figure 6 with varying outlier proportion $\eta$ under $p \in \{0.05, 0.3\}$. We also fix $\eta = 0.2$ and vary the outlier perturbation level by changing $\sigma_r$, with $\sigma_t = 0.01\sigma_r$ for each level, as shown in Figure 7. In all settings, DQGPM remains the most accurate method among DQGPM, SPEC, and EIG. These results indicate that DQGPM is robust to heavy-tailed perturbations; handling completely arbitrary gross outliers would require an additional robust reweighting or trimming layer. Finally, to examine scalability, we fix $p = 0.1$ and the base noise level $(\sigma_t, \sigma_r) = (0.1, 10°)$, and vary the problem size $n$. The CPU times in Figure 8 show that DQGPM scales favorably and remains substantially faster than SPEC, while being competitive with EIG.

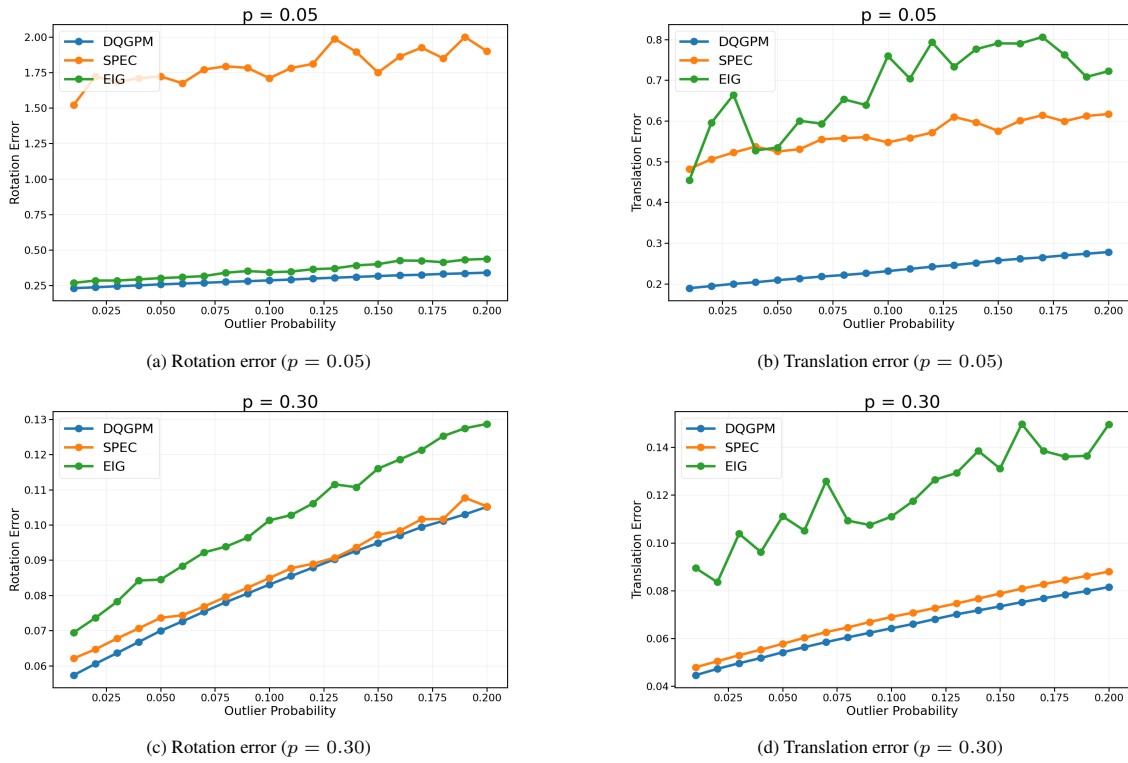

*Figure 6.* Rotation and translation errors vs. outlier proportion under Gaussian-mixture heavy-tail noise, with base perturbation level $(\sigma_t, \sigma_r) = (0.1, 10°)$ and outlier perturbation level $(\sigma_t, \sigma_r) = (0.4, 40°)$.

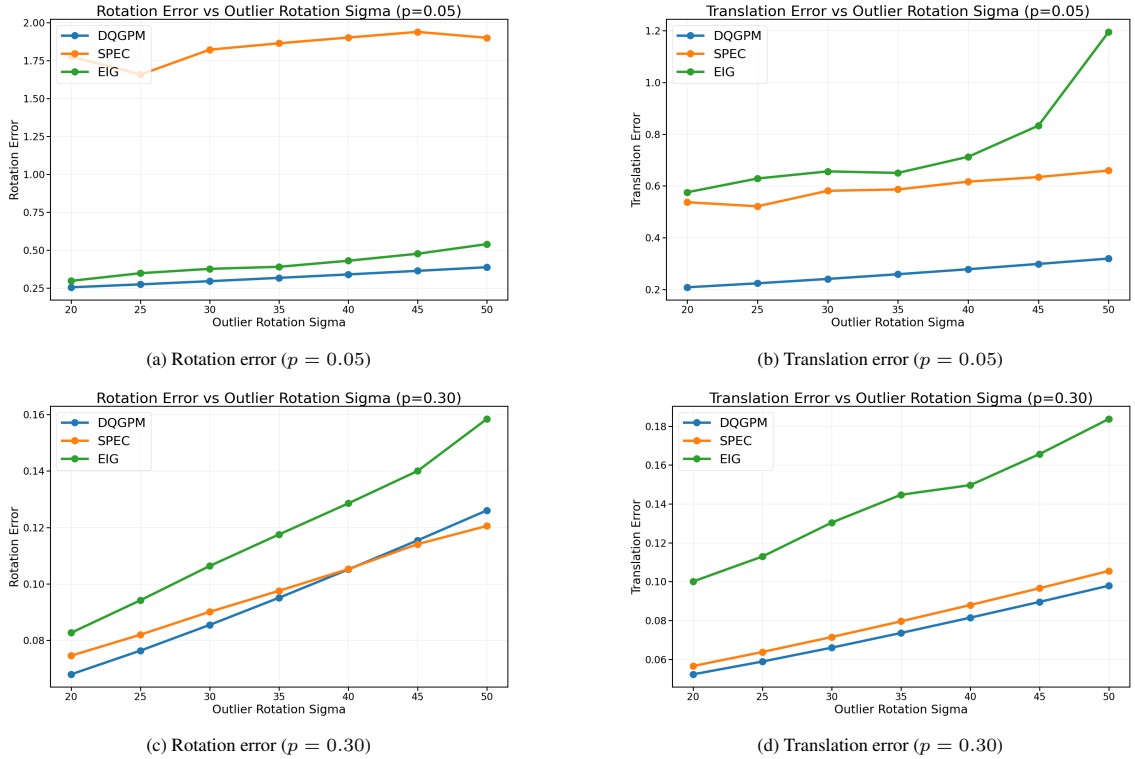

*Figure 7.* Rotation and translation errors vs. outlier $\sigma_r$ under Gaussian-mixture heavy-tail noise, with $\sigma_t = 0.01\,\sigma_r$ for each outlier perturbation, base perturbation level $(\sigma_t, \sigma_r) = (0.1, 10°)$, and outlier proportion 0.2.

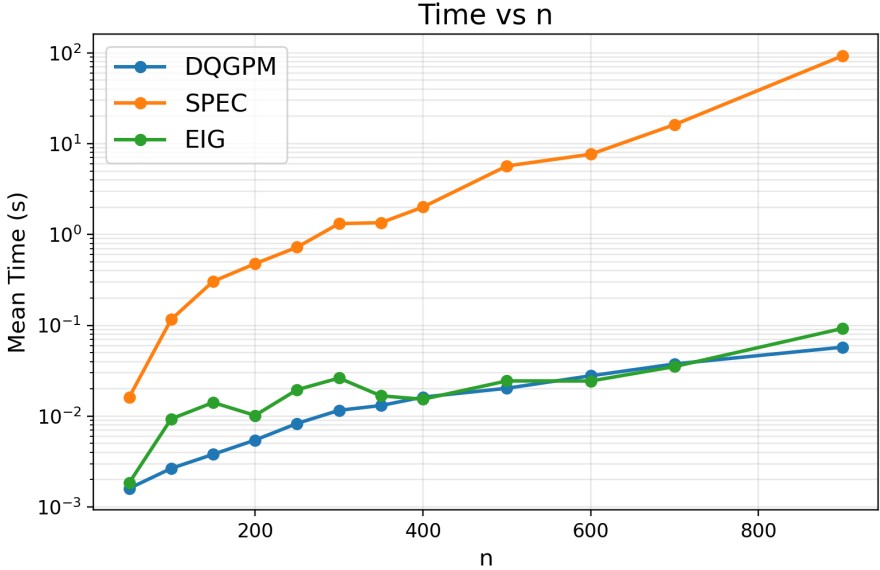

*Figure 8.* CPU time versus problem scale $n$ under observation rate $p = 0.1$ and base noise level $(\sigma_t, \sigma_r) = (0.1, 10°)$.

