# OpenReview forum: "Dual Quaternion SE(3) Synchronization with Recovery Guarantees"
_ICML.cc/2026/Conference — ICML 2026 regular_

### Official Review · Reviewer_3vXk · 2026-03-09

**Soundness:** 3
**Presentation:** 2
**Significance:** 3
**Originality:** 3
**Overall Recommendation:** 4
**Confidence:** 3

**Summary:**

The author formulates the SE(3) synchronization problem over the multi-dimensional unit dual quaternion sets UDQ^n and proposes a dual-quaternion generalized power method (DQGPM) for synchronization. DQGPM estimates the largest eigenvalue of a Hermitian dual-quaternion matrix via a power iteration method while enforcing feasibility constraints, thereby obtaining a solution to the synchronization problem in a principled manner. The author also provides a principled error-bound analysis of DQGPM and its variants. Experiments are conducted to demonstrate its computational cost and accuracy.

**Compliance With Llm Reviewing Policy:**

Affirmed.

**Final Justification:**

The rebuttal fully addressed my concerns.

**Key Questions For Authors:**

1. In Theorem 3.2, aren’t the assumptions on the measurement noise overly restrictive? Does DQGPM fail in practice under more realistic noise conditions?

**Limitations:**

1. **(Weakness #1)** The paper may be difficult for some readers to follow. In particular, many mathematical terms are introduced abruptly in the introduction. For example: what exactly are representation gaps, the isometry group, intrinsic global gauge symmetry, and the eigenspace-sync gap? What exactly does the rounding or projection step mean? What is projected onto which space?
For clarity, the introduction could be separated into an introduction section and a related work section. Alternatively, some of the mathematical concepts could be explained in the appendix.
2. **(Weakness #1)** A conclusion or discussion section should be included at the end of the main paper.
3. **(Weakness #2)** It would be beneficial to include an analysis of computational speed with respect to the number of dimensions n in order to better validate the computational efficiency of DQGPM.
4. **(Weakness #2)** The baselines used in the numerical analysis could be outdated. The most recent baseline used in the paper was proposed in 2019. More recent works such as [1] or the cited work [2] may serve as stronger baselines.
5. **(Weakness #2)** The experimental results in Table 4 appear to be obtained under a very restrictive setting. The data are perturbed by Gaussian noise with very small variance (0.001 m and 1 degree).
6. **(Weakness #2)** The proposed method could potentially be applied to various domains such as SLAM or urban 3D point cloud registration. The paper would better demonstrate its practical impact if additional application examples were included.

* **References**

[1] Y. Heng, J. Shi, and L. Carlone. "Teaser: Fast and certifiable point cloud registration." IEEE Transactions on Robotics (TRO) 2020

[2] C. Chunfeng, and L. Qi. "A power method for computing the dominant eigenvalue of a dual quaternion Hermitian matrix." Journal of Scientific Computing 2024

**Strengths And Weaknesses:**

* **Strengths**
1. A principled approach without heuristics is proposed to solve the SE(3) synchronization problem while ensuring feasibility.
2. The authors provide theoretical error-bound analysis of power iteration methods over multi-dimensional unit dual quaternion sets, including DQGPM.

* **Weaknesses**
1. Considering the target audience of a machine learning conference, the paper is written in a way that is difficult to understand. In particular, the introduction contains too many mathematical concepts without sufficient explanation. Moreover, a conclusion or discussion section should be included.
2. The experiments are not sufficiently well designed. Additional analyses, more recent baselines, and clearer experimental details are required.

---

> ### Author Rebuttal · Authors · 2026-03-31
>
> Thanks for your comments. We hope that the following discussion can clear up your concerns.
>
> **Q1. In Theorem 3.2, aren’t the assumptions on the measurement noise overly restrictive? Does DQGPM fail in practice under more realistic noise conditions?**
>
> Although the noise conditions in Theorem 3.2 appear restrictive, they actually reflect the fundamental information-theoretic limits (with respect to $n$) for establishing exact recovery guarantees in SE(3) synchronization. Empirically, DQGPM remains effective well beyond the theorem's regime, maintaining robust performance and the predicted linear convergence to a noise floor even under severe perturbations up to $(\sigma_t,\sigma_r)=(0.20, 20^\circ)$ on extremely sparse graphs. We have also added experiments under heavy-tailed noise, please see our response to Reviewer few9 Q1.
>
> **Q2. The paper may be difficult for some readers to follow. In particular, many mathematical terms are introduced abruptly in the introduction. A conclusion or discussion section should be included at the end of the main paper.**
>
> We thank the reviewer for the constructive feedback on presentation. In the revision, we will add brief, intuitive explanations of these key concepts in the introduction and include a dedicated conclusion section. Indeed, the mentioned terminologies directly highlights the core motivation of our paper: standard matrix relaxations embed 6-DoF rigid motions into higher-dimensional spaces, creating "representation gaps" and introducing artificial mathematical "isometries" (symmetries) not present in the physical 3D world. Within these gaps, solvers can lower the objective via non-rigid deformations, leading to the "eigenspace-sync gap" where relaxed solutions fail to yield valid rigid motions, particularly under sparsity. Consequently, matrix methods require costly "rounding or projection" steps with heuristic algorithms that force these distorted matrix outputs back onto the valid SE(3) manifold. In contrast, our formulation natively mirrors the intrinsic physical symmetries (global gauge symmetry) of 3D space, preventing these artificial gaps and reducing the projection onto the $\mathrm{UDQ}$ manifold to a simple normalization.
>
> **Q3. It would be beneficial to include an analysis of computational speed with respect to the number of dimensions n in order to better validate the computational efficiency of DQGPM.**
>
> Please see **Figure 5 in [PDF](https://anonymous.4open.science/r/dq_supplement_pdf-C728/supple_pdf.pdf)**.
>
> **Q4. The baselines used in the numerical analysis could be outdated. The most recent baseline used in the paper was proposed in 2019. More recent works such as [1] or the cited work [2] may serve as stronger baselines.**
>
> We thank the reviewer for suggesting these recent works. However, the suggested methods address different scopes: [1] focuses on pairwise point-set registration (generating relative measurements) rather than the global SE(3) synchronization problem considered here. Meanwhile, [2] provides a numerical solver for the DQ eigenproblem rather than a complete synchronization pipeline; in fact, we already utilize this exact solver from [2] for our Stage 1 spectral initialization. To strengthen our empirical evaluation, we have incorporated a highly relevant recent baseline "SPEC" by Doherty et al. (2022) across both synthetic and real-world experiments (**please see the attached [PDF](https://anonymous.4open.science/r/dq_supplement_pdf-C728/supple_pdf.pdf)**). More broadly, recent baselines that are directly comparable for global synchronization remain limited, as most recent work has focused on pairwise registration rather than the global problem.
>
> **Q5. The experimental results in Table 4 appear to be obtained under a very restrictive setting. The data are perturbed by Gaussian noise with very small variance (0.001 m and 1 degree).**
>
> We thank the reviewer for raising this point, as it allows us to clarify the effective noise scale in this dataset. While the nominal perturbation values (0.001 m and 1 degree) may appear small in absolute terms, they must be contextualized against the dataset's ground-truth translation magnitudes, which are on the order of $10^{-2}$ to $10^{-3}$ meters. Consequently, the relative perturbation or the SNR is actually quite significant and challenging for the solvers. Nevertheless, we have conducted an entirely new set of experiments applying substantially stronger perturbations (0.1 m and 10 degrees). Furthermore, we have included the recent baseline ``SPEC'' by Doherty et al. (2022) in these new trials. As shown in the attached **[PDF](https://anonymous.4open.science/r/dq_supplement_pdf-C728/supple_pdf.pdf) (see Table 3)**, our method still holds an advantage under these challenging conditions.
>
> **Q6. The proposed method could potentially be applied to various domains such as SLAM or urban 3D point cloud registration.**
>
> Please see the answer to Reviewer few9 Q3.

---

> > ### Author Rebuttal · Reviewer_3vXk · 2026-04-02
> >
> > Thank you for resolving most of my concerns. I believe the authors will kindly introduce the mathematical concepts used in the paper that were not previously explained.
> >
> > However, one concern remains regarding Q6. Although I appreciate that the authors compare DQGPM with other baselines on the 3DMatch dataset (Figures 1 and 2), it would be beneficial to also report the numerical results for the 3DMatch experiments in a table.
> > _____
> >
> > **Response to Reply Rebuttal Comment.** Thank you for providing the numerical results of the 3DMatch experiments. My concerns are fully addressed so that I will raise the score.

---

> > > ### Author Response · Authors · 2026-04-05
> > >
> > > We sincerely thank the reviewer for the careful reading of our response and the constructive feedback.
> > >
> > > Regarding Q6, please refer to the below table, which reports the numerical results of the 3DMatch experiments. As shown, our method consistently outperforms the baselines across all metrics.
> > >
> > > | | | Office1 | Office2 | Livingroom1 | Livingroom2 | Redkitchen | Hotel3 |
> > > |:--|:--|:--:|:--:|:--:|:--:|:--:|:--:|
> > > | NO_EDGES | | 53 | 50 | 57 | 47 | 60 | 37 |
> > > | MISSING | | 81.69% | 87.36% | 78.12% | 84.11% | 70.22% | 89.41% |
> > > | ERROR_R | DQGPM | **0.1209** | **0.0066** | **7.78e-7** | **9.00e-6** | **2.89e-5** | **3.96e-7** |
> > > | | SPEC | 1.5195 | 1.7705 | 1.3342 | 2.0904 | 1.0184 | 1.2469 |
> > > | | SDR | 1.5038 | 1.7616 | 1.3195 | 2.0946 | 1.0082 | 1.2460 |
> > > | | EIG | 1.5093 | 1.7661 | 1.3238 | 2.0951 | 1.0128 | 1.2469 |
> > > | ERROR_T | DQGPM | **0.2454** | **0.0155** | **3.12e-6** | **1.06e-5** | **4.71e-5** | **3.87e-7** |
> > > | | SPEC | 1.8254 | 1.5662 | 1.7862 | 1.6364 | 0.8466 | 0.3994 |
> > > | | SDR | 2.4180 | 2.0358 | 2.2108 | 1.7699 | 1.1597 | 0.4336 |
> > > | | EIG | 1.8860 | 1.6429 | 1.7988 | 1.8343 | 0.8658 | 0.4029 |
> > >
> > > **Evaluation Metrics:** For each 3DMatch scene, let $G=(V,E)$ denote the graph of point-cloud fragments, where each observed edge $(i,j)\in E$ is associated with a ground-truth relative transformation $\hat{X}\_{ij}\in SE(3)$. Given the estimated absolute poses $\lbrace X_i\rbrace\_{i=1}^n$, we reconstruct the relative motion on each observed edge as $X_{ij}=X_iX_j^{-1}$ (equivalently $x_ix_j^*$ in the dual quaternion representation).
> > > For each observed edge, we compare the reconstructed relative motion $X_{ij}$ with the reference motion $\hat{X}\_{ij}$, and calculate the rotation and translation error with following formula, which is also mentioned in our paper:
> > > $$d\_R(q\_1,q\_2)=2\arccos (2\langle q\_1,q\_2\rangle^2-1),\ d\_T(t\_1,t\_2)=\Vert t\_1-t\_2\Vert\_2,$$
> > > where $q\in \mathbb{Q},t\in\mathbb{R}^3$ are the rotation and translation extracted from the SE(3) elements $X\_{ij}$ and $\hat{X}\_{ij}$.
> > > The reported numerical results in above table are the averages of over all observed edges in the scene.

---

### Official Review · Reviewer_PMXV · 2026-03-10

**Soundness:** 2
**Presentation:** 2
**Significance:** 2
**Originality:** 4
**Overall Recommendation:** 4
**Confidence:** 4

**Summary:**

The paper proposes a novel approach to the SE(3) synchronization problem. Standard matrix-based methods often suffer from an "eigenspace-synchronization gap" because the relaxed eigenspace does not naturally map to the non-compact SE(3) manifold, requiring complex heuristic rounding. To solve this, the authors formulate the problem directly over unit dual quaternions. The algorithm has two stages: 1) Computes the dominant eigenvector of a Hermitian dual quaternion measurement matrix using a power method. 2) Refines the initializer by applying a per-iteration elementwise projection onto the unit dual quaternion space to strictly enforce feasibility.The paper's main contributions are largely theoretical, providing finite-iteration error bounds and proving linear error contraction up to a noise-dependent threshold.

**Compliance With Llm Reviewing Policy:**

Affirmed.

**Final Justification:**

Responses from authors have addressed most of my concerns, I raise my score to weak accept.

**Key Questions For Authors:**

- What is the fundamental advantage of optimizing rotation and translation jointly within the dual quaternion framework, as opposed to the standard practice of performing rotation averaging first and substituting the result to solve for translation?
- A footnote in Table 2 mentions SDR was excluded from synthetic experiments due to "limited scalability under uniform edge dropout". Could you elaborate on the specific computational bottlenecks encountered, and why DQGPM does not suffer from them?

**Limitations:**

yes

**Strengths And Weaknesses:**

Strengths

- The paper provides a highly complete theoretical framework. It establishes non-asymptotic estimation error guarantees for the spectral estimator and proves that the proposed DQGPM achieves linear error contraction up to a specific noise floor. For global solvers, these rigorous recovery guarantees are highly valuable.
- Experimental results demonstrate that the proposed DQGPM is particularly robust to high noise levels, especially when the observation graph is highly sparse (e.g., $p=0.05$).

Weaknesses

- While the authors effectively argue the theoretical benefit of closing the "Eigenspace-Sync Gap" using dual quaternions, they fail to cleanly translate this into practical algorithmic advantages. It remains ambiguous whether the primary contribution is faster convergence, superior scalability, or solely robustness to extreme noise.
- For a problem central to large-scale 3D reconstruction and SLAM pipelines, testing solely on small-scale object scans (Bunny, Buddha, Dragon, Armadillo) is inadequate. The experiments lack the large-scale trajectory sequences or complex environments necessary to truly validate the algorithm's scalability and real-world utility.
- The real-world experiments lack soundness. The authors explicitly inject artificial noise into the ICP-derived relative poses. This invalidates the claim that this reflects a real-world multi-scan registration scenario, where noise distributions are heavily structured.
- Because the authors engineered the real-world data to have high noise, baselines like SDR and EIG naturally fell into local minima. If the goal is to demonstrate resilience against heavily corrupted relative poses or outliers, the comparison is unfair unless matched against robustified versions of SDR or EIG.

---

> ### Author Rebuttal · Authors · 2026-03-31
>
> Thanks for your comments. We hope that the following discussion can clear up your concerns.
>
> **Q1. It remains ambiguous whether the primary contribution is faster convergence, superior scalability, or solely robustness to extreme noise.**
>
> The practical advantages of closing the eigenspace-sync gap are from exact geometric alignment, yielding both structural robustness and computational efficiency. First, standard matrix relaxations introduce artificial mathematical degrees of freedom  absent in the physical world. Dual quaternions natively mirror the exact rigid symmetries of 3D space. Due to this physical alignment, our formulation naturally decouples rotational estimation from large translation scales, enabling robust recovery even under extreme graph sparsity and severe noise. Second, because matrix eigenspaces are physically misaligned, they require costly, heuristic multi-stage rounding to force solutions back to SE(3). Conversely, our natively aligned representation reduces projection to a simple normalization with strictly controlled error (Lemma 2.5 and Proposition 2.7).
>
> **Q2. The real-world experiments lack soundness.**
>
> This question is similar to Reviewer few9 Q3. Please refer to the answer there.
>
> **Q3. ...baselines like SDR and EIG naturally fell into local minima. ... the comparison is unfair unless matched against robustified versions of SDR or EIG.**
>
> We respectfully clarify that neither EIG nor SDR suffers from local minima. SDR relies on a convex formulation that is globally optimal within its exactness regime; its deterioration under high noise simply indicates that the corruption level exceeds this regime, rather than a local trap. Similarly, EIG is a spectral decomposition where degradation reflects the inherent sensitivity of its relaxation and rounding pipeline. Regarding robustified baselines, techniques like EIG+IRLS employ an outer reweighting wrapper around the base estimator. In fact, Arrigoni et al. explicitly disable IRLS in fair real-data comparisons when competing methods are not robust. Therefore, comparing EIG+IRLS against plain SDR or plain DQGPM would be unfair, since it conflates the base solver with an additional robustification layer. A fair comparison is either base methods versus base methods, or robustified variants for all methods where such wrappers are available.
>
> **Q4. What is the fundamental advantage of optimizing rotation and translation jointly within the dual quaternion framework, as opposed to the standard practice of performing rotation averaging first and substituting the result to solve for translation?**
>
> The advantage of our joint DQ formulation is most apparent under extreme sparsity and high noise, which are common in real-world settings. First, unlike decoupled two-stage methods where rotation is estimated in isolation, joint estimation allows translation measurements to continuously feed back and correct rotation errors. This mutual anchoring is vital in highly sparse graphs, where rotation edges alone often lack the topological rigidity to prevent global collapse. Second, baselines like EIG rely on matrix representations where high translation noise quadratically overwhelms the rotation signal, and then require costly heuristic multi-step rounding to recover valid motions. In contrast, our DQ representation leverages the dual unit ($\epsilon^2=0$) to algebraically shield rotational updates from extreme translation noise. Furthermore, mapping back to the feasible set reduces to a simple normalization that is provably stable (Lemma 2.5, Proposition 2.7, Theorem 2.8).
>
> **Q5. SDR was excluded. Could you elaborate on the specific computational bottlenecks encountered, and why DQGPM does not suffer from them?**
>
> In the SDR method, after eliminating translations, SE-Sync reduces the problem to
> $$\min\limits_{R\in\mathrm{SO(d)}^n} \mathrm{tr} (QR^\top R),\ Q=L(G^\rho)+\Sigma-V^\top L(W^\tau)^\dagger V.$$
> The computational bottleneck of SDR under uniform edge dropout (ER graphs) comes from marginalization of translations, which introduces the Schur-complement term $V^\top L(W^\tau)^\dagger V$. Under uniform random sparsification, the algebraic connectivity $\lambda_2(L(W^\tau))$ drops significantly, causing the pseudo-inverse bound $||L(W^\tau)^\dagger||_2 = 1/\lambda_2(L(W^\tau))$ to explode. This not only causes severe ill-conditioning that stalls iterative solvers, but critically, computing this pseudo-inverse forms a dense matrix, destroying the problem's sparsity and inducing prohibitive $O(n^3)$ complexity (unlike structured SLAM graphs where sparse factorization remains viable). In fundamental contrast, DQGPM natively couples rotation and translation within a unit dual quaternion without any marginalization. Consequently, our state matrix inherently preserves the exact $O(|E|)$ sparsity pattern of the original measurement graph.

---

> > ### Author Rebuttal · Reviewer_PMXV · 2026-04-02
> >
> > I would like to thank the authors for their detailed responses, which have addressed most of my concerns. Consequently, I am raising my score to "Weak Accept."
> >
> > Regarding Q1, I now better appreciate the advantages of the proposed algorithm over SDR and EIG, particularly its robustness under high noise and extreme graph sparsity. However, I remain somewhat unclear about its computational efficiency and scalability relative to other methods. Additionally, since high noise and extreme sparsity rarely occur simultaneously in practical scenarios, I still have some concerns regarding its real-world applicability.
> >
> > I especially appreciate the clarification for Q4 and Q5; the performance degradation of SDR (both in accuracy and efficiency) under high sparsity is an insightful finding. Finally, I strongly encourage the authors to evaluate the proposed algorithm against current SOTA methods on large-scale Pose Graph Optimization (PGO) datasets to further demonstrate its practical utility.

---

> > > ### Author Response · Authors · 2026-04-05
> > >
> > > We sincerely thank the reviewer for the thoughtful follow-up and for raising the score. It is encouraging to know that our responses have helped clarify the advantages of the proposed method.
> > > Regarding concerns about computational efficiency and scalability, please refer to our detailed response to Question 3 from Reviewer 3vXk.
> > >
> > > **1. Real-world applicability under high noise or extreme sparsity**
> > >
> > > High noise and extreme sparsity often characterize different application contexts such as the extreme sparsity (>85\% missing edges) addressed in our 3DMatch point-cloud registration experiments. In highly degraded environments, front-end pipelines must aggressively filter ambiguous matches to prevent catastrophic outliers, inherently inducing extreme graph sparsity. Yet, because the underlying geometry remains ambiguous, the few edges that survive this strict culling still frequently suffer from significant heavy-tailed translation noise. Therefore, this combination is not an artificial edge case, but rather the exact structural bottleneck where standard 3D perception pipelines fail.
> > >
> > > **2. Evaluation on large-scale PGO datasets**
> > >
> > > We sincerely thank the reviewer for this excellent and highly constructive suggestion. We completely agree that evaluating on PGO datasets is a crucial step to demonstrate practical utility in SLAM. However, we respectfully clarify that incorporating standard PGO datasets requires a highly non-trivial data and algorithmic adaptations to our current framework, which we plan to rigorously address in a dedicated future work. The current formulation is a general synchronization framework based on Least Square formulation over dual quaternions (Problem (3) in our paper), which treats measurement over all edges with uniform noise level. However, PGO datasets contain information matrices that encode per-edge reliability as separate noise levels $\kappa_{ij}$, $\tau_{ij}$ for rotation and translation. In particular, the standard PGO problem takes the form $$\min_{t_i \in \mathbb{R}^3, R_i \in \mathrm{SO}(d)} \sum_{(i,j)\in E} \kappa_{ij}\Vert R_j - R_i R_{ij} \Vert_F^2 +
> > > \tau_{ij}\Vert t_j - t_i - R_i t_{ij} \Vert_2^2.
> > > $$
> > >
> > > To handle this within our framework, one needs to derive the transformed correlation matrices $W$ under the dual quaternion representation, replacing the least squares objective (3) with its weighted counterpart $$\min_{x\in \mathrm{UDQ}^n}(C-xx^\*)^\*W( C-xx^*),$$
> > > and design a corresponding weighted generalized power method to solve it. Correctly integrating the anisotropic information matrices of PGO datasets requires a highly non-trivial theoretical extension to map these uncertainties into dual quaternion weights.

---

### Official Review · Reviewer_few9 · 2026-03-12

**Soundness:** 3
**Presentation:** 3
**Significance:** 2
**Originality:** 3
**Overall Recommendation:** 4
**Confidence:** 3

**Summary:**

To address the representation gap and eigenspace–synchronization mismatch caused by conventional matrix embeddings of SE(3), this work adopts a unit dual quaternion representation and proposes a two-stage solution. A spectral initializer computed via the power method on a Hermitian dual quaternion measurement matrix, followed by a dual quaternion generalized power method that enforces feasibility through per-iteration projection. Experiments demonstrate improved robustness under sparse measurements and competitive accuracy and efficiency compared to matrix-based approaches.

**Compliance With Llm Reviewing Policy:**

Affirmed.

**Key Questions For Authors:**

1. How does the proposed method behave under heavy-tailed noise distributions or in the presence of gross outliers in the relative transformations? Demonstrating robustness in such regimes would further substantiate the claimed stability and modeling advantages.
2. Could the authors provide one or two concrete real-world application scenarios where this method can be directly integrated? Further releasing the code of these examples can improve reproducibility.

**Limitations:**

No. The authors could expand on practical constraints and discuss implications for machine learning applications.

**Strengths And Weaknesses:**

**Strengths:**

- S1: The paper identifies the limitation of matrix representation of SE(3). By adopting the unit dual quaternion, the work insightfully provides a more geometrically aligned representation where projection admits a closed-form, Lipschitz-stable operation.
- S2: Based on the two-stage solution, the paper establishes non-asymptotic estimation error bounds for both the spectral initializer and its projected feasible counterpart, and further proves linear error contraction of the proposed DQGPM up to an explicit noise-dependent floor.
- S3: The proposed work is validated on both synthetic settings with simulated noise and real multi-scan registration benchmarks, where it shows consistently better stability and accuracy with lower runtime.

**Weaknesses:**

- W1: The derivation is based on the assumption of operator norm bounds, and the experiment also simulates additive noise. It is unclear how the method behaves under heavy-tailed noise or outliers, which are common in practical estimation problems.
- W2: The baselines are primarily EIG and SDR methods, and SDR is excluded for some in some setting due to scalability. More recent or competitive nonconvex optimization approaches for SE(3) synchronization are not extensively benchmarked.
- W3: The real-world evaluation is limited to a single application scenario (point-cloud alignment). Given that SE(3) synchronization is fundamental to many practical problems, including additional real-world application would better demonstrate the generality and practical impact of the proposed method.

---

> ### Author Rebuttal · Authors · 2026-03-31
>
> Thanks for your comments. We hope that the clarification and additional experiments may help to clear the reviewer's concerns.
>
> **Q1. It is unclear how the method behaves under heavy-tailed noise or outliers, which are common in practical estimation problems. How does the proposed method behave under heavy-tailed noise distributions or in the presence of gross outliers in the relative transformations?**
>
> Thanks for raising this important point. We tested on heavy-tail noise generated by Gaussian mixture, where most noise are generated by Gaussian noise with small perturbation ($\sigma_r=10^\circ, \sigma_t=0.1$), and some proportion of outlier observation have Gaussian noise with larger perturbation. **Please refer to [PDF](https://anonymous.4open.science/r/dq_supplement_pdf-C728/supple_pdf.pdf) for the corresponding plots of the following experiments.** Figure 3 show the error versus outlier proportion with outlier follow $\sigma_r=40^\circ, \sigma_t=0.4$, and Figure 4 show the error versus perturbation level with outlier proportion 0.2. DQGPM has shown superior performance across all settings. However, we respectfully clarify the distinction between perturbative heavy-tailed noise and gross random contamination. Because our core $\ell_2$ objective strictly enforces valid $\mathrm{SE}(3)$ coupling, it cannot inherently reject completely arbitrary, unrelated rigid-motion outliers. Handling such gross contamination requires an outer robust reweighting or trimming layer (analogous to EIG+IRLS). Thus, DQGPM serves as a stable core solver that performs well under heavy-tailed perturbations, and can be naturally combined with standard robust wrappers to handle outlier rejection.
>
> **Q2. The baselines are primarily EIG and SDR methods, and SDR is excluded for some in some setting due to scalability. More recent or competitive nonconvex optimization approaches for SE(3) synchronization are not extensively benchmarked.**
>
> We agree with the reviewer that our initial evaluation primarily focused on classical EIG and SDR baselines. Directly comparable recent baselines for exact global SE(3) synchronization are relatively limited, as much of the recent literature focuses instead on pairwise registration or correspondence estimation (i.e., the stage that generates relative pose measurements) rather than the global synchronization problem studied here. To strengthen our empirical evaluation, we have incorporated a more recent and highly competitive baseline (SPEC) by Doherty et al. (2022) [1] across both synthetic and real-world experiments. While SPEC proves to be generally stronger than classical EIG and SDR methods, our DQ approach remains highly competitive and consistently demonstrates robust performance against this stronger benchmark. **Please see Tables 1 and 2 in [PDF](https://anonymous.4open.science/r/dq_supplement_pdf-C728/supple_pdf.pdf) for synthetic experiments, and Table 3 for real-data results.**
>
> [1] Doherty, Kevin J., David M. Rosen, and John J. Leonard. "Performance guarantees for spectral initialization in rotation averaging and pose-graph SLAM." 2022  International conference on robotics and automation (ICRA). IEEE, 2022.
>
> **Q3. Could the authors provide one or two concrete real-world application scenarios where this method can be directly integrated? Further releasing the code of these examples can improve reproducibility.**
>
> We thank the reviewer for the suggestion. To this end, we integrated our method into a real-world pipeline for large-scale indoor scene reconstruction on the 3DMatch dataset. Unlike object-level registration, this setting involves extreme measurement sparsity (over 85% missing entries after filtering unreliable partial overlaps), where local errors easily accumulate and cause standard solvers to drift globally. As a drop-in replacement for the backend pose-graph optimizer, our method avoids structural collapse and yields more accurate and visually consistent reconstructions than competing baselines. **Please see Figures 1 and 2 in [PDF](https://anonymous.4open.science/r/dq_supplement_pdf-C728/supple_pdf.pdf).** Finally, We will also release the code for these experiments in future versions to further improve reproducibility.

---

### Decision · Program_Chairs · 2026-04-30

**Decision:**

Accept (regular)

**Comment:**

This submission makes a clear and original contribution by formulating SE(3) synchronization in the dual quaternion space and by providing recovery guarantees for a principled two stage solver. The reviewers consistently praised the theoretical strength and the strong empirical behavior under sparse and noisy settings, and the rebuttal addressed most concerns about baselines, robustness, and practical relevance, leading to a broadly positive consensus. The main remaining weaknesses are the dense presentation and the still limited large scale real world validation, but these do not outweigh the paper’s technical merit and potential value to the community.